# SARAMIS: Simulation Assets for Robotic Assisted and Minimally Invasive Surgery

**Nina Montaña-Brown**[*,1,2]     **Shaheer U. Saeed**[1,2]     **Ahmed Abdulaal**[1]     **Thomas Dowrick**[2]
**Yakup Kilic**[3]     **Sophie Wilkinson**[3]     **Jack Gao**[3]     **Meghavi Mashar**[3]
**Alkisti Stavropoulou**[1]     **Emma L. Thomson**[1]     **Zachary MC Baum**[1,2]     **Chloe He**[1,2]
**Simone Foti**[1,2]     **Brian Davidson**[3]     **Yipeng Hu**[1,2]     **Matthew J Clarkson**[1,2]

[1] Centre for Medical Image Computing, UCL, London, United Kingdom
[2] Welcome/EPSRC Centre for Interventional And Surgical Sciences, London, United Kingdom
[3] University College London Hospitals
[*] n.montanabrown@cs.ucl.ac.uk

## Abstract

Minimally-invasive surgery (MIS) and robot-assisted minimally invasive (RAMIS) surgery offer well-documented benefits to patients such as reduced post-operative pain and shorter hospital stays. However, the automation of MIS and RAMIS through the use of AI has been slow due to difficulties in data acquisition and curation, partially caused by the ethical considerations of training, testing and deploying AI models in medical environments. We introduce `SARAMIS`, the first large-scale dataset of anatomically derived 3D rendering assets of the human abdominal anatomy. Using previously existing, open-source CT datasets of the human anatomy, we derive novel 3D meshes, tetrahedral volumes, textures and diffuse maps for over 104 different anatomical targets in the human body, representing the largest, open-source dataset of 3D rendering assets for synthetic simulation of vision tasks in MIS+RAMIS, increasing the availability of openly available 3D meshes in the literature by three orders of magnitude. We supplement our dataset with a series of GPU-enabled rendering environments, which can be used to generate datasets for realistic MIS/RAMIS tasks. Finally, we present an example of the use of `SARAMIS` assets for an autonomous navigation task in colonoscopy from CT abdomen-pelvis scans for the first time in the literature. `SARAMIS` is publically made available at `https://github.com/NMontanaBrown/saramis/`, with assets released under a CC-BY-NC-SA license.

## 1   Introduction

Laparoscopy and endoscopy are techniques in surgical and medical practice which involve inserting video cameras into a patient in order to diagnose and treat a number of conditions, and have made it possible to perform minimally invasive surgery (MIS). These techniques obviate the need for large incisions at the operative site, replacing them with small incisions into which cameras and tools are inserted to perform the intervention. The benefits of MIS have been well documented [8, 49, 71], and can be summarised as follows: 1) Reduced post-operative pain, 2) Shortened hospital stays [71, 47], 3) Improved rates of patient recovery [12], and 4) Lowered costs to hospital systems in a number of interventions [8, 66, 47, 22]. Additionally, recent advances in robotics have enabled the pairing of robotic elements with laparoscopic equipment, which provides further benefits such as an improved ergonomic environment for surgeons [76] and the possibility of teleoperation [11]. In tandem, (partially) autonomous robotic surgery has emerged as an increasingly important research

37th Conference on Neural Information Processing Systems (NeurIPS 2023) Track on Datasets and Benchmarks.

topic [10, 62, 26]. Indeed, many surgeons consider the full automation of robot-assisted minimally invasive surgery (RAMIS) as the 'end goal' of surgical practice [26].

Although there have been advances in technologies to facilitate both MIS and RAMIS [58] –such as image overlay [64, 37, 77, 7], or 3D localisation of tools and cameras relative to a pre-op scan [1]– the data collection and validation of such solutions has been limited by the equipment required for validation. This can include, for example, optical trackers, stereo cameras, and/or LIDAR-like sensors which are non-standard surgical objects that interrupt surgical workflow and are expensive to accrue and implement [27, 10].

Whilst traditional computer vision applications have long exploited such devices to create large-scale annotated datasets for relevant tasks such as camera-pose estimation or scene-reconstruction [23, 42], the aforementioned difficulties regarding surgical logistics (e.g. sterilisation of all objects in theatre requiring repeated calibration, time-sensitivity of surgical environments, and overhead equipment cost) have resulted in limited datasets for these tasks in MIS/RAMIS. In parallel, synthetic data and rendering environments have emerged as promising, alternative resources to enable computer vision at scale [53, 68], and are important for the development and testing of safe autonomous systems. However, *in silico* datasets for the development of deep learning algorithms and autonomous systems in MIS/RAMIS are limited in number and application [38].

In this paper, we introduce Simulation Assets for Robotic Assisted and Minimally Invasive Surgery (SARAMIS), the first large-scale, multi-organ, open-source collection of rendering assets for the simulation of robotic and minimally invasive surgery. We summarise the contributions of this work as follows:

- We provide the first, large-scale database of patient-data-derived rendering assets representing anatomical organs, textures, and tetrahedral meshes for the simulation of abdominal minimally invasive interventions.

- We integrate SARAMIS with existing open-source environments for the procedural simulation of endoscopic and laparoscopic procedures, including simulation of depth maps, stereo and monocular cameras.

- We develop a Markov decision process environment for navigation within the colon, using the above-described simulations, and subsequently use this environment to train an autonomous reinforcement learning (RL) function which learns to navigate to four different structures within the colon and is generalisable to different patient cases; we open-source this environment for further research and development.

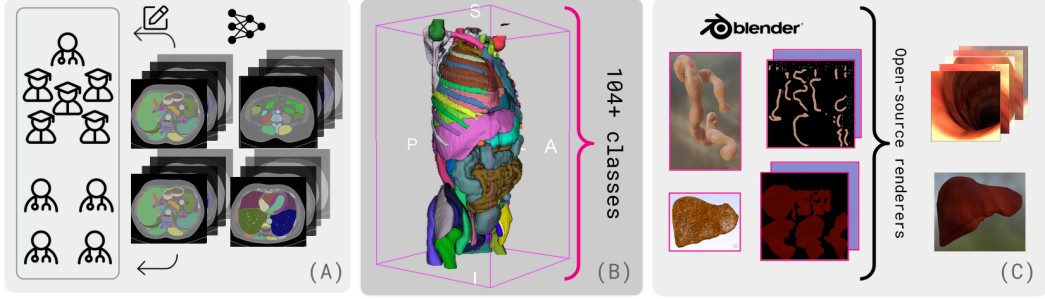

Figure 1: Summary of SARAMIS pipeline. We annotate a large-scale dataset of open-source CT scans (A), remesh and convert them into simulation files (triangular and tetrahedral meshes, normal maps, diffuse maps) (B) that can be used with a open-source renderers (C) to produce synthetic training data for MIS applications.

## 2  Related Work

**Open Source MIS Datasets**    In contrast to tasks such as 3D medical image segmentation [45, 35], the number of freely available, annotated datasets focussed on MIS tasks is small [10]. Most available datasets focus on 2D segmentation from intra-operative images for tasks such as organ [9, 30],

pathology [5, 41], and tool [61] segmentation, as well as action recognition[74]. Datasets to validate steps in MIS pipelines which are critical to workflow automation, such as camera pose estimation, 3D-to-3D registration, and organ deformation, are comparatively limited. The lack of varied datasets can be attributed to the comparatively high cost of label acquisition and cleaning [46, 39], which involves the introduction of (previously discussed) non-standard equipment into the surgical workflow [39]. Furthermore, unlike traditional computer vision applications, deformable object modelling is a prerequisite to achieving clinically relevant accuracies [59, 70, 63, 14]. Whilst animal models [48] may be used to validate algorithms through the use of irradiating scans, few patient open-source datasets to validate deformation models exist in the literature [59, 36].

Synthetic datasets for applications related to MIS represent an alternative approach, with promising results in terms of simulation-to-real transfer for deformation simulation [55, 54, 63, 57, 70, 67, 69], segmentation [16, 15] and depth estimation [60, 72]. However, current work either does not release 3D assets to simulate or manipulate scenes [54] or uses non-open-source frameworks [60, 16] and can be limited in terms of application [60, 72, 16, 70, 69]. Existing work is further limited in the number of anatomical variations of 3D assets due to the use of a small cohort to produce the datasets [60, 36, 16, 67, 69]. We summarise the existing literature of 3D assets of simulation of MIS+RAMIS tasks in Table 1.

| Dataset | # 3D Assets | # Organs | # Subjects | Open-Source? |
|---|---|---|---|---|
| CV3D [72] | 1 | 1 (Colon) | 1 | ✗ |
| Dowrick et al. [16] | 1 | 1 (Liver) | 1 | ✗ |
| Tagliabue et al. [70] | 1 | 1 (Tissue Retraction) | 1 | ✗ |
| Suwelack et al. [69] | 1 | 1 (Liver) | 1 | ✗ |
| DEPOLL [59] | 2 | 1 (Porcine Liver) | 2 | ✔ |
| Dowrick et al. [15] | 1 | 1 (Liver, Colon ) | 1 | ✔ |
| OpenHELP [36] | 18 | 18 | 1 | ✔ |
| SimCol [60] | 1 | 1 (Colon) | 1 | ✔ |
| IRCAD 3D Liver Dataset [67] | 20 | 1 (Liver) | 20 | ✔ |
| **SARAMIS** | **114,838** | **106** | **2527** | ✔ |

Table 1: Summary of existing 3D datasets for simulation of MIS+RAMIS tasks in the literature as compared to SARAMIS

**Simulation Environments for MIS**    Rendering frameworks are abundant in the computer vision literature [24, 20, 34], and there are a number that support physics-based multi-object rigid-body interactions [73, 20, 24]. Whilst there has been an interest in deformable-object interactions in computer vision [3, 43], robotics, and MIS tasks [55, 54], the majority of these works are not open-source [70], or have limited support for realistic soft-body interactions [73]. Frameworks that use finite-element modelling, required for realism and accuracy in MIS/RAMIS, are limited [70, 21, 2].

**Autonomy in RAMIS**    Several advances have been made towards task-level automation in the field of RAMIS [28], with reported success in tasks such as path planning [65], suturing of various structures [62, 40], and tissue retraction in an *ex vivo* environment [63]. However, there exist significant ongoing ethical questions surrounding the regulation and deployment of autonomous surgical systems [51]. This is especially the case for more complex tasks such as navigation or full surgical automation. Many studies evaluate tasks in phantom models [62], animal models [62], or a limited number of synthetic patient-specific models [63]. Synthetic assessment environments are promising, but much like the MIS simulation environment literature, these suffer from very small patient cohorts to generate the datasets [38] and thus a limited representation of anatomical variance.

SARAMIS tackles a number of these issues in important ways. It provides one of the largest dataset of heterogenous patient-derived meshes to date, with a total of 116,018 meshes from 2529 patient models over more than 104 different anatomical structures. Additionally, SARAMIS may be paired with commonly-used rendering environments to sample monocular video with different camera intrinsics, depth maps, pose labels, optical flow, and segmentation maps, such as Blender-based Kubric [24], or Mitsuba3 [34] - we provide examples of interfacing the SARAMIS assets with Mitsuba3, Kubric, and PyBullet, which can leverage GPU simulation of finite-element modelling or particle-based dynamics deformation simulation, for RL or otherwise.

# 3 Dataset Generation

**Data Collection and Annotation**  Three open-source, anonymised, medical image datasets of patient CT scans [45, 35, 75] were selected for analysis. A human-in-the-loop, semi-automatic data annotation strategy (Fig. 1, Panel A) was used to generate 3D rendering assets from patient-specific CT scans using 3DSlicer [56], PyTorch [52], and MeshLab [13]. Initially, CT scans were automatically segmented with TotalSegmentator [75], which is composed of several nnU-Nets [32] trained to detect 104 anatomical labels from CT scans. Given that the TotalSegmentator dataset contains 3D segmentations for all anatomical organs of interest verified by a radiologist, only the Abdomen-1k [45] and AMOS [35] datasets are processed using the segmentation pipeline. The generated labels were assessed with a collaborative-iterative strategy involving seven trained anatomical annotators and four radiologists. Initially, all annotations were inspected by the anatomical annotators under the supervision of a clinician. The following criteria were adopted to flag cases in need of further review: 1) Verify class homogeneity within an anatomical structure, 2) Flag topological errors (e.g., slices missing, holes within an anatomical structure), 3) Flag under- or over-segmentation, and 4) Flag potential pathology. Additionally, annotators were instructed to log the type of CT scan from the data {full-body (FBCT), chest-abdomen-pelvis (CTCAP), abdomen-pelvis (CTAP), abdominal (ACT)}. The full annotation protocol, including training practice for the junior annotators, is made available in the Supp. Mat. (Appendix B Datasheet for Datasets). Subsequently, cases that were flagged for potential errors were individually reviewed and manually corrected under the supervision of a clinician. Finally, a subset of 450 of the verified CT scans were reviewed by four radiologists, instructed to verify the correctness of the segmentations and note any other relevant pathology or errors.

**Colon Mesh Generation**  Human bowel segmentation in CT is a challenging task due to a combination of tortuous anatomy and inconsistent contrast (itself due to the air-fluid interfaces in the colon), which can result in an incomplete tubular segmentation in non-contrast enhanced CT scans. Therefore, in order to generate more realistic and continuous mesh models that are tubular, a procedural generation approach was considered instead [15]. All colon models from FBCT, CTCAP, and CTAP scans were manually inspected in 3DSlicer to detect the presence of the rectum, hepatic and splenic flexures, and the caecum. Models were then categorised as complete segmentations (all landmarks detected and a full segmentation is obtained), partial segmentations (all landmarks are detected, but may be disconnected in regions), or erroneous segmentations ($\geq 1$ of the previous landmarks missing). Complete and partial segmentations were then manually processed with the Vascular Modelling Toolkit (VMTK) [33] in order to extract the colon centerlines. This was achieved by the manual placement of landmarks at the beginning and end of distinct regions (e.g., the beginning and end-point of the rectum) in order to extract line segments describing the tubular structure of that region. A matching algorithm was used to find a continuous line segment describing the entire colon by defining a start point on each colon, the subsequent closest segments are matched one by one (Appendix A, Supp. Mat.). A BSpline curve was then fit to the data points for each colon to obtain a smooth representation of the centerline and resampled to 1000 points each. We provide generic interfaces from which the extracted curves can be converted into mesh representations of colons. We consider the colon topologically as a closed tube - by extruding the mesh along the centerline with varying radius parameters, we can obtain a patient-derived representation of the colon. Full procedural simulation parameters are summarised the Supplementary Materials, and provided open-source for future research.

**Mesh Generation**  All anatomical segmentations bar the colon were automatically extracted and converted into 3D mesh (Fig. 1, Panel B) models using a marching cubes algorithm [44]. Given the voxelised nature of CT scans with varying resolution between 0.5-5mm, the resulting meshes were post-processed using Laplacian smoothing [18]. Additionally, meshes are mean-centered in their patient frame-of-reference as well as their local frame-of-reference. Finally, patient frame-of-reference meshes are converted into tetrahedral volumes using fTetWild [31].

**Texture and normal mapping**  We procedurally generate bone, bowel, soft abdominal organ, and muscle normal and diffuse maps using Blender. Based on open-source images of the aforementioned textures, we create Shader nodes (Supp. Mat. Appendix D) in Blender using Principled BSDF nodes to replicate the visual appearance of the structures. Subsequently, each mesh, it's corresponding

Table 2: Summary of CT data of three datasets from which `SARAMIS` is derived. FBCT = Full Body CT, CTCAP = chest-abdomen-pelvis CT, CTAP = abdomen-pelvis CT, ACT = Abdomen CT. Other refers to a alternative CT scans, as described in the datasheet for [75].

| Dataset | Initial | Type of CT Scan | | | | | Included | No changes |
|---|---|---|---|---|---|---|---|---|
| | | FBCT | CTCAP | CTAP | ACT | Other | | |
| Abdomen-1k | 1063 | 10 | 366 | 71 | 592 | 0 | 1048 | 526 |
| Amos | 600 | 0 | 72 | 220 | 0 | 0 | 321 | 140 |
| TotalSegmentator | 1200 | 169 | 197 | 110 | 0 | 724 | 1200 | 1200 |
| `SARAMIS` | 2863 | 179 | 635 | 401 | 592 | 724 | 2527 | 1866 |

normal maps and diffuse characteristics are baked in 2k resolution. Example renders can be visualised in Fig 2 B).

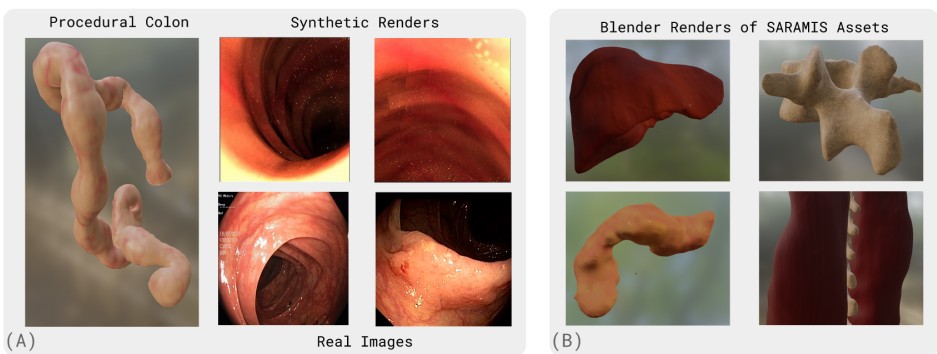

Figure 2: Textured and shaded assets from `SARAMIS`. In A), we render a procedurally generated colon, with two examples of synthetic renders of the colon, as well as reference real images from the HyperKvasir dataset [6]. We showcase other assets from `SARAMIS` in B), namely the liver (top left), vertebrae (top right), pancreas (bottom left), and muscle (bottom right).

## 4 SARAMIS

### 4.1 3D Dataset Generation

Overall, `SARAMIS` consists of a total of 114,838 meshes, textures, and normal map tuples that are derived from a total of 2527 patient scans. From the initial 2863 scans, a total of 336 were excluded from segmentation analysis for the following reasons: 194 due to lack of availability of test set label, 15 due to significant pathology making organ differentiation difficult, 13 due to the presence of fluid in the abdomen (e.g. haemoperitoneum or ascites) occluding organs of interest, 100 due to alternative imaging modality (MRI), 2 due to metallic artefacts in the scan, 1 due to a poor quality scan, and 1 due to original file corruption leading to lack of a segmentation file. In total, 15, 321, and 0 scans were excluded from the Abdomen-1k, AMOS, and TotalSegmentator datasets, respectively. Overall, this results in the inclusion of 1048, 279, 1200 scans from the Abdomen-1k, AMOS, and TotalSegmentator datasets in the `SARAMIS` dataset. The average voxel resolution is $[0.77 \times 0.77 \times 3.26] \pm [0.13 \times 0.14 \times 1.64]$mm (mean voxel size for Abdomen-1k, AMOS and TotalSegmentator $[0.81 \times 0.81 \times 2.70]$, $[0.70 \times 0.70 \times 4.23]$, $[0.70 \times 0.70 \times 4.23]$mm each, respectively). The data split is documented in Table 2, and the dataset split by mesh type is documented in Fig 3. The resulting dataset and full compilation, processing, and texturing instructions are provided in the `SARAMIS` Datasheet for Datasets (Datasheet for Datasets, Supp. Mat.).

In total, 637 (48 %) of reviewed scans required correction. In addition, 78 scans (17% of the meta-reviewed subset) were flagged by the radiologists and required additional corrections. A total of 343,495,425 voxels were edited, where 18,526,556 voxels were corrected in the AMOS dataset, and 324,968,869 voxels were corrected in the Abdomen-1k dataset. Where corrections were necessary, a

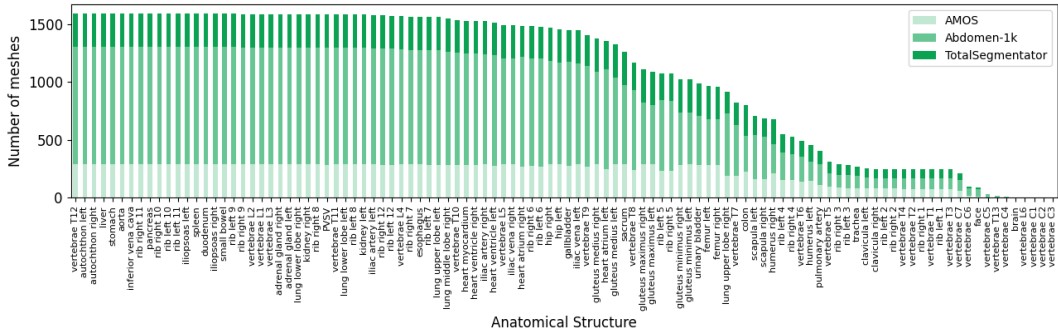

Figure 3: Number of meshes per organ in SARAMIS, split by constituent datasets.

median 27,924 [IQR=7239, 98857] voxels were corrected per scan, with an average 12.6 [IQR=3.0, 15.0] structures corrected per scan. The most commonly corrected structure was the liver (315 instances corrected), with the least corrected structures being vertebrae C1, C2, and C3, and the brain (1 correction each). In addition, two previously unseen labels were corrected in the dataset: L6 and T13, denoting transitional vertebrae (which can result e.g. from congenital spinal deformity, resulting in additional or fewer lumbar or thoracic vertebra). A total of 48 scans were flagged as having transitional vertebra, with a total of 8 L6 segments, 26 T13 segments, and 14 as sacralised L5. A full description of organs changed per dataset is supplied in the Supp. Mat. Appendix I. Additional analyses of mesh density, surface area, and vertex are provided in Supp. Mat. Appendix G.

## 5 Autonomous Navigation with Colonoscopy

The SARAMIS dataset provides a reference set of data to simulate intraoperative navigation tasks. These simulations may then be used to train autonomous agents to navigate within the anatomies of interest. One such example explored in this work, specifically in the application of colonoscopy, is detailed in the following subsections.

### 5.1 Methods

In this work, the navigation task in colonoscopy is formulated as a sequential decision problem modelled by a finite horizon partially observable Markov Decision Process (MDP). The decision policy is learnt using RL as described in the following sections. The navigation is performed based on an image acquired from a camera inside the colon, where the task is for the camera to navigate to a desired target which is visible from the camera pose.

#### 5.1.1 The Markov decision process environment

The MDP environment for RL is modelled as a tuple $(\mathcal{S}, \mathcal{A}, p, r, \pi, \gamma)$.

**States** Here, $\mathcal{S}$ is the state-space from which a state at time-step $t$ may be sampled $s_t \in \mathcal{S}$. In our formulation $s_t$ is an image acquired from a synthetic camera. In this work, the image is simulated using Mitsuba3 [34], using the previously generated textures and diffuse maps.

**Actions** The pose of the camera is defined as $c_t \in \mathbb{R}^6$ and a change in the pose is defined as the action $a_t \in \mathcal{A} \in \mathbb{R}^6$, where $\mathcal{A}$ may be denoted as the continuous action space, such that $a_t$ is the sampled action at time-step $t$. The updated pose may then be defined as $c_{t+1} = c_t + a_t$, which is the pose at which the new camera image $s_{t+1}$ is rendered. The state transition distribution conditioned on state-action pairs is given by $p : \mathcal{S} \times \mathcal{S} \times \mathcal{A} \to [0, 1]$ where $p(s_{t+1}|s_t, a_t)$ denotes the probability of the next state $s_{t+1} \in \mathcal{S}$ given the current state $s_t \in \mathcal{S}$ and action $a_t \in \mathcal{A}$ pair.

**Rewards** The reward function $r : \mathcal{S} \times \mathcal{A} \to \mathbb{R}$ produces a reward at time-step $t$ denoted by $R_t = r(s_t, a_t)$ given the current state $s_t$ and action $a_t$ pair. In our formulation the reward is formed of two parts: 1) $r_{dist}(\cdot)$ which is the inverse of the distance between the camera position defined by $c_t$

and the target (clipped to prevent finding only the target centre); 2) $r_{image}(\cdot)$ which tests conditions with the help of the image plane position and camera position, as follows:

$$r_{image}(s_t, a_t) = \begin{cases} -1 & \textbf{if } \text{target not in image } s_t \\ +10 & \text{(and terminate episode) } \textbf{if } \text{target in image } s_t \\ -10 & \text{(and terminate episode) } \textbf{if } \text{camera intersects with wall} \end{cases} \tag{1}$$

where the target detection in the image $s_t$ is done by checking the intersection of the camera line of sight with coordinates of the target structure (a sphere placed in the region of interest), both computed based on $c_t$ and additional preset camera parameters. The wall intersection of the camera is computed using $c_t$ and a tolerance from centre-line coordinates of the colon. The final reward function $r$, is then given by $R_t = r(s_t, a_t) = r_{image}(s_t, a_t) + r_{dist}(s_t, a_t)$. This reward is scaled in order to balance the constituent rewards, with further details found within the implementation (Supp. Mat.). The episode termination with high reward values triggered by the 'target in image $s_t$' and 'camera intersects with wall' conditions prevents undesirable solutions e.g., navigating to structures through walls or hovering around a target to maximise the distance-based reward.

### 5.1.2 The policy

The policy $\pi(a_t|s_t) : \mathcal{S} \times \mathcal{A} \in [0, 1]$ denotes the probability of performing an action $a_t$ given state $s_t$. An action may then be sampled using $a_t \sim \pi(\cdot)$.

Following the state transition distribution $p$ and the policy $\pi$, for sampling next states and current actions, respectively, together with the reward function $r$, we can generate a trajectory of collected states, actions and corresponding rewards for multiple time-steps $(s_1, a_1, R_1, \ldots, s_T, a_T, R_T)$.

If we consider the policy $\pi_\theta$ to be parameterised by policy parameters $\theta$ then our aim is to find the optimal parameters $\theta^*$ such that if you follow $\pi_{\theta^*}$, the accumulated reward $r$ is maximised.

In practice the policy may be modelled as a neural network with parameters $\theta$, that predicts a distribution, from which to sample the action $a_t$. Practically, for continuous actions, the policy may be defined by two parametric functions (neural networks) with shared parameters, which specify a diagonal Gaussian distribution from which to sample the action; one function specifying the mean of the distribution $\mu = \mu_\theta(s_t)$ and one specifying the standard deviation $\sigma = \sigma_\theta(s_t)$. The policy may then be given by $\log \pi_\theta(a_t|s_t) = -\frac{1}{2} \left( \sum_{i=1}^{k} \left( \frac{(a_{t,i} - \mu_i)^2}{\sigma_i^2} \right) + k \log 2\pi \right)$. However, for notational convenience in further analysis, we simply use $\pi_\theta$ instead of modelling two separate networks that predict the parameters of the diagonal Gaussian distribution.

A cumulative reward over a trajectory may be used to compute optimal policy parameters $\theta^*$. The cumulative reward may be computed as a discounted sum of rewards over a trajectory, starting from time-step $t$ and is given by:

$$Q^{\pi_\theta}(s_t, a_t) = \sum_{k=0}^{T} \gamma^k R_{t+k} \tag{2}$$

where the discount factor $\gamma$ discounts future rewards. An expectation of this cumulative reward may be denoted as the return:

$$J(\theta) = \mathbb{E}_{\pi_\theta}[Q^{\pi_\theta}(s_t, a_t)] \tag{3}$$

which may be computed over multiple trajectories.

The optimisation problem may then be summarised as:

$$\theta^* = arg \max_\theta J(\theta), \tag{4}$$

and the objective function is maximised using gradient ascent.

The training procedure to obtain an optimised policy $\pi_{\theta^*}$ which maximises the cumulative reward, representative of navigation performance, is summarised in the Supplementary Materials (Appendix J "Reinforcement Learning Training Algorithm"). After training, this policy may be used to perform navigation intraoperatively.

## 5.2 Experiments and results

**Data**   A total of 155 colon meshes were selected from the TotalSegmentator subset of the `SARAMIS` dataset. To define navigation targets for the RL experiment, a single colon was manually labelled for all anatomically relevant landmarks (namely the rectum, hepatic and splenic flextures, and the caecum). Subsequently, the other 154 colons were deformably registered to the manually labelled colon in order to obtain anatomically appropriate labels for these regions. Registration was performed using Coherent Point Drift [50] ($\alpha = 1, \beta = 10$) in order to propagate label annotations through the dataset. An analysis of registration accuracy is provided in Supp. Mat. Appendix H. All centerline labels were then mapped onto the procedural mesh using kd-Tree [4] search (n=20), given the sufficiently dense procedural meshes. This defined the navigation areas on the surface mesh for each patient. The subset was split into 91 meshes used for model training, 32 for model development and 32 meshes as a hold-out test set. Images for navigation were subsequently simulated with a Mitsuba3 renderer, with size 200x200 pixels. Full hyperparameters are available in the provided implementation (Supp. Mat. Appendix A).

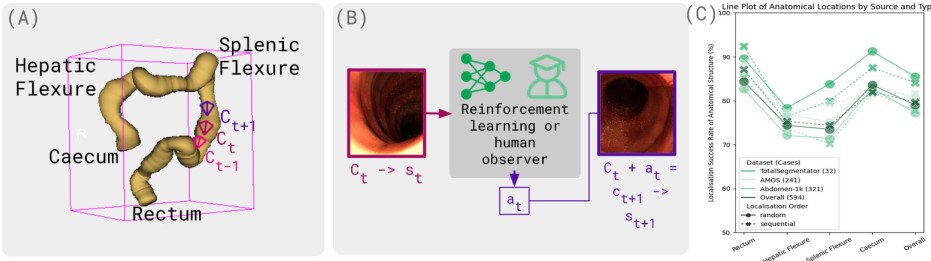

Figure 4: Summary of autonomous navigation experiment. Given a patient-derived mesh model of the colon, and defined navigation targets (A), the camera pose in the environment is used to render the synthetic view inside the colon. Using the rendering as the state $s_t$, a human observer or an RL agent may sample action $a_t$ with the aim of reaching navigation structures (B). We report the success rates by sub-dataset on the hold-out test set in (C), showing good generalisation accross unseen test sets.

**Evaluating RL policy vs a human observer**   To evaluate the efficacy of the autonomous navigation, we compare the RL-learnt policy with the policy of a human observer ([S.U.S.], biomedical imaging researcher with 4 years of experience with medical imaging). The efficacy is evaluated by the number of steps taken to reach the target, across four patient volumes which were not encountered during training. The human observer policy was within the same RL environment where interaction with the environment was done by sampling actions, where the action space was limited and the step size for $a_t$ was fixed (i.e., movement allowed in only orthogonal directions to the camera line of sight, controlled using arrow keys; and camera rotation controlled in the same orthogonal planes, controlled using 'W, A, S, D' keys). A visualisation of the rendered scenes from the environment are presented in Fig. 4. Example trajectories are generated in a representative navigation task from the rectum to the caecum in the Supp. Mat. Appendix. E. The average number of steps to find targets for RL and the human over 24 test cases were $77.8 \pm 13.2$ and $75.3 \pm 15.6$, respectively. There was no statistically significant difference in RL vs human performance (p-value= 0.83).

**Evaluating RL policy success rate for navigation**   We report the success rate of finding all four targets, across 594 different patient cases held out from training, with 100 randomised starting locations for each. For the random start locations experiment, a random location was picked as the starting point before navigating to the next structure. This was repeated 100 times for each patient case. For the sequential localisation task, the order in which to visit the 4 target locations was randomised 100 times per case. If the RL function failed to localise the structure, a random starting location was assigned for localising the next structure. Failure in the task is defined as the inability to navigate to the structure within 256 steps or collision with the colonic wall. Results are presented in Table 3, and performance split by sub-constituent dataset is reported in Fig. 4C. It should be noted that the lowest success rate is for the hepatic flexure and splenic flexure localisation tasks and the highest success rate is observed for the rectum localisation task. We observe a small performance decline from the TotalSegmentator datasets in comparison to the unseen datasets during training (Fig. 4 C).

| Structure | Rectum | Hepatic Flexure | Splenic Flexure | Caecum | Overall |
|---|---|---|---|---|---|
| Random start locations | 84.4 | 74.4 | 73.5 | 83.6 | 79.0 |
| Sequential localisation | 87.1 | 75.3 | 74.4 | 82.0 | 79.7 |

Table 3: Overall success rate (%) of navigating to a structure within the colon for 594 held-out subjects across the AMOS, Abdomen-1k and TotalSegmentator datasets.

The RL policy took approximately 7 days to train on a single Nvidia Tesla V100 GPU. During inference the model predictions coupled with an environment had a speed of approximately 20 iterations per second.

# 6   Discussion

SARAMIS presents the first, large-scale dataset of patient-specific 3D rendering assets representing the major structures of the visceral anatomy. As reported in Table 1, in comparison to previous works, SARAMIS offers the following distinct advantages: 1) Scale: SARAMIS is over three orders of magnitude larger than any previous set of 3D rendering assets for MIS in the literature, 2) Heterogeneity: SARAMIS offers an order of magnitude larger number of anatomical targets than previous datasets, and 3) Patient variability: SARAMIS features a significantly larger number of subjects compared to previous datasets. Through multi-organ segmentation, the creation of new labelling for all assets, data curation, and camera path generation, SARAMIS enables simulation-based experimentation not possible from the underlying CT scans alone.

In addition to SARAMIS, we developed a Markov decision process environment for navigation within the colon using simulated intraoperative images derived from patient CTAP scans (i.e., not from dedicated CT colonography scans). The observed performance of the RL function and human observer in the colonoscopy navigation task across four patients was comparable, which indicates that an effective generalisable cross-patient navigation policy was learnt using our proposed training scheme. Furthermore, it is interesting to note the overall high (79.0% and 79.7%) success rates of finding structures within the colon, within 256 steps, for the held out test set across a variety of randomised starting locations for the intraoperative imaging probe. The highest success rates were observed for the rectum and caecum, possibly due to their distinct appearances compared to the remainder of the colon, and tight curvature and blind-loop nature of the colonic flexures. While we model wall intersection constraints within our work, we do not account for all possible constraints in the endoscopic settings - for example, camera pose constraints such that the camera may not face directly opposite the direction of endoscope insertion from one step to the next, or extra-luminal boundaries imposed by surrounding visceral organs. Additionally, we qualitatively observe (Supp. Mat. Appendix E) that human trajectories are smoother than RL trajectories, which may arise from the lack of a smoothness prior or regularization on the generated actions. Accounting for these constraints represents a natural avenue of future research. Overall, whilst training was performed on a subset of the available assets, this indicates the robustness of the proposed training scheme which allows for the trained policy to be generalisable not only across patients but also across starting locations and the four target structures included during training.

# 7   Limitations

A limitation of this work lies in the design of shading nodes for the procedural texturing of SARAMIS assets. Despite designing the anatomical textures under the supervision of a clinician with surgical training and by referencing 2D intra-operative images of different anatomy, it is likely that deviations from the proposed parameters in the associated shader nodes may result in non-clinically feasible renders. Whilst procedural texturing is still an industry standard in the computer-graphics community [17, 29, 19], this manual approach could be paired or replaced with learning-based texturing [25] in order to texture SARAMIS meshes in a data-driven way. Another important limitation of this work is that the dataset does not capture all structures of relevance for the full simulation of MIS/RAMIS scenes (e.g. ligaments, fatty tissue, and fluids), due to the limited resolution of CT scans, and its relatively poor detection of soft-tissue structures. This limitation presents a natural avenue for future research, which could facilitate using SARAMIS to simulate different autonomous navigation tasks within the human abdominal anatomy.

## 8 Conclusions

In this work we introduce SARAMIS: Simulation Assets for Robotic-Assisted and Minimally-Invasive Surgery, a large-scale dataset of 3D rendering assets composed of 3D meshes, textures and diffuse maps for over 104 human anatomical structures. We warmly invite the wider research community to use SARAMIS assets for vision tasks in RAMIS/MIS such as depth estimation, camera pose estimation, or pairing tetrahedral meshes with open-source deformation modelling environments [20, 24] to further develop surgical vision applications and autonomous navigation tasks in MIS/RAMIS research.

## Acknowledgments and Disclosure of Funding

This work is supported by the Wellcome/EPSRC Centre for Interventional and Surgical Sciences [203145Z/16/Z]. NMB, AA, ET, AS, and SF are supported by the EPSRC-funded UCL Centre for Doctoral Training in Intelligent, Integrated Imaging in Healthcare (i4health) [EP/S021930/1]. AA is supported by an EPSRC Industrial Case grant [EP/W522077/1], and a Microsoft Research PhD Scholarship Wellcome Trust award [221915/Z/20/]. MJC, YH, NMB, and SUS are supported by [EP/T029404/1]. TD is supported by [EP/V052438/1]. ZMCB is supported by the Natural Sciences and Engineering Research Council of Canada Postgraduate Scholarships-Doctoral Program, and the University College London Overseas and Graduate Research Scholarships. This work is also supported by the International Alliance for Cancer Early Detection, an alliance between Cancer Research UK [C28070/A30912, C73666/A31378], Canary Center at Stanford University, the University of Cambridge, OHSU Knight Cancer Institute, University College London and the University of Manchester.

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
