# "SARAMIS: Simulation Assets for Robotic Assisted and Minimally Invasive Surgery" Supplementary Materials

**Nina Montaña-Brown**[*,1,2]    **Shaheer U. Saeed**[1,2]    **Ahmed Abdulaal**[1]    **Thomas Dowrick**[2]
**Yakup Kilic**[3]    **Sophie Wilkinson**[3]    **Jack Gao**[3]    **Meghavi Mashar**[3]
**Alkisti Stavropoulou**[1]    **Emma L. Thomson**[1]    **Zachary MC Baum**[1,2]    **Chloe He**[1,2]
**Simone Foti**[1,2]    **Brian Davidson**[3]    **Yipeng Hu**[1,2]    **Matthew J Clarkson**[1,2]

[1] Centre for Medical Image Computing, UCL, London, United Kingdom
[2] Welcome/EPSRC Centre for Interventional And Surgical Sciences, London, United Kingdom
[3] University College London Hospitals
[*] n.montanabrown@cs.ucl.ac.uk

## A    Data Location and Code

We provide an AWS S3 bucket hosting the data, which may be downloaded from the following links:

- `https://saramis.s3.eu-north-1.amazonaws.com/abdomen.tar.gz`

- `https://saramis.s3.eu-north-1.amazonaws.com/amos.tar.gz`

- `https://saramis.s3.eu-north-1.amazonaws.com/total.tar.gz`

- `https://saramis.s3.eu-north-1.amazonaws.com/metadata.tar.gz`

- `https://saramis.s3.eu-north-1.amazonaws.com/rl_expt.tar.gz`

The data includes the original `SARAMIS` dataset, as well as the data used to replicate the navigation experiments detailed in the paper. The code is made publicly available at the associated `SARAMIS` repository.

### A.1    Data Structure and Contents

Data is provided as a .tar.gz files. Within the S3 buckets are five subfolders - three constituent datasets, Abdomen-1k, AMOS and TotalSegmentator, the data used for the autonomous navigation (Sec 5) in the paper in a subfolder "rl_expt", and a metadata folder.

Within the Abdomen-1k, AMOS, and TotalSegmentator folders exist a number of sub-folders. Each sub-folder refers to an anonymised patient case, with matching names to the original CT datasets.

The following files are listed in the Abdomen-1k and AMOS subfolders:

1. .nii.gz: CT scan and label.

2. slicer_segs: subfolder containing original .nii.gz files outputted from the TotalSegmentator model applied to the .nii.gz CT files.

3. auto_seg_pre.seg.nrrd: original labelling node, converted from the slicer_segs to an individual .seg.nrrd file, prior to the editing phase.

4. auto_seg.seg.nrrd: edited labelling node post-editing phase.

---

Submitted to the 37th Conference on Neural Information Processing Systems (NeurIPS 2023) Track on Datasets and Benchmarks. Do not distribute.

5. analysis: subfolder containing the .npy arrays extracted from the auto_seg files for the analysis portion of the paper, pre- and post-editing, and with filtering, such that they could be converted into meshes.

6. full_meshes: subfolder containing processed meshes for every organ, derived from the filtered .npy arrays.

Within each full_meshes are a number of sub-folders referring to individual organs and their assets. Each sub-folder, for a given organ, contains a number of files:

1. organ.ply: original meshed organ from the filtered npy array post-correction, extracted using a binary marching cubes algorithm.

2. organ_laplace_smooth_mesh_decimation.ply: processed organ file, laplace smoothed and mesh decimated.

3. organ_laplace_smooth_mesh_decimation_centered_local.ply: processed organ, centered in the local frame of reference (such that the organ is centered to 0 in it's own frame of reference).

4. organ_laplace_smooth_mesh_decimation_centered_global.ply: processed organ, centered in the patient frame of reference (such that all the organs in the patient are positioned relative to each other and globally mean 0 centered). Additionally, s-t coordinates for texturing are added to the mesh via Blender processing.

5. organ_laplace_smooth_mesh_decimation_centered_global.vtk: tetrahedral mesh obtained from the global centered mesh.

6. bake_map_diffuse_1000.png: diffuse maps baked from Blender.

7. bake_map_normals_1000.png: normal maps baked from Blender.

The formatting of the TotalSegmentator sub-dataset is slightly different, as it was not reviewed by the annotation team (considering that the ground truth labels from the original dataset were reviewed by a clinician and match the set of labels generated for the AMOS and Abdomen-1k dataset). Therefore, each subfolder contains the following data:

1. ct.nii.gz: CT scan in .nii format.

2. segmentations: subfolder with .nii.gz files labelled according to the organ the label corresponds to.

3. analysis: folder containing .npy array for analysis of structures.

4. full_meshes: subfolder containing folders with meshing output from the ground truth segmentations. If the segmentation extracted for a given organ from the segmentations subfolder is empty, there will be no associated subfolder for that organ in the full_meshes subfolder.

Finally, some subfolders in the three datasets contain additional folders labelled colon, which contain the results of the procedural generation process as detailed in the paper. This folders contain files such as:

1. bake*.png: baked diffuse and normal maps.

2. Centerline curve*.csv and .json: files detailing the output of the manual segment picking using 3DSlicer as described in the paper.

3. interp_curve.txt: interpolated BSpline curve from the individual centerline segments.

4. indices_*.txt: points and indices (int) on curve or mesh corresponding to anatomy detailed in the paper.

We describe which patient cases contain procedurally generated colons in the metadata .txt files corresponding to each dataset.

 **A.2   Data Loading Instructions**

The `SARAMIS` dataset contains multi-modal data that can be interacted with in different manners. Here we provide further information on how to load all the image formats supplied by the dataset. Code to load the data is also provided throughout the `SARAMIS` code repository.

1. .nii.gz and .seg.nrrd files: the NIFTI file standard (.nii.gz) and NRRD file standard, respectively, are common medical imaging formats, and can be loaded using the 3DSlicer GUI (via drag and drop), or using the python packages `niibabel` and `pynrrd`.

2. .vtk: tetrahedral mesh format, which can be read using gmsh.

3. .ply: Polygon File Format may be loaded using a GUI such as MeshLab, 3DSlicer, and Blender. It can also be loaded using VTK data formats, PyTorch Geometric, PyTorch3D, amongst others.

4. .png, .npy, .csv: common image and data formats that can be loaded with the `numpy` Python package.

The metadata folder contains a number of files:

1. {dataset}_interp_colons.txt: detailing the folders for each dataset which were manually processed to extract the centerlines for procedural colons as detailed in the paper.

2. {dataset}_old.csv and {dataset}_new.csv: files containing the pixel values per organ used to perform the analysis reported in the paper.

3. exclude.txt: comma separated txt file with case folder and reason why it is excluded from the dataset.

# B   Appendix - Datasheet for Datasets

**Motivation**

**For what purpose was the dataset created?** Was there a specific task in mind? Was there a specific gap that needed to be filled? Please provide a description.

Laparoscopy and endoscopy are techniques in surgical and medical practice which involve inserting video cameras into a patient in order to diagnose and treat a number of conditions, and have made it possible to perform minimally invasive surgery (MIS). The benefits of MIS have been well documented [2, 18, 27], and can be summarised as follows: 1) Reduced post-operative pain, 2) Shortened hospital stays [27, 17], 3) Improved rates of patient recovery [6], and 4) Lowered costs to hospital systems in a number of interventions [2, 24, 17, 7]. Additionally, recent advances in robotics have enabled the pairing of robotic elements with laparoscopic equipment, which provides further benefits such as an improved ergonomic environment for surgeons [31] and the possibility of teleoperation [5]. In tandem, (partially) autonomous robotic surgery has emerged as an increasingly important research topic [4, 22, 9]. Indeed, many surgeons consider the full automation of robot-assisted minimally invasive surgery (RAMIS) as the 'end goal' of surgical practice [9].

Traditional computer vision applications have long exploited tracking devices and LIDAR-like sensors to create large-scale annotated datasets for relevant tasks such as camera-pose estimation or scene-reconstruction [8, 14]. However, these devices are logistically challenging to incorporate into surgical workflow, as they require sterilisation, consequently multiple calibrations, and are expensive to accrue. Overall, this has resulted in limited open-source datasets for computer vision tasks in MIS/RAMIS. In parallel, synthetic data and rendering environments have emerged as promising, alternative resources to enable computer vision at scale [20, 25], and are important for the development and testing of safe autonomous systems. However, *in silico* datasets for the development of deep learning algorithms and autonomous systems in MIS/RAMIS are limited in number and application [13].

The proposed dataset, "Simulation Assets for Robotic Assisted and Minimally Invasive Surgery" (SARAMIS), aims to provide the first large scale dataset of rendering assets for the tasks of MIS and RAMIS.

**Who created this dataset (e.g., which team, research group) and on behalf of which entity (e.g., company, institution, organization)?**

The dataset was created by researchers at the Centre for Medical Image Computing (CMIC), Wellcome/EPSRC Centre for Interventional And Surgical Sciences (WEISS), on behalf of University College London (UCL), London, United Kingdom.

**Who funded the creation of the dataset?** If there is an associated grant, please provide the name of the grantor and the grant name and number.

This work is supported by the Wellcome/EPSRC Centre for Interventional and Surgical Sciences [203145Z/16/Z]. NMB, AA, ET, AS, and SF are supported by the EPSRC-funded UCL Centre for Doctoral Training in Intelligent, Integrated Imaging in Healthcare (i4health) [EP/S021930/1]. AA is supported by an EPSRC Industrial Case grant [EP/W522077/1], and a Microsoft Research PhD Scholarship Wellcome Trust award [221915/Z/20/]. MJC, YH, NMB, and SUS are supported by EPSRC grant [EP/T029404/1]. TD is supported by EPSRC grant [EP/V052438/1]. ZMCB is supported by the Natural Sciences and Engineering Research Council of Canada Postgraduate Scholarships-Doctoral Program, and the University College London Overseas and Graduate Research Scholarships. This work is also supported by the International Alliance for Cancer Early Detection, an alliance between Cancer Research UK [C28070/A30912, C73666/A31378], Canary Center at Stanford University, the University of Cambridge, OHSU Knight Cancer Institute, University College London and the University of Manchester.

**Any other comments?**

---

## Composition

**What do the instances that comprise the dataset represent (e.g., documents, photos, people, countries)?** Are there multiple types of instances (e.g., movies, users, and ratings; people and interactions between them; nodes and edges)? Please provide a description.
Each instance in the dataset represents organs and anatomical features of the human body. Each instance is acquired from a singular human subject.

**How many instances are there in total (of each type, if appropriate)?**

There are a total of 2529 instances across the dataset.

**Does the dataset contain all possible instances or is it a sample (not necessarily random) of instances from a larger set?** If the dataset is a sample, then what is the larger set? Is the sample representative of the larger set (e.g., geographic coverage)? If so, please describe how this representativeness was validated/verified. If it is not representative of the larger set, please describe why not (e.g., to cover a more diverse range of instances, because instances were withheld or unavailable).

The dataset contains all possible instances.

**What data does each instance consist of? "Raw" data (e.g., unprocessed text or images) or features?** In either case, please provide a description.

Each instance consists of the following:

- Computed Tomography (CT) Scan: the original CT scan the data is derived from is included for reference and re-analysis. The CT scan is in the format of a .nii.gz file, a common

medical imaging data format. The CT scans were previously anonymised by the respective centres, and as such do not contain identifying information.

- Segmentation Map: two segmentation labels are provided describing anatomical features within the CT scan. Each segmentation map describes the voxel class of the CT scan from the following classes: the spleen, kidney right, kidney left, gallbladder, liver, stomach, aorta, inferior vena cava, portal vein and splenic vein, pancreas, adrenal gland right, adrenal gland left, lung upper lobe left, lung lower lobe left, lung upper lobe right, lung middle lobe right, lung lower lobe right, vertebrae L5, vertebrae L4, vertebrae L3, vertebrae L2, vertebrae L1, vertebrae T12, vertebrae T11, vertebrae T10, vertebrae T9, vertebrae T8, vertebrae T7, vertebrae T6, vertebrae T5, vertebrae T4, vertebrae T3, vertebrae T2, vertebrae T1, vertebrae C7, vertebrae C6, vertebrae C5, vertebrae C4, vertebrae C3, vertebrae C2, vertebrae C1, esophagus, trachea, heart myocardium, heart atrium left, heart ventricle left, heart atrium right, heart ventricle right, pulmonary artery, brain, iliac artery left (common iliac left artery), iliac artery right (common iliac right artery), iliac vena left (common iliac left vein), iliac vena right (common iliac right vein), small bowel, duodenum, colon, rib left 1, rib left 2, rib left 3, rib left 4, rib left 5, rib left 6, rib left 7, rib left 8, rib left 9, rib left 10, rib left 11, rib left 12, rib right 1, rib right 2, rib right 3, rib right 4, rib right 5, rib right 6, rib right 7, rib right 8, rib right 9, rib right 10, rib right 11, rib right 12, humerus left, humerus right, scapula left, scapula right, clavicula left, clavicula right, femur left, femur right, hip left, hip right, sacrum, face, gluteus maximus left, gluteus maximus right, gluteus medius left, gluteus medius right, gluteus minimus left, gluteus minimus right, autochthon left (erector spinae left), autochthon right (erector spinae right), iliopsoas left (psoas major left), iliopsoas right (psoas major right), and the urinary bladder. The segmentation files are provided as .seg.nrrd files, a common medical imaging data format. One segmentation label corresponds to the data pre-review, and the other label corresponds to the data post-review by a team of 7 trained clinical annotators and 4 radiologists. From the above list, several structures were corrected in name compared to the original model's output (found in brackets next to the original label).

- Ground Truth Label: the original segmentation label from the original segmentation datasets is also provided for reference and further analysis.

- 3D mesh models (.ply): each label in the post-processed segmentation map is converted into a surface model representation in standard .ply format.

- Tetrahedral volumes (.vtk): each surface model representation is converted into a tetrahedral volume for collision simulations.

- Normal maps (.png): each surface model has baked a normal/bump map simulating geometric textures of the organ.

- Diffuse maps (.png): each surface model has a baked diffuse map simulating color of the organ.

**Is there a label or target associated with each instance?** If so, please provide a description.

We include the original ground truth labels from the parent datasets for reference. These labels were generated by clinicians, and are voxel-wise segmentations of different anatomical structures within the scan.

- Abdomen-1k: (label=1), kidney (label=2), spleen (label=3), and pancreas (label=4).

- AMOS: (label=1) spleen, (label=2) right kidney, (label=3) left kidney, (label=4) gallbladder, (label=5) esophagus, (label=6) liver, (label=7) stomach, (label=8) aorta, (label=9) postcava, (label=10) pancreas, (label=11) right adrenal gland, (label=12) left adrenal gland, (label=13) duodenum, (label=14) bladder, (label=16) prostate/uterus

- TotalSegmentator: the segmentations match the voxel classes of the proposed dataset (see above).

**Is any information missing from individual instances?** If so, please provide a description, explaining why this information is missing (e.g., because it was unavailable). This does not include intentionally removed information, but might include, e.g., redacted text.

The dataset was derived from a variety of CT scans. These were classified as belonging to one of the following set: {full-body (FBCT), chest-abdomen-pelvis (CTCAP), abdomen-pelvis (CTAP), abdominal (ACT)} (see Table 1). Given that each CT scan images different parts of the human anatomy, the presence of each label in the segmentation map will vary. For example, the cervical vertebrae (vertebrae C*) or the brain will not be imaged in an ACT. Therefore, different instances will contain different sets of derived assets in the form of .ply, .tet and .png files.

The split of datasets is summarised in Table. 1.

Table 1: Summary of CT data of three datasets from which SARAMIS is derived. FBCT = Full Body CT, CTCAP = chest-abdomen-pelvis CT, CTAP = abdomen-pelvis CT, ACT = Abdomen CT. Other refers to a alternative CT scans, as described in the datasheet for [30].

| Dataset | Initial | Type of CT Scan | | | | | Excluded | No changes |
|---|---|---|---|---|---|---|---|---|
| | | FBCT | CTCAP | CTAP | ACT | Other | | |
| Abdomen-1k | 1063 | 10 | 366 | 71 | 592 | 0 | 15 | 526 |
| Amos | 600 | 0 | 72 | 220 | 0 | 0 | 321 | 140 |
| TotalSegmentator | 1200 | 169 | 197 | 110 | 0 | 724 | 0 | 1200 |
| SARAMIS | 2863 | 179 | 635 | 401 | 592 | 724 | 336 | 1866 |

**Are relationships between individual instances made explicit (e.g., users' movie ratings, social network links)?** If so, please describe how these relationships are made explicit.

Yes. We maintain the original population splits as defined by their parent datasets.

**Are there recommended data splits (e.g., training, development/validation, testing)?** If so, please provide a description of these splits, explaining the rationale behind them.

No.

**Are there any errors, sources of noise, or redundancies in the dataset?** If so, please provide a description.

Elements of the dataset were generated procedurally:

1. Baked diffuse reflectance maps and normal maps: using Blender's CYCLES ray-tracing engine, the properties of the shader nodes were baked into 2D images for ease of rendering in other platforms. The ray-tracing platform involves probabilistic sampling.

**Is the dataset self-contained, or does it link to or otherwise rely on external resources (e.g., websites, tweets, other datasets)?** If it links to or relies on external resources, a) are there guarantees that they will exist, and remain constant, over time; b) are there official archival versions of the complete dataset (i.e., including the external resources as they existed at the time the dataset was created); c) are there any restrictions (e.g., licenses, fees) associated with any of the external resources that might apply to a future user? Please provide descriptions of all external resources and any restrictions associated with them, as well as links or other access points, as appropriate.

The dataset was derived from the AMOS [12], Abdomen-1k [15] and TotalSegmentator [30] datasets. The TotalSegmentator dataset is available on Zenodo, the AMOS dataset is available on Zenodo, and the Abdomen-1k dataset is available on Zenodo.

**Does the dataset contain data that might be considered confidential (e.g., data that is protected by legal privilege or by doctor-patient confidentiality, data that includes the content of individuals non-public communications)?** If so, please provide a description.

No.

**Does the dataset contain data that, if viewed directly, might be offensive, insulting, threatening, or might otherwise cause anxiety?** If so, please describe why.

No.

**Does the dataset relate to people?** If not, you may skip the remaining questions in this section.

Yes.

**Does the dataset identify any subpopulations (e.g., by age, gender)?** If so, please describe how these subpopulations are identified and provide a description of their respective distributions within the dataset.

No.

**Is it possible to identify individuals (i.e., one or more natural persons), either directly or indirectly (i.e., in combination with other data) from the dataset?** If so, please describe how.

No.

**Does the dataset contain data that might be considered sensitive in any way (e.g., data that reveals racial or ethnic origins, sexual orientations, religious beliefs, political opinions or union memberships, or locations; financial or health data; biometric or genetic data; forms of government identification, such as social security numbers; criminal history)?** If so, please provide a description.

No.

**Any other comments?**

---

**Collection Process**

---

**How was the data associated with each instance acquired?** Was the data directly observable (e.g., raw text, movie ratings), reported by subjects (e.g., survey responses), or indirectly inferred/derived from other data (e.g., part-of-speech tags, model-based guesses for age or language)? If data was reported by subjects or indirectly inferred/derived from other data, was the data validated/verified? If so, please describe how.

1. The initial CT data was collected by compounding existing datasets of CT scans: Abdomen1k, AMOS and TotalSegmentator.

2. The data was preliminarily annotated using an open-source deep learning segmentation model [30] trained to predict 104 anatomical classes in CT scans. The open source model is available here. Given that the TotalSegmentator dataset contains the same labels, and was inspected by a clinical team, we exclude it from the revision process.

3. All the preliminary annotations derived from the AMOS and Abdomen-1k dataset were inspected by a team of 7 trained annotators and 4 radiologists.

4. Initially, all the preliminary annotations were inspected by trained annotators under the following protocol:

(a) Annotators were recruited from the host centre, and consist of 7 junior researchers in medical imaging, with at least 4 years of medical imaging expertise.

(b) Annotators were instructed to visually inspect the veracity of the preliminary annotations by inspecting the 3D reconstructions of the preliminary annotations in 3DSlicer. Additionally, they were instructed to review the overlay of the annotations on the original CT scan slice by slice.

(c) Annotators were instructed to: 1) Verify class homogeneity within an anatomical structure, 2) Flag topological errors (e.g., slices missing, holes within an anatomical structure), 3) Flag under- or over-segmentation, and 4) Flag potential pathology for each of the scans to be reviewed Annotators were requested to log the most superior and inferior vertebral body visible in the scan, as well as the type of CT scan from the set {full-body (FBCT), chest-abdomen-pelvis (CTCAP), abdomen-pelvis (CTAP), abdominal (ACT)}.

(d) Annotators then received a 2h training session on the how to use the annotation software (3DSlicer), as well as jointly carrying out a reviewing task with the guidance of a clinician.

(e) Annotators carried out the reviewing task under the supervision of a clinician, which could be consulted in cases where the individual annotator could not resolve the presence or not of an error.

(f) Annotators were requested to fill in a spreadsheet with any errors as described above.

5. Subsequent to the initial review phase, cases that were flagged were individually re-inspected. Under the supervision of a clinician, the segmentation errors were manually corrected.

6. Post-review and correction, 450 scans were allocated to radiologists for review of segmentation and correction quality. Review was carried out under the following protocol:

(a) 4 radiologists were recruited from the host centre partner hospitals.

(b) Radiologists were instructed to visually inspect the veracity of the corrected annotations, and note any significant errors (in the form of gross mistakes versus small pixel-wise deviations in segmentation veracity), as well as any pathology arising from the scan.

(c) Radiologists received a brief training on using the segmentation platform 3DSlicer, and were requested to fill in a spreadsheet with errors noted in the scans.

7. Once the review phase was concluded, the data was post-processed to obtain, firstly, the 3D meshes this consisted of the following steps:

(a) Label cleanup: removal of noise in the verified segmentations, consisting of salt-and-pepper removal.

(b) Meshing: following the label cleanup, the 3D volumes were converted into .ply files using the marching cubes algorithm (vtk.vtkMarchingCubes()).

(c) Mesh decimation and smoothing: given the voxel resolution of the original CT scans could vary between 0.5-5+mm in each direction, the meshes are smoothed using Laplacian smoothing to better represent smooth surfaces. Additionally, a mesh decimation is performed; specifically, we perform a quadric edge collapse using an implementation from MeshLab.

(d) Tetrahedral volume generation: the algorithm detailed in [10] through an open-source implementation. .msh files are converted into .vtk files using gmsh.

8. The 3D meshes were then processed using Blender to obtain normal maps (to texture the surfaces) and diffuse maps (to add colour to the surfaces). The normal maps and diffuse maps were generated procedurally.

(a) We design procedural textures and Principled Bi-directional Scattering Distribution Functions (BSDFs) for a number of anatomy groups using Blender's shading node by referencing open-source datasets of intra-operative images [3, 29, 1], surgical journal papers [11, 28, 16], and open-source tutorials [21]. Final procedural materials were inspected and verified by a clinician.

- Bones: all vertebrae, all ribs, sacrum, all humerus, all tibia, all hips, all femur
- Lungs: Lung segments, trachea
- Stomach: stomach, urinary bladder
- Pancreas: pancreas, adrenal gland
- Bowels: duodenum, colon, small bowel, oesophagus
- Gallbladder: gallbladder
- Liver: liver, kidneys
- Spleen: spleen
- Vascular: all veins and arteries
- Muscle: gluteus, autochthons, iliopsoas, all heart segments

(b) A Shader node in Blender was created for each reference texture and diffuse map.

(c) The associated organ meshes were procedurally unwrapped, and the textures and diffuse maps were baked using GPU Cycles in Blender (cycles=1).

(d) Full shader nodes are provided open-source for the procedural simulation of textures, or modification of parameters. We refer the reader to the implementation at the associated `SARAMIS` repository.

**What mechanisms or procedures were used to collect the data (e.g., hardware apparatus or sensor, manual human curation, software program, software API)?** How were these mechanisms or procedures validated?

The data was collected through manual human curation of an open source dataset. The annotation was performed through the use of an Apple IPad (8th Gen) with an Apple Pencil (1st Gen) with an instance of 3DSlicer (5.2.2) mirrored onto the IPad. The meshing was performed using open-source tools, such as meshio, VTK, and MeshLab, and using Blender. All post-processing was performed on a desktop with an Intel Core i9 24-Core Processor i9-13900KF (3.0GHz) 36MB Cache, 64GB of RAM, and an NVIDIA 3090Ti 24GB GPU. The procedural texturing and creation of diffuse maps was performed through the use of shader nodes in Blender. The full software stack is released open-source with the dataset and associated paper.

**If the dataset is a sample from a larger set, what was the sampling strategy (e.g., deterministic, probabilistic with specific sampling probabilities)?**

Several of the original scans that were used to extract the data-points were excluded. From the initial 2863 scans, a total of 336 were excluded from segmentation analysis for the following reasons: 194 due to lack of availability of test set label, 15 due to significant pathology making organ differentiation difficult, 13 due to the presence of fluid in the abdomen (e.g. haemoperitoneum or ascites) occluding organs of interest, 100 due to alternative imaging modality (MRI), 2 due to metallic artefacts in the scan, 1 due to a poor quality scan, and 1 due to original file corruption leading to lack of a segmentation file. Overall, this results in 1048, 279, 1200 scans from the Abdomen-1k, AMOS, and TotalSegmentator datasets, respectively. We detail the excluded data in the metadata folder excluded.txt file.

**Who was involved in the data collection process (e.g., students, crowdworkers, contractors) and how were they compensated (e.g., how much were crowdworkers paid)?**

7 trained annotators (junior medical image researchers with 4+ years of experience in medical imaging) and 4 radiologists (specialty training levels 1-4, NHS England).

**Over what timeframe was the data collected? Does this timeframe match the creation timeframe of the data associated with the instances (e.g., recent crawl of old news articles)?** If not, please describe the timeframe in which the data associated with the instances was created.

The `SARAMIS` was annotated and processed between Jan-Jun 2023. The original CT scans were published in 2021 (Abdomen-1k, collected between 2019 and 2021), 2022 (AMOS and TotalSegmentator).

**Were any ethical review processes conducted (e.g., by an institutional review board)?** If so, please provide a description of these review processes, including the outcomes, as well as a link or other access point to any supporting documentation.

No.

**Does the dataset relate to people?** If not, you may skip the remaining questions in this section.

Yes.

**Did you collect the data from the individuals in question directly, or obtain it via third parties or other sources (e.g., websites)?**

The data was collected from open-source medical imaging datasets.

**Were the individuals in question notified about the data collection?** If so, please describe (or show with screenshots or other information) how notice was provided, and provide a link or other access point to, or otherwise reproduce, the exact language of the notification itself.

The original datasets which were post-processed are provided under either CC-BY-4.0 or a CC-BY-NC-SA licenses, which allows for the redistribution of the material in any medium or format, as well as adaptation of the material for any purpose for non-commercial purposes under a similar license. The original individuals would have consented to such a license, and thus not notified of further amendments.

**Did the individuals in question consent to the collection and use of their data?** If so, please describe (or show with screenshots or other information) how consent was requested and provided, and provide a link or other access point to, or otherwise reproduce, the exact language to which the individuals consented.

See above.

**If consent was obtained, were the consenting individuals provided with a mechanism to revoke their consent in the future or for certain uses?** If so, please provide a description, as well as a link or other access point to the mechanism (if appropriate).

No further consent beyond that of the original datasets was obtained.

**Has an analysis of the potential impact of the dataset and its use on data subjects (e.g., a data protection impact analysis) been conducted?** If so, please provide a description of this analysis, including the outcomes, as well as a link or other access point to any supporting documentation.

No.

**Any other comments?**

| Preprocessing/cleaning/labeling |
| --- |

**Was any preprocessing/cleaning/labeling of the data done (e.g., discretization or bucketing, tokenization, part-of-speech tagging, SIFT feature extraction, removal of**

**instances, processing of missing values)?** If so, please provide a description. If not, you may skip the remainder of the questions in this section.

- The automatic segmentations were manually corrected under the supervision of a clinician, and consisted in adding and removing pixels to adjust the segmentations as needed.
- The corrected segmentations were filtered using binary morphological closing operation (cross kernel, size=1). Additionally, the intra-patient segmentations were verified against each other to ensure they did not intersect (as this is not anatomically plausible). Where intersection was found, the intersection of both classes were set to 0.
- The extracted surface representations were smoothed using Laplacian smoothing.
- The smoothed surfaces were decimated using mesh decimation.

**Was the "raw" data saved in addition to the preprocessed/cleaned/labeled data (e.g., to support unanticipated future uses)?** If so, please provide a link or other access point to the "raw" data.

Yes - the original CT data, as well as the pre-corrected segmentations and post-corrected segmentations are saved and provided.

**Is the software used to preprocess/clean/label the instances available?** If so, please provide a link or other access point.

Yes - see the associated `SARAMIS` repository.

**Any other comments?**

| Uses |
| --- |

**Has the dataset been used for any tasks already?** If so, please provide a description.

Beyond the usage in the paper associated to the dataset, the data has not been used for other tasks.

**Is there a repository that links to any or all papers or systems that use the dataset?** If so, please provide a link or other access point.

N/A

**What (other) tasks could the dataset be used for?**

The uses for this dataset are multiple.

- Synthetic data generation: the 3D models can be paired with a rendering environment to obtain 2D RGB images, 2D depth maps, 2D segmentation maps, and 2D optical flow images.
- Deformation simulation: the tetrahedral volumes provided can be used for the simulation of deformation of organs in a surgical setting.
- Generative 3D models: The 3D models could be used to create a 3D generative model of given organs.
- Learning textures in surgery: the 3D models could be paired with real intra-operative video (2D RGB images) to learn how to texture different organs in the human body.
- Camera-pose estimation: pose labels may be generated from a rendering environment, paired with a 2D image, to learn how to perform camera pose-estimation on different organs in surgery.

- Navigation: Like the exemplified case in the paper for this dataset, different organs could be used to design surgical scenes or scenarios, to teach reinforcement learning algorithms how to navigate to different targets, how to perform certain actions, or how to interact with the shapes in the environment.

**Is there anything about the composition of the dataset or the way it was collected and preprocessed/cleaned/labeled that might impact future uses?** For example, is there anything that a future user might need to know to avoid uses that could result in unfair treatment of individuals or groups (e.g., stereotyping, quality of service issues) or other undesirable harms (e.g., financial harms, legal risks) If so, please provide a description. Is there anything a future user could do to mitigate these undesirable harms?

No.

**Are there tasks for which the dataset should not be used?** If so, please provide a description.

Given that this dataset could be used to train autonomous agents for medical purposes, we would recommend careful validation of any autonomous systems prior to translational research.

**Any other comments?**

---

**Distribution**

---

**Will the dataset be distributed to third parties outside of the entity (e.g., company, institution, organization) on behalf of which the dataset was created?** If so, please provide a description.

Yes. The dataset will be provided by a CC BY-NC-SA to the wider public.

**How will the dataset will be distributed (e.g., tarball on website, API, GitHub)** Does the dataset have a digital object identifier (DOI)?

The full dataset will be released to the public further to review at a minted DOI within the UCL Research Data Repository.

**When will the dataset be distributed?**

The dataset is made publically available at the SARAMIS repository, with source code and links to download the data: https://github.com/NMontanaBrown/saramis.

**Will the dataset be distributed under a copyright or other intellectual property (IP) license, and/or under applicable terms of use (ToU)?** If so, please describe this license and/or ToU, and provide a link or other access point to, or otherwise reproduce, any relevant licensing terms or ToU, as well as any fees associated with these restrictions.

The dataset is provided under a CC BY-NC-SA license. The dataset may be shared, re-used and re-mixed for any purpose, subject to the condition that the original dataset is credited. The dataset is provided "as-is" and "as-available", and makes no representations or warranties of any kind concerning the dataset, whether express, implied, statutory, or other. This includes, without limitation, warranties of title, merchantability, fitness for a particular purpose, non-infringement, absence of latent or other defects, accuracy, or the presence or absence of errors, whether or not known or discoverable. The dataset cannot be used for commercial purposes. The dataset or any adaptations and derivations must be licensed under a similar license.

**Have any third parties imposed IP-based or other restrictions on the data associated with the instances?** If so, please describe these restrictions, and provide a link or other

access point to, or otherwise reproduce, any relevant licensing terms, as well as any fees associated with these restrictions.

The original datasets used to generate the `SARAMIS` dataset were provided by:

1. Abdomen1k: CC BY 4.0 license

2. AMOS: CC BY NC SA license

3. TotalSegmentator: CC BY 4.0 license.

As we derive from the AMOS dataset, we license the dataset entirely on a CC BY NC SA license.

**Do any export controls or other regulatory restrictions apply to the dataset or to individual instances?** If so, please describe these restrictions, and provide a link or other access point to, or otherwise reproduce, any supporting documentation.

No.

**Any other comments?**

| Maintenance |
| --- |

**Who will be supporting/hosting/maintaining the dataset?**

The dataset will be hosted on UCL's Research Data Repository, and it's supporting repository at the associated `SARAMIS` repository. These will be maintained in part, but not limited to, Nina Montana-Brown and Matt Clarkson (first author, and principal investigator of the work, correspondingly), both at University College London, United Kingdom at the time of publication.

**How can the owner/curator/manager of the dataset be contacted (e.g., email address)?**

The curator can be contacted at: nina.brown.15@ucl.ac.uk, or alternatively m.clarkson@ucl.ac.uk.

**Is there an erratum?** If so, please provide a link or other access point.

Errata will be modified in this section.

Errata: N/A

**Will the dataset be updated (e.g., to correct labeling errors, add new instances, delete instances)?** If so, please describe how often, by whom, and how updates will be communicated to users (e.g., mailing list, GitHub)?

Yes. Where errors are encountered, data is deleted, or more data is included into the dataset, the versioned data will be uploaded to the UCL Research Data Repository, with links to the original data. Issues may be raised on the original `SARAMIS` repository, and errata will be appended to the arXiv version of the paper as well as the datasheet associated to the dataset.

**If the dataset relates to people, are there applicable limits on the retention of the data associated with the instances (e.g., were individuals in question told that their data would be retained for a fixed period of time and then deleted)?** If so, please describe these limits and explain how they will be enforced.

No.

**Will older versions of the dataset continue to be supported/hosted/maintained?** If so, please describe how. If not, please describe how its obsolescence will be communicated to users.

Yes. The data will remain hosted on the UCL Research Data Repository.

**If others want to extend/augment/build on/contribute to the dataset, is there a mechanism for them to do so?** If so, please provide a description. Will these contributions be validated/verified? If so, please describe how. If not, why not? Is there a process for communicating/distributing these contributions to other users? If so, please provide a description.

The dataset is originally released under a CC-BY-NC-SA license, so authors may extend, augment, or build on SARAMIS for non commercial purposes provided the data is shared under the same/similar license.

If external parties wish to contribute directly to the dataset, we invite them to raise an issue on the SARAMIS dataset repository (`https://github.com/NMontanaBrown/saramis`) with their proposed contribution, steps to replicate, as well as a link to the contribution for review by the archivists of the dataset. The data will be manually reviewed by archivists of the dataset, and may involve third-parties associated to the archivists for speed of review. This will ensure contributions are open-source and open to the rest of the public.

**Any other comments?**

## C Procedural Generation of Colon Anatomy

In this section we describe the details for the procedural generation of colon anatomy for the `SARAMIS` dataset.

### C.1 Matching Algorithm for Manually Extracted Colon Centerlines

Firstly, we describe the matching algorithm relating to Section 3.1 "Mesh Generation" of the paper in Algo. 1.

---

**Data:** N unordered, line segments $L = \{l_1, l_2, ..., l_n\}$, each $l_i$ a set of ordered points,
$\quad l_i = \{p_1, p_2, ..., p_m\}$ $p_i \in \mathbb{R}^3$ of variable length, and start coordinate $P \in \mathbb{R}^3$.
**Result:** N ordered line segments $L_o$

start $\leftarrow$ P ;
$L_o \leftarrow$ [] ;
**while** *len (L)* **do**
$\quad$ startPoints = [$l$[0] for $l$ in $L$];
$\quad$ endPoints = [$l$[-1] for $l$ in $L$];
$\quad$ closestStart = min(EuclideanDistance(start, startPoints)) ;
$\quad$ closestEnd = min(EuclideanDistance(start, endPoints)) ;
$\quad$ **if** $closestStart > closestEnd$ **then**
$\quad\quad$ $l_{next} \leftarrow$ L[endPoints.index(next)] $\qquad\qquad$ ▷ Corresponding segment match
$\quad\quad$ $l_{next} \leftarrow$ reverse($l_{next}$)
$\quad$ **end**
$\quad$ **else**
$\quad\quad$ $l_{next} \leftarrow$ L[startPoints.index(next)]
$\quad$ **end**
$\quad$ $L_o$.insert($l_{next}$) ;
$\quad$ start $\leftarrow l_{next}[-1]$ ;
$\quad$ L.pop($l_{next}$);
**end**
**Algorithm 1:** Pseudocode to order sets of line segments to create a discontinous, ordered line segment

 ## C.2 Procedural Generation of Colon Meshes

Having obtained the ordered line segments in $L_o$, the curve is filtered for duplicate points, then filtered for points where the difference between subsequent points are larger than 2 times the median of points filtered using a 1D Gaussian (SD=5). Filtered points were subsequently used to fit a BSpline, and resampled to 1000 points.

The resampled BSpline curve can be used as the centerline to extrude a closed mesh using Blender. Three different functions - implemented via Blender Geometry Nodes - are provided to vary the curve radius parameters in the generated mesh in order to replicate anatomical features in real colon's: colonic folds and Haustra. The radius $r \in \mathbb{R}$ at each point $p \in \mathbb{R}^3$ on the centerline is generally parametrised by a function $r(p, \cdot)$. We implement three functions as follows, a jitter radius function, $r_{jitter}(p, \cdot)$, parametrised by the values scale, detail, roughness, distortion, and a multiplier $\mathcal{J} = (s, d, r, m, l) \in \mathbb{R}$ respectively, such that:

$$r_{jitter}(p, \mathcal{J}) = l \cdot f_{perlin}(s, d, r, m, \bar{p}) \tag{1}$$

where $f_{perlin}(.)$ evaluates the Perlin noise at the point $\bar{p}$ with given parameters in $\mathcal{J}$. This function was created to replicate smaller, internal colonic folds in the colon. The effects of different parameters are illustrated in Fig. 2. We further define a function $r_{sin}(p, \mathcal{S})$, with arguments $\mathcal{S} = (b, k, h) \in \mathbb{R}$, representing base radius $b$, amplitude $k$ and frequency $h$, such that:

$$r_{sin}(p, \mathcal{S}) = b \cdot (1 + k \cdot (0.5 \cdot sin(\bar{p} \cdot h) + 0.5)) \tag{2}$$

This function aims to replicate the Haustra in the colon, which vary periodically along the length of the colon. The effects of different parameters are illustrated in Fig. 1.

Finally, we combine $r_{jitter}$ and $r_{sin}$ to in the function $r_{sin,jitter}$:

$$r_{sin,jitter}(p, \mathcal{S}, \mathcal{J}) = r_{sin}(p, \mathcal{S}) + r_{jitter}(p, \mathcal{J}) \tag{3}$$

to combine both effects into one mesh. These are illustrated in Fig. 3.

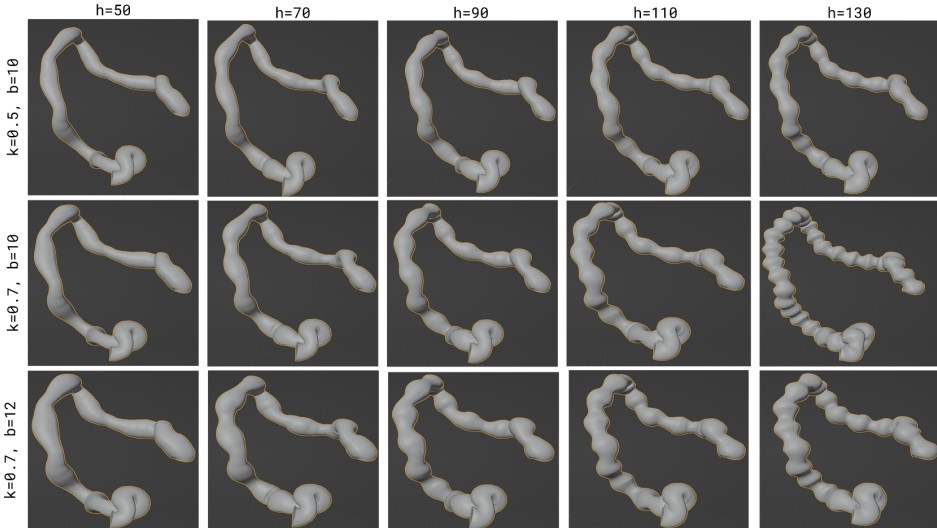

Figure 1: Example renders of procedurally generated colons with different parameters for sinusoidal radius defined in Eqn. 2. Along the columns, the frequency parameter varies between 50-130Hz, and along the rows, we showcase different combinations of amplitude and base radius parameters. In these renders, the curve was re-sampled to 500 points.

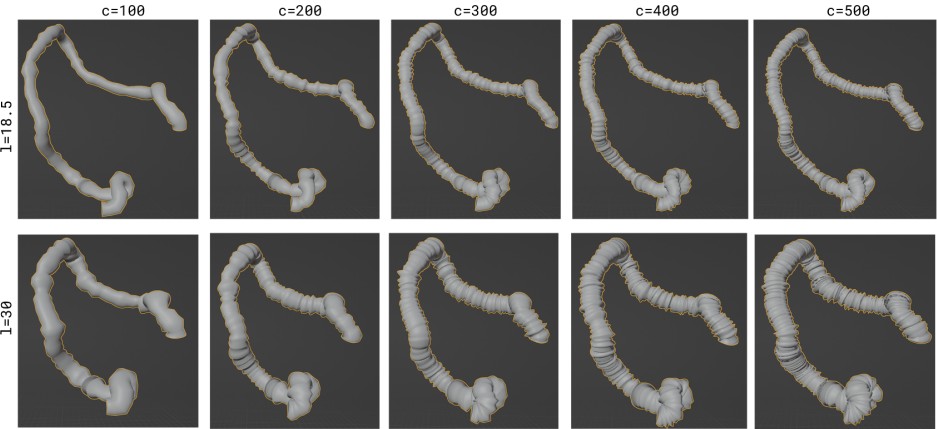

Figure 2: Example renders of procedurally generated colons with different parameters for the jitter radius defined in Eqn. 1. Along the columns, the number of points on the curve is resampled between 100-500 points, and along the rows, we vary the multiplier $l$ of the radius. We fix the other parameters at: scale=6.2, detail=15.5, roughness=0, distortion=12.1

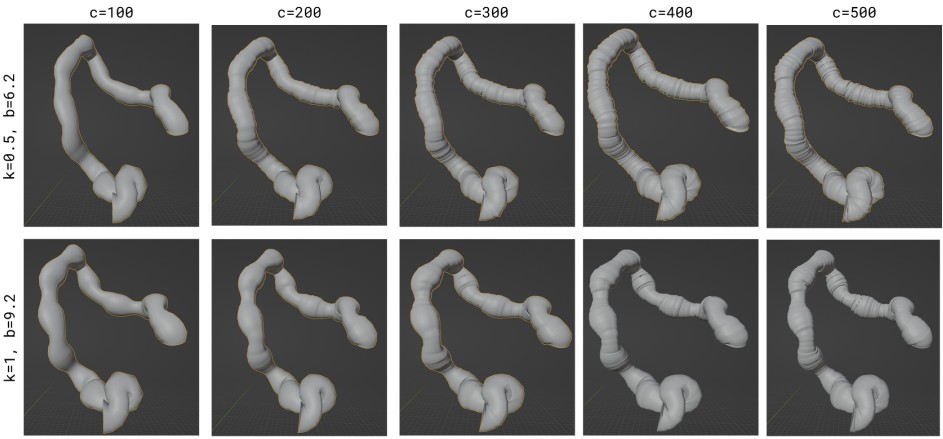

Figure 3: Example renders of procedurally generated colons with different parameters for the combined radius defined in Eqn. 3. Along the columns, the number of points on the curve is resampled between 100-500 points. Along the rows, we showcase different combinations of multiplication values (k) and base radius parameters (b). We fix the other parameters at: scale=6.2, detail=15.5, roughness=0, distortion=12.1

Blender files for the procedural simulation of meshes defined by input centerlines are provided open-source in the associated `SARAMIS` repository.

### C.3 Parameters Describing Autonomous Navigation Dataset

For the experiments performed with `SARAMIS` centerlines, colon meshes were extruded with $r_{jitter}$ in Eqn. 1, $\mathcal{J} = (scale = 6.2, detail = 15.5, smoothness = 0, distortion = 12.1, l = 12.0)$ with curves resampled to 1000 points.

# D   Procedural Texturing of Meshes

We design procedural textures and principled bi-directional scattering distribution functions (BSDFs) for a number of anatomy groups using Blender's shading node implementation.

## D.1   Introduction to Blender Shader Nodes

We illustrate a few examples of how procedural texturing nodes using Blender can generate different textures. Consider the function $f_{perlin}(., v) : \mathbb{R}^n \to \mathbb{R}$, which evaluates the fractal Perlin noise with parameters $\mathcal{P} = (s, d, r, m) \in \mathbb{R}$, representing scale, detail, roughness, and distortion, respectively for a given coordinate N-D coordinate, $v \in \mathbb{R}^n$, where $ns \in \{1, 2, 3, 4\}$. A simple Blender graph can be constructed such that each texture coordinate $t \in \mathbb{R}^2$ of a given object can be mapped to certain parameters of the principal BSDF. The construction of the above Blender graph, and resulting renders for different Perlin noise parameters used to modify the base color of the material output is illustrated in Fig.4

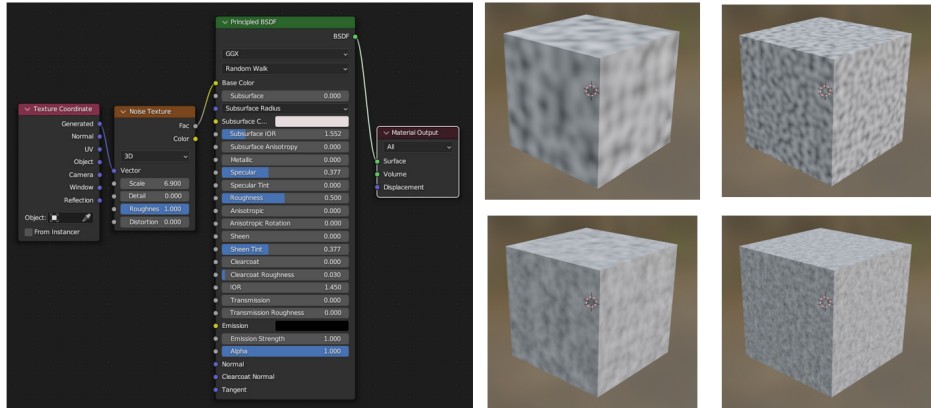

Figure 4: Example renders of procedurally generated textures using Perlin noise to calculate an object's base color. Row cubes are generated with d = [0, 1.5], columns are generated with scale = [6.9, 17]. Roughness and distortion are kept at 1, 0 respectively.

We can expand the above set of shader nodes by adding a color ramp node that modifies the output $p = f_{perlin}(\mathcal{P}, v)$ linearly with the following equation values $c_{max}, c_{min} \in \mathbb{R}$, and clamping it around the equation values:

$$y_{lin} = \frac{p}{c_{max} - c_{min}} + \frac{c_{min}}{c_{min} - c_{max}} \tag{4}$$

$$y_{\mathrm{ramp}}(x) = \begin{cases} 0 & \text{if } y(x) < 0 \\ y(x) & \text{if } 0 \leq y_{lin}(p) \leq 1 \\ 1 & \text{if } y(x) > 1 \end{cases} \tag{5}$$

We demonstrate the use of color-ramp to modify the base color of a cube in Fig. 5.

Additionally, the same original set of nodes may be used to modify other properties in the principal BSDF. In Fig. 6 we showcase using the metallic, roughness and clearcoat properties of the BSDF to modify the clamped Perlin texture on the render. These do not directly affect the base color of the object (which is set to black), but rather the way that light interacts with the material.

The same functions can be used to generate different normal mappings on the texture coordinates, therefore modifying the texture appearance of the object without modifying the underlying geometry. In Fig. 7, we showcase how to use a Blender displacement node in order to modify the material displacement procedurally.

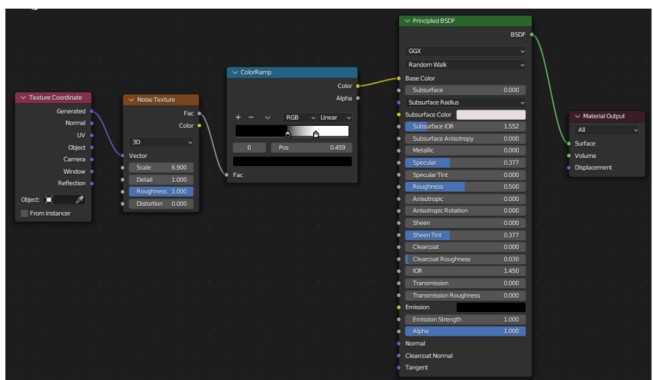
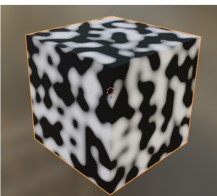
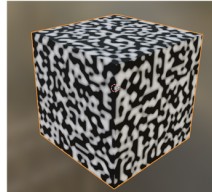

Figure 5: Example renders of procedurally generated textures using Perlin noise and a color ramp node to calculate an object's base color. Cubes are generated scale = [6.9, 17]. Roughness, distortion and detail are kept at 1, 0, and 0, respectively.

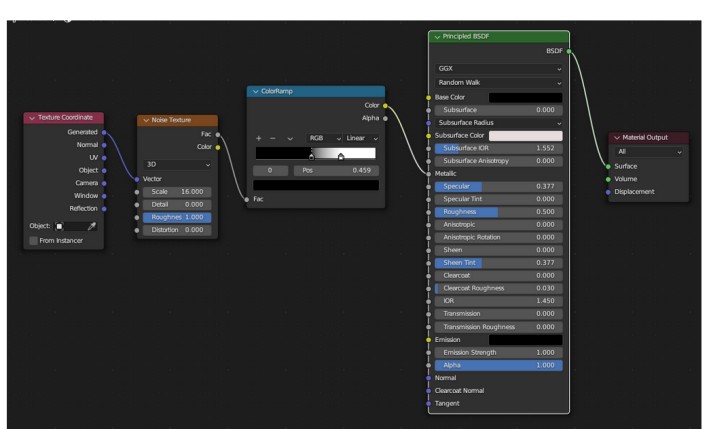
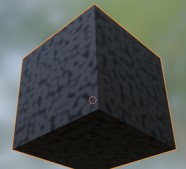
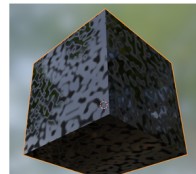
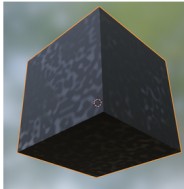

Figure 6: Example renders of procedurally generated textures using Perlin noise and a color ramp node to calculate an object's different properties, with the same base color. Top, middle, and bottom renders are generated using the metallic, roughness, and clearcoat parameters, respectively. Cubes are generated scale = [6.9, 17]. Roughness, distortion and detail are kept at 1, 0, and 0, respectively.

Furthermore, combinations of functions may be used in order to generate more complex textures. For example, we may generate a novel texture with two different $\mathcal{P}$, and multiplying their output. Let $\mathcal{P}_1 = [0.6, 1.8, 1.0, 2.6]$ and $\mathcal{P}_2 = [2.8, 1.8, 1.0, 2.2]$. The material normal displacement output is defined by the function $t_{displacement}(., v) : \mathbb{R}^2 \to \mathbb{R}$:

$$t_{displacement}(\mathcal{P}_1, \mathcal{P}_2, v) = f_{perlin}(\mathcal{P}_1, v) \times f_{perlin}(\mathcal{P}_2, v) \tag{6}$$

and is described in Fig.8.

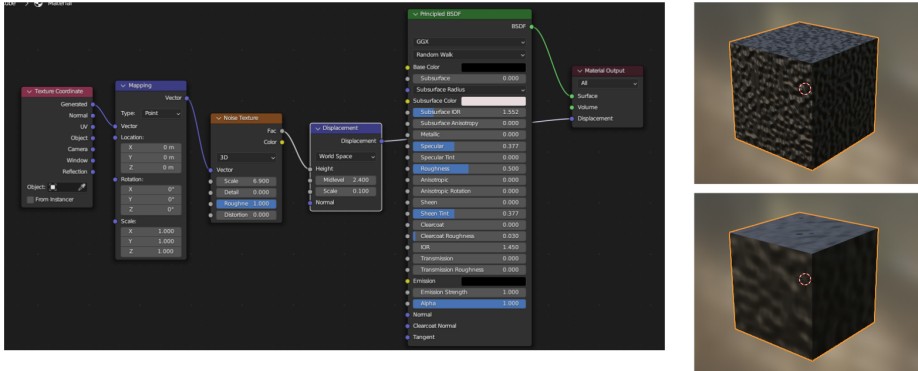

Figure 7: Example renders of procedurally generated textures using Perlin noise and a displacement node, with the same base color. Cubes are generated scale = [6.9, 17]. Roughness, distortion and detail are kept at 1, 0, and 0, respectively, as well as displacement parameters [midlevel=2.4, displacement=0.2].

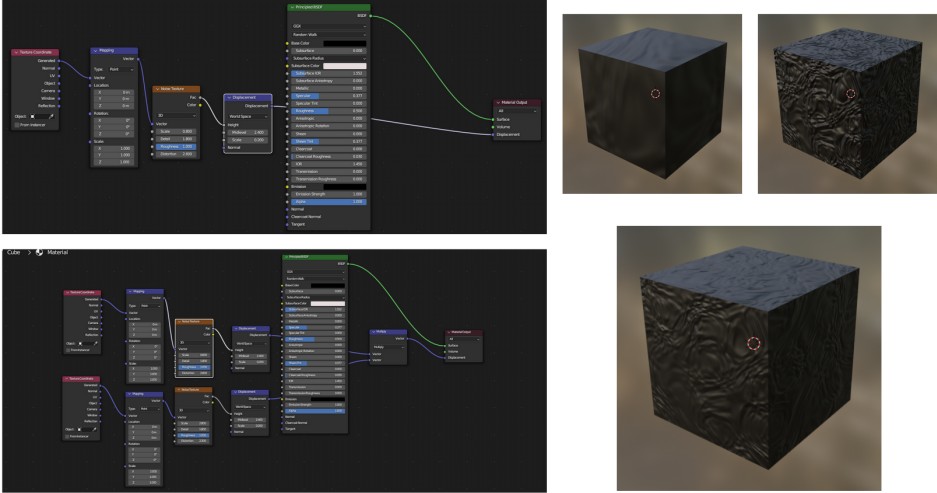

Figure 8: Example renders of procedurally generated textures using combination of Perlin noise, displacement node and vector math nodes, with the same base color. The textures are generated with $\mathcal{P}_1 = [0.6, 1.8, 1.0, 2.6]$ and $\mathcal{P}_2 = [2.8, 1.8, 1.0, 2.2]$, respectively. We showcase the individual Blender shader nodes on the left, with the top panel representing the individual configuration, and the combined configuration on the bottom. The three renders represent the different parametrisations, with the bottom representing the combined parametrisation.

## D.2 Generation of SARAMIS Textures

We use the basic principles described in Sec. D.1 to iteratively create textures to describe the appearance of different organs and organ groups in the human body. To replicate the materials, we reference open-source datasets of intra-operative images [3, 29, 1], surgical journal papers [11, 28, 16], and open-source tutorials [21]. The materials were generated under the supervision of a clinician with surgical experience, and final procedural materials were inspected and verified by a clinician.

Due to the complexity of the generated textures, we provide screenshots of each of the reported textures, as well as example renders resulting from the Blender shader nodes. We additionally point the readers to the open-source implementation of the full shading graph that is provided for each texture in the SARAMIS repository: https://github.com/NMontanaBrown/saramis.

### D.2.1 Bowels

We consider the stomach, oesophagus, small bowel, duodenum, and the colon as the bowels. We texture the small bowel, duodenum, oesophagus and colon with the same texture, whilst maintaining a different texture for the stomach.

Summary of Blender shading node for the stomach is displayed in Fig.9. Due to the complexity of this graph, we refer the reader to the implementation provided in Blender in the SARAMIS for full detail.

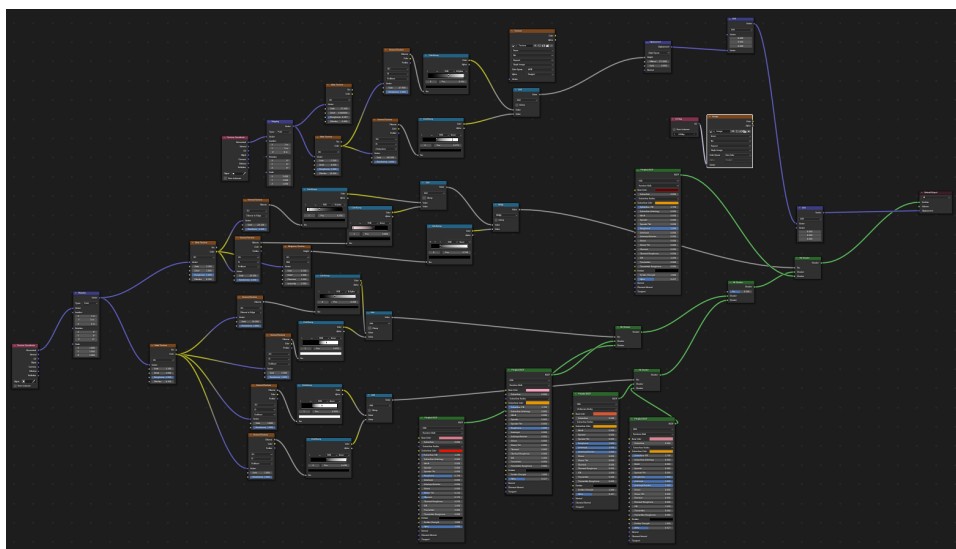

Figure 9: Summary of Blender shading graph generated with stomach material

We showcase renders of the stomach in Fig. 11.

Summary of Blender shading node for the colon and bowels is displayed in Fig.10. Due to the complexity of this graph, we refer the reader to the implementation provided in Blender in the SARAMIS for full detail.

We showcase renders of the colon in Fig. 12.

### D.2.2 Liver, Pancreas, Gallbladder, Spleen, Kidneys, Adrenal Glands

The liver, pancreas, gallbladder and spleen have a reference textures derived from the Dresden Anatomy Dataset [3] and the Cholec80k dataset [29]. The adrenal glands were textured using the pancreas texture, due to similarity of found reference image of healthy adrenal glands[16] to the pancreas reference images. The kidneys were textured using the liver texture, again due to similarity between liver and kidney textures.

The liver shader graph is split across two figures (Fig.13 - 14),

The gallbladder shader graph is split across two figures (Fig.15 - 16).

We showcase example renders of the liver and gallbladder in Fig. 21.

The pancreas shader graph is described in Figs.17 - 18.

We showcase example renders of the pancreas in Fig. 23.

The pancreas shader graph is described in Figs.19 - 20.

We showcase example renders of the spleen in Fig. 22.

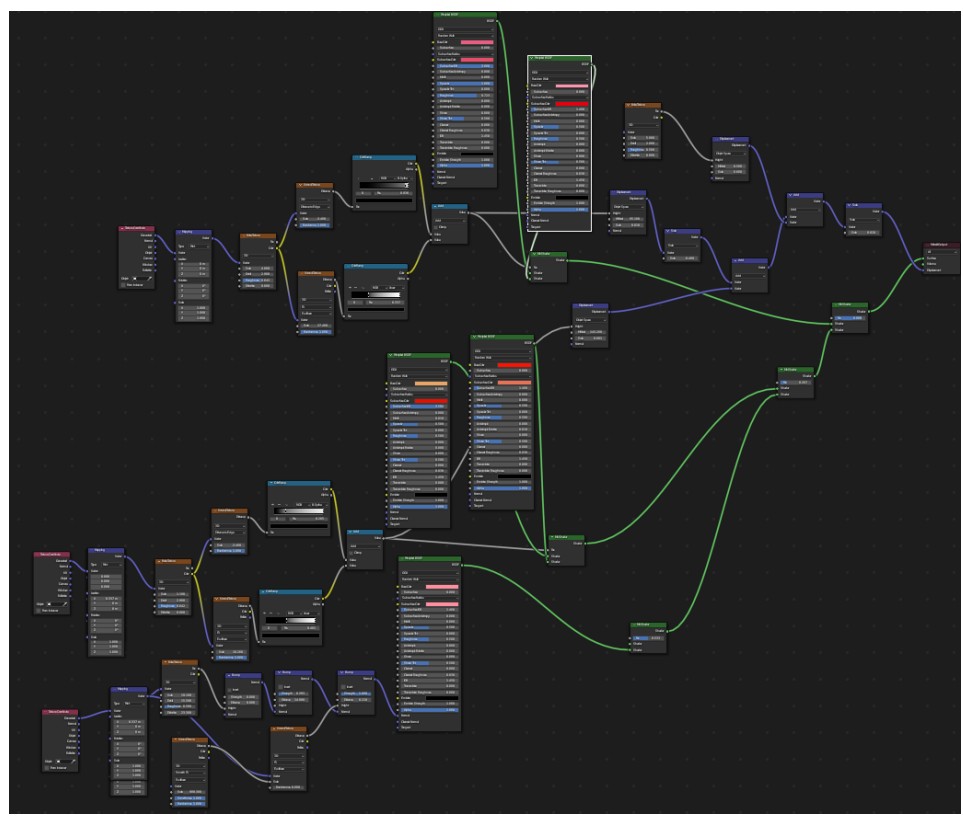

Figure 10: Summary of Blender shading graph generated with colon material

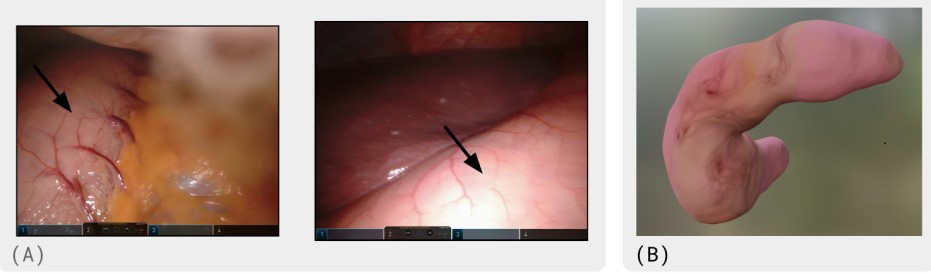

(A)   (B)

Figure 11: Reference images for stomach, alongside Blender renders of procedural texturing and shading of a stomach. Panel A) shows reference images from the Dresden Surgical Dataset [3] used to generate the procedural nodes, panel B) showcases a Blender render of the procedural textures on a `SARAMIS` stomach.

### D.2.3   Lungs, Bone, and Muscle

The lungs, bone and muscle were textured by referencing a surgical journal [11], an open source tutorial [21], and a surgical journal respectively [28]; we showcase renders in Fig. 24. Bone shader graph is included in Fig. 25 and muscle shader graph is included in Fig. 26. Due to complexity, the lung shader graph is split across three figures (Figs. 27 - 29).

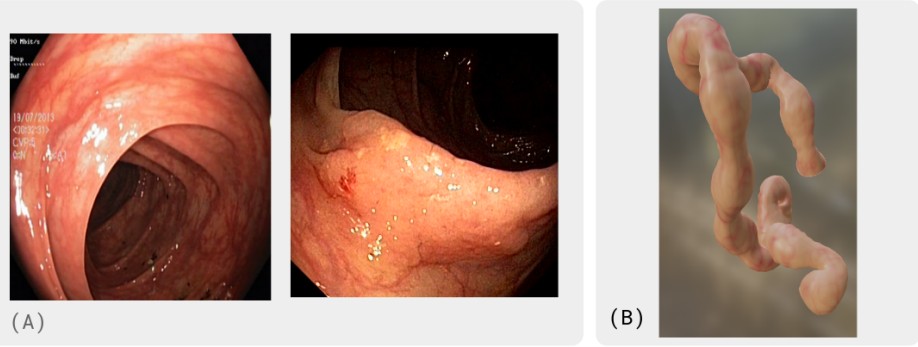

Figure 12: Reference images for the colon, alongside Blender renders of procedural texturing and shading of a colon. Panel A) shows reference images from the HyperKvasir dataset [1] used to generate the procedural nodes, panel B) showcases a Blender render of the procedural textures on a SARAMIS procedurally generated colon.

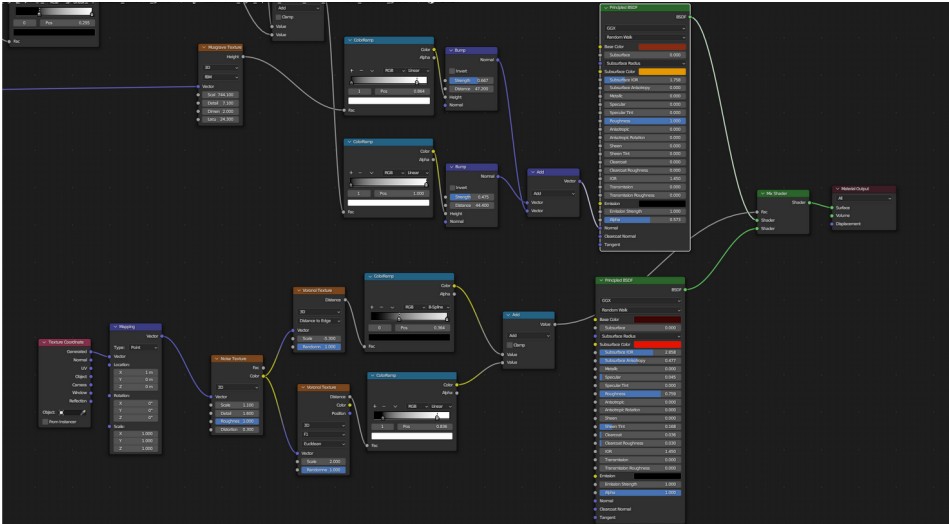

Figure 13: 1st half of Blender shading graph generated with liver material, outlining mainly the color mapping.

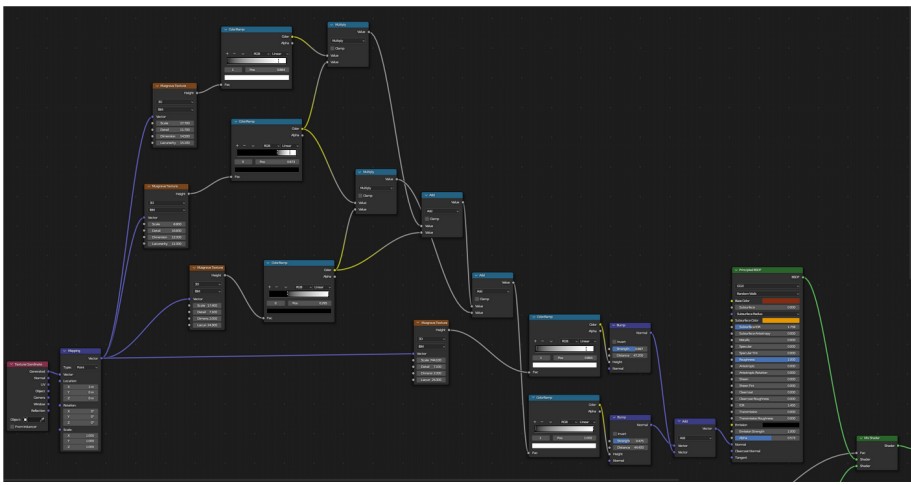

Figure 14: 2nd half of Blender shading graph generated with liver material, outlining mainly the bump map generation of the BSDF.

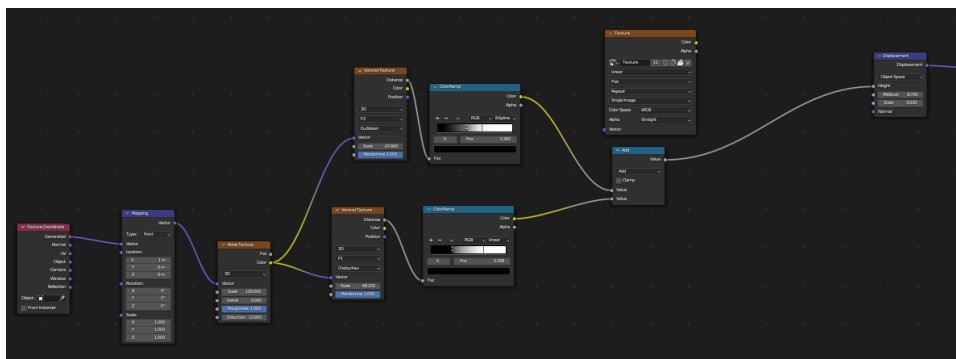

Figure 15: 1st half of Blender shading graph generated with gallbladder material, outlining mainly the color mapping.

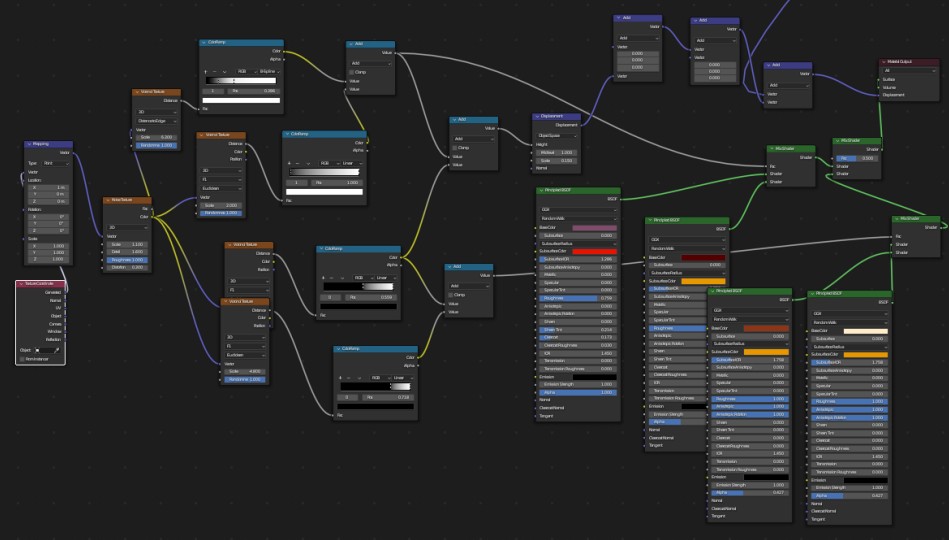

Figure 16: 2nd half of Blender shading graph generated with galbladder material, outlining mainly the bump map generation of the BSDF.

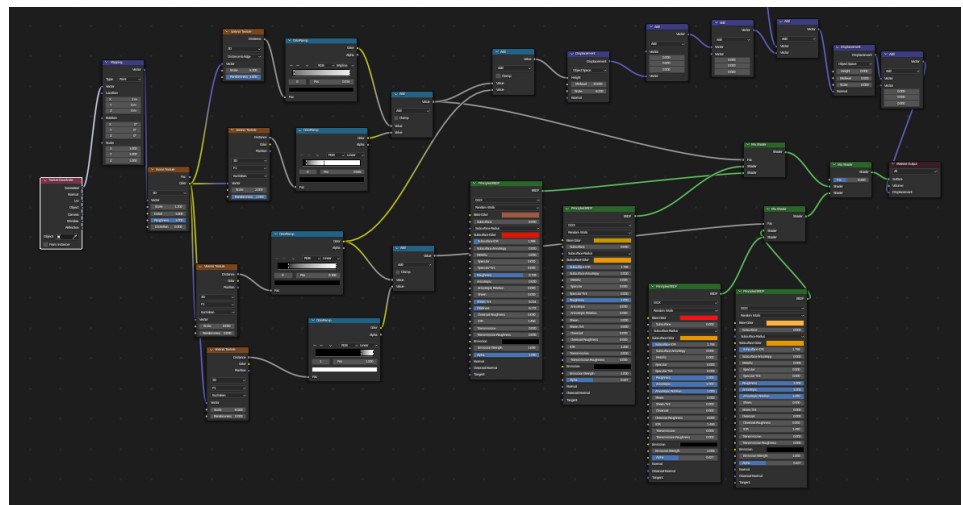

Figure 17: 1st half of Blender shading graph generated with pancreas material, outlining mainly the color mapping.

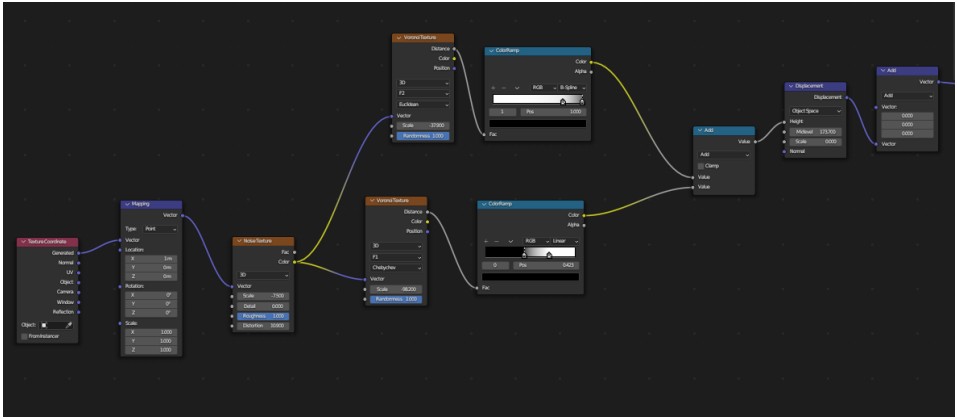

Figure 18: 2nd half of Blender shading graph generated with pancreas material, outlining mainly the bump map generation of the BSDF.

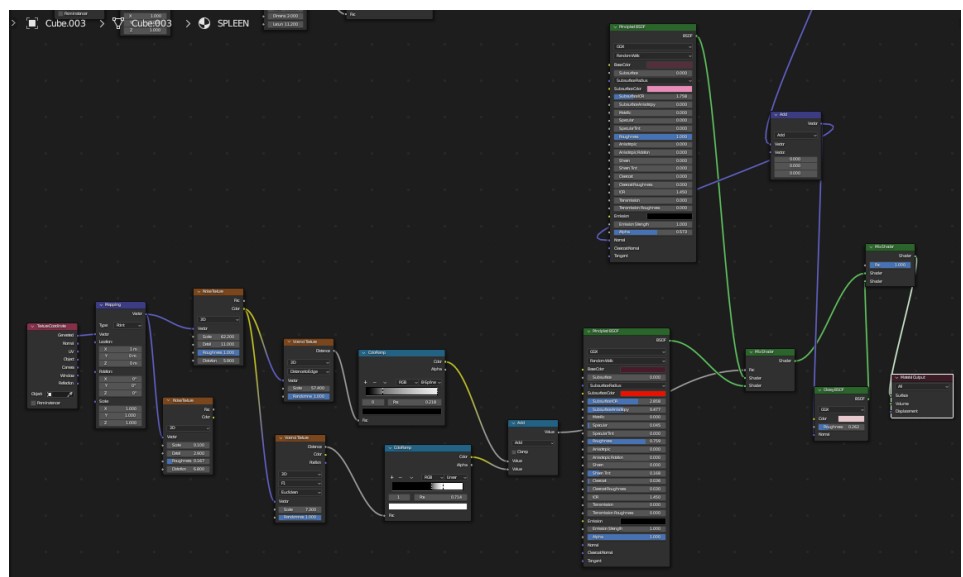

Figure 19: 1st half of Blender shading graph generated with spleen material, outlining mainly the color mapping.

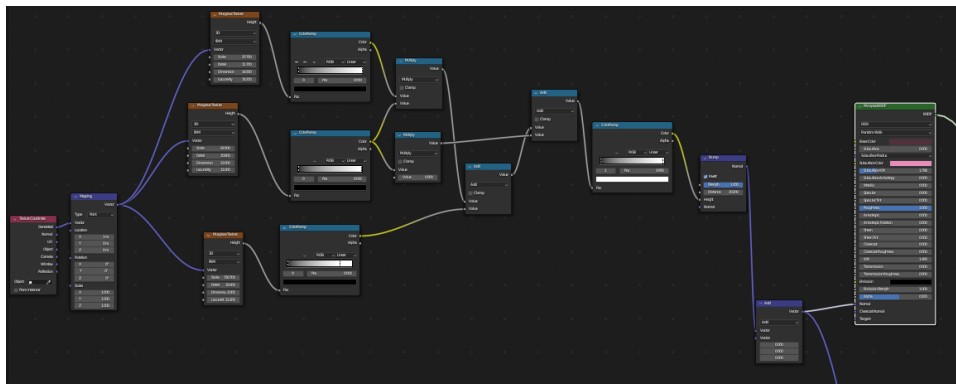

Figure 20: 2nd half of Blender shading graph generated with spleen material, outlining mainly the bump map generation of the BSDF.

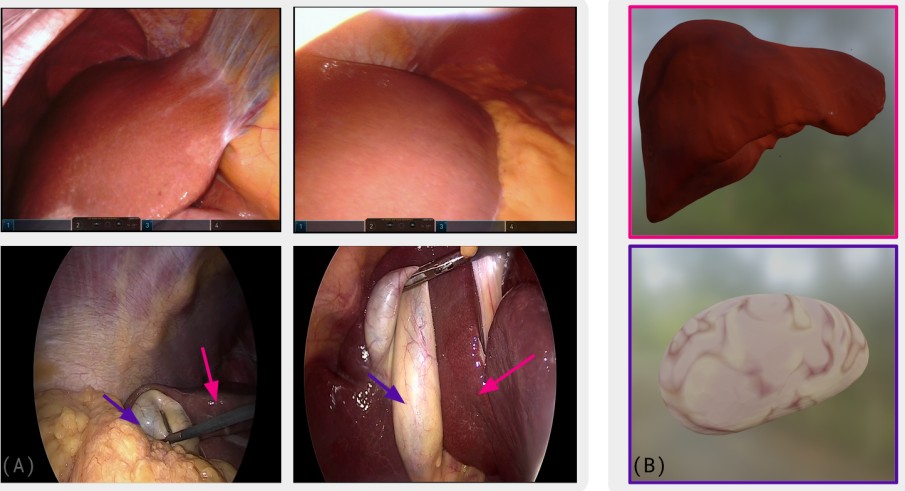

Figure 21: Reference images for liver and gallbladder, alongside Blender renders of procedural texturing and shading of the same structures. Panel A) shows reference images used to generate the procedural nodes (top row: liver, from the Dresden Anatomy dataset [3], bottom row: liver signalled with a pink arrow, gallbladder with a purple arrow, from the Cholec80k [29] dataset). Panel B) showcases Blender renders of the procedural textures, top showing the liver and bottom showing the gallbladder.

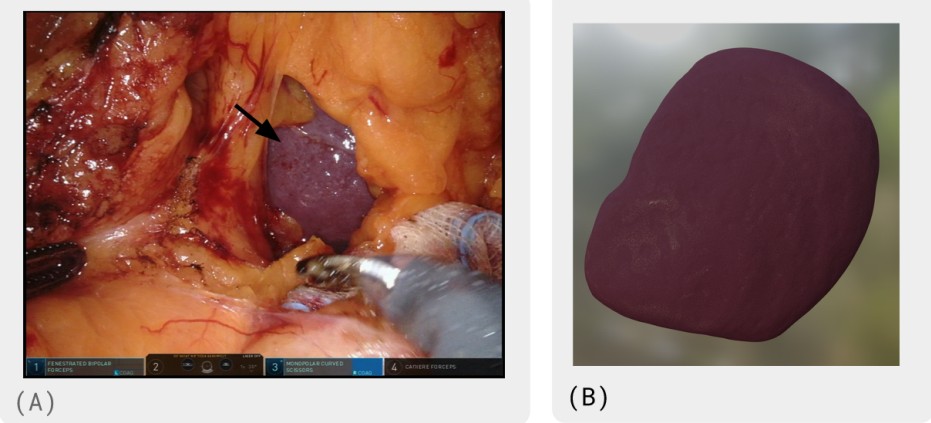

Figure 22: Reference image for the spleen with a Blender render of a SARAMIS spleen. Panel A) shows reference images used to generate the procedural nodes from the Dresden Anatomy dataset [3]. Panel B) showcases Blender renders of the procedural textures on a SARAMIS spleen.

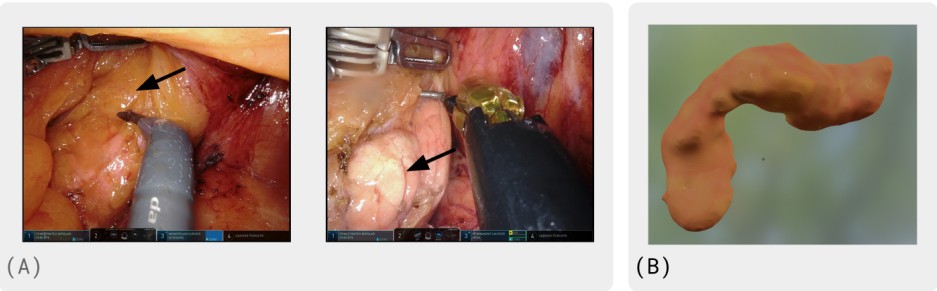

Figure 23: Reference images for pancreas, alongside Blender renders of procedural texturing and shading of a pancreas. Panel A) shows reference images from the Dresden Surgical Dataset [3] used to generate the procedural nodes, panel B) showcases a Blender render of the procedural textures on a SARAMIS pancreas.

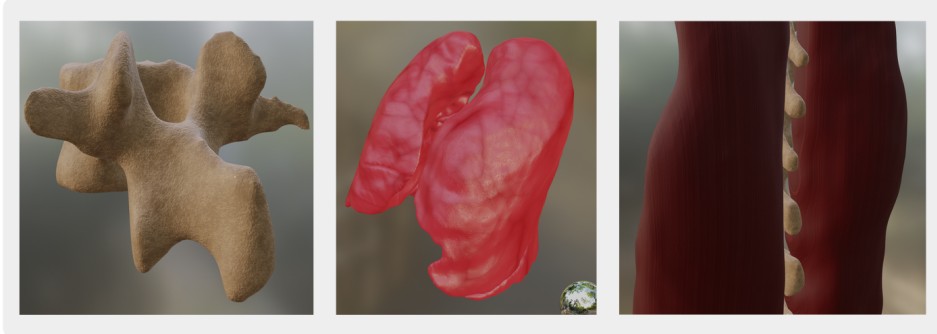

Figure 24: Blender renders for bone (left), lungs (center), and muscle (right). Bone material was generated with reference to [21], lungs with reference to [11], and muscle with reference to [28]

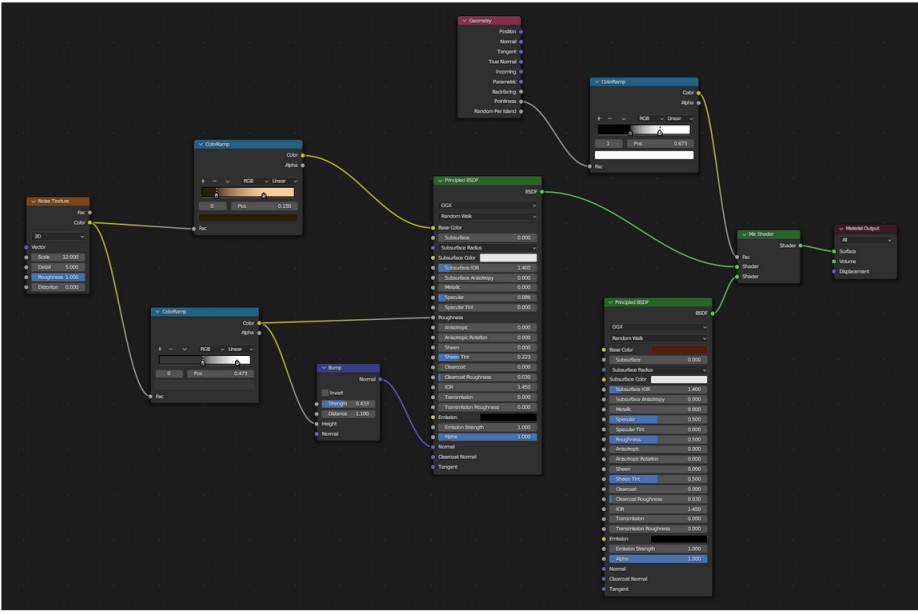

Figure 25: Blender shading graph generated with reference to [21] for bone material

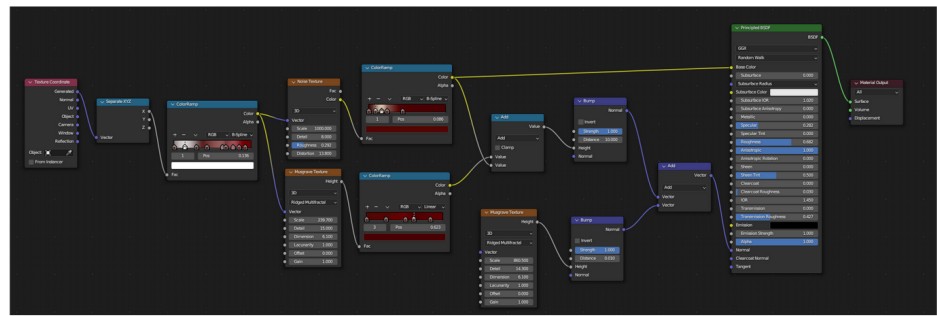

Figure 26: Blender shading graph generated with reference to [28] for muscle material

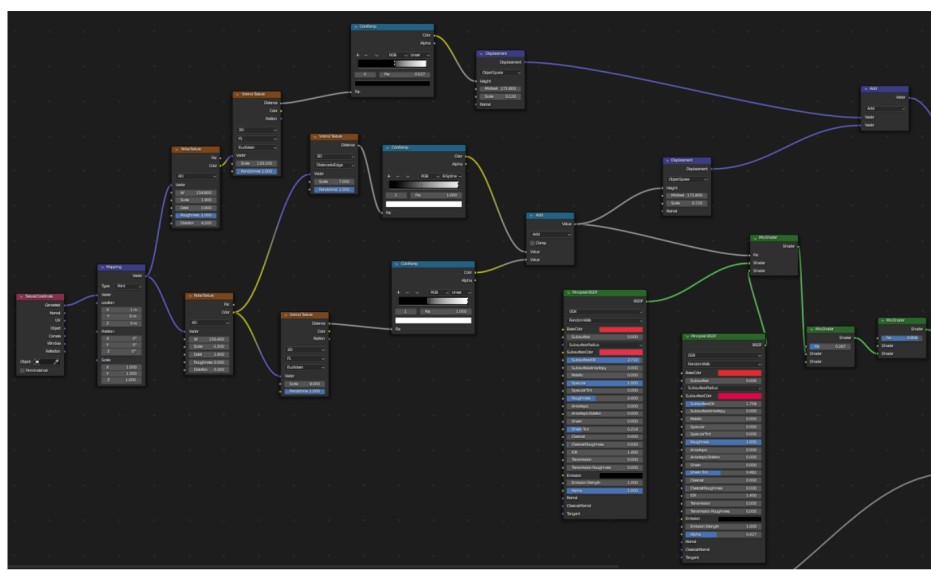

Figure 27: 1st 3rd of Blender shading graph generated with reference to [11] for lung material

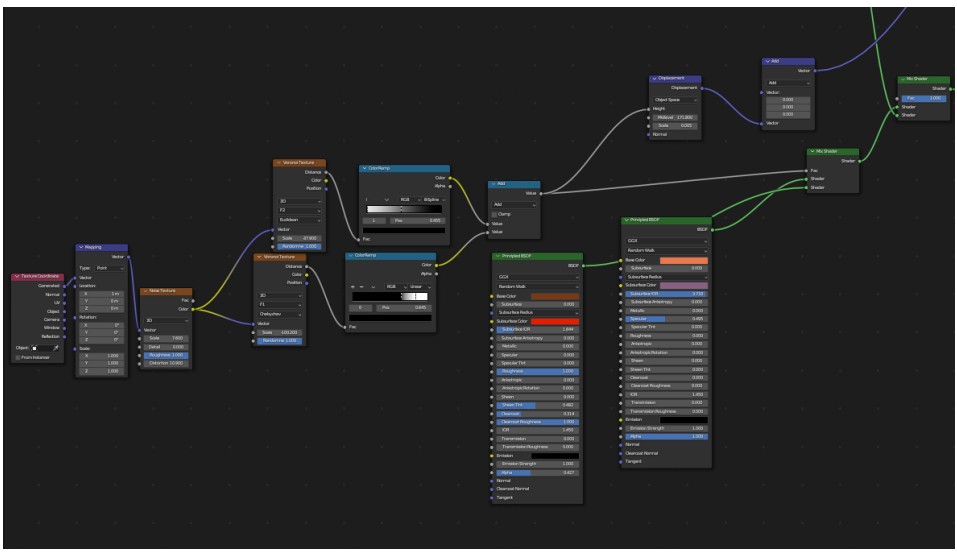

Figure 28: 2nd shading graph generated with reference to [11] for lung material. The output of the mix shader node from Fig.27 (lower furthest right green node) is connected to the mix shader node on the far right of this figure. The output of the vector math node (purple, furthest right node) is connected to an add math node in Fig.29.

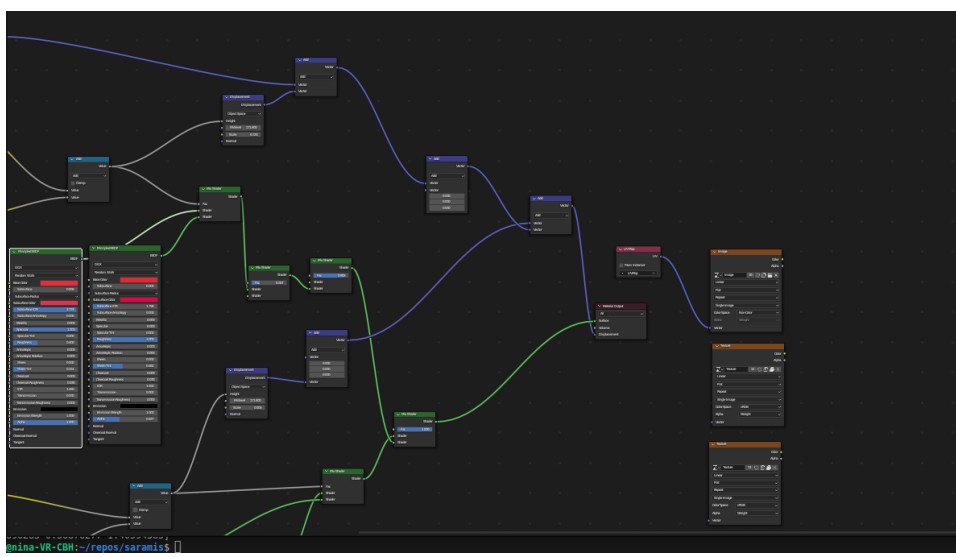

Figure 29: 3rd shading graph generated with reference to [11] for lung material, which combines outputs from the previous two portions of the graph to the final texture output.

## E  Trajectory Comparison Between Human and RL Agent

We compare human and RL performance by plotting five trajectories obtained on test cases of the TotalSegmentator sub-test set in Fig. 30. To better represent the colonoscopy case, we set the navigation target to the caecum and initialise navigation from the rectum (highlighted in blue and green bounding boxes, respectively). Human trajectories are qualitatively found to be smoother than RL trajectories.

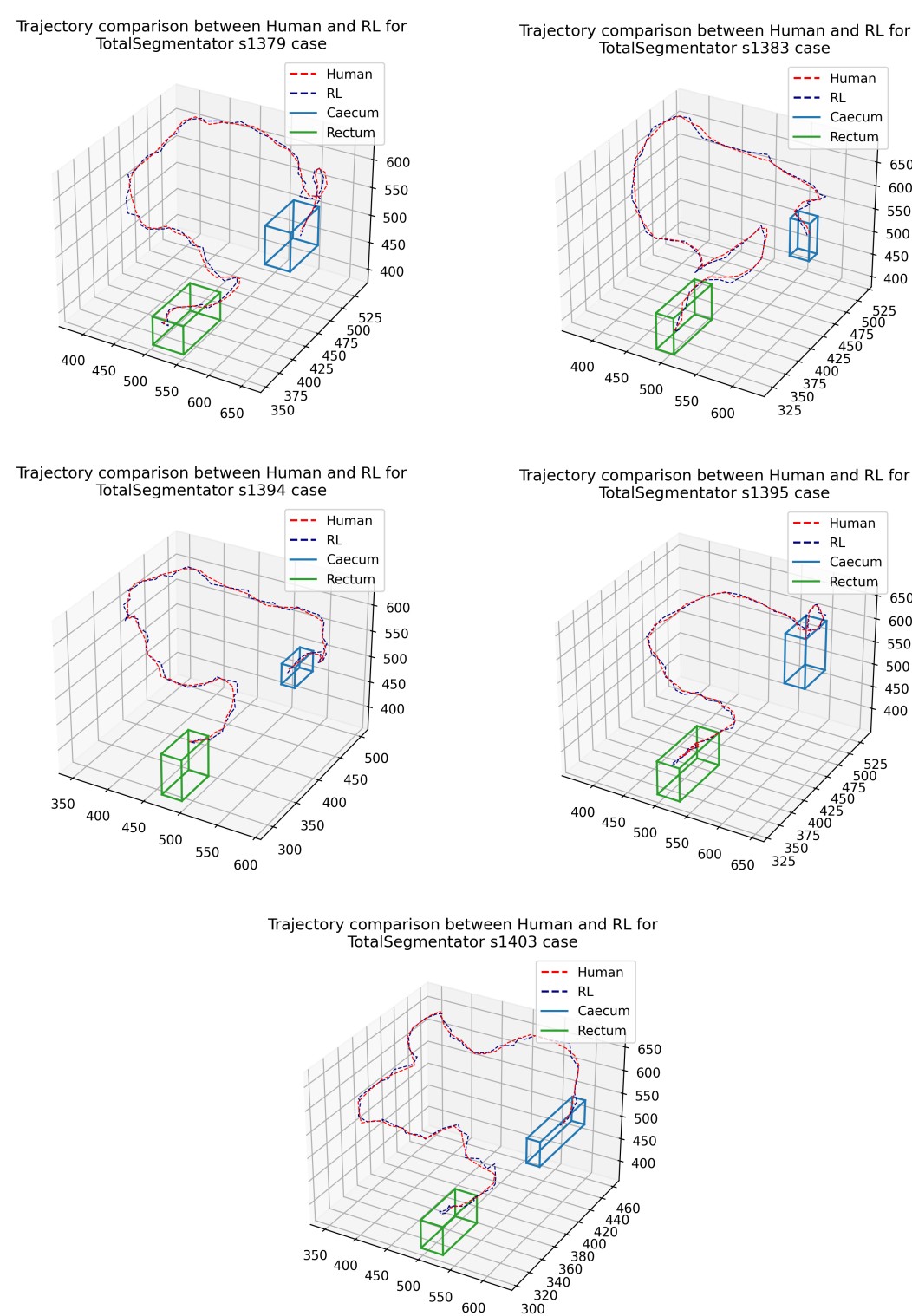

Figure 30: Comparison of trajectories between Human and RL agent to navigate from the rectum to the caecum for 5 cases in the held-out test set. Human and RL trajectories are plotted in red and navy dashed lines, respectively. We additionally designate the bounding box of the navigation target in green and blue for the rectum and caecum, respectively.

## F    Mechanical Properties of Human Tissue

We collate the values reported in [23] for the majority of soft tissue, [19] for bone, and [26] for the pancreas as a dictionary in the SARAMIS source code. We summarise said values in the following table.

| Organ | Elastic Modulus (MPa) | Standard Deviation (MPa) | Reference |
|---|---|---|---|
| muscle | 1.58 | 0.64 | [23] |
| brain | 0.00366 | 0.00012 | [23] |
| oesophagus | 0.004 | 0.014 | [23] |
| lung | 0.0034 | 0.002 | [23] |
| liver | 0.006 | 0.002 | [23] |
| galbladder | 0.641 | 0.028 | [23] |
| stomach | 0.005 | - | [23] |
| spleen | 0.0245 | 0.006 | [23] |
| pancreas | 0.002 | 0.004 | [26] |
| colon | 1.19 | 1.23 | [23] |
| small bowel | 2.69 | 0.37 | [23] |
| kidney | 41.5 | - | [23] |
| urinary bladder | 1.9 | 0.2 | [23] |
| bone | 179000 | 3900 | [19] |

## G    Mesh Analysis and Resolution

To better quantify the resolution for the meshes, we report the additional analyses:

- Number of vertices per mesh: We report the average number of vertices per mesh, split by organ.

- Surface area of meshes: in mm$^2$ units, we calculate the mesh surface area as the sum of triangular face areas through the area of each triangle.

- Average vertex density over the surface of the mesh: we additionally report the mesh density as vertices per 1mm$^2$ by dividing the total mesh surface area by the number of vertices.

We report the mean number of vertices per mesh, total surface area of meshes, sorted by number of vertices and by surface area, and mean vertex density in Figs. 31 - 34.

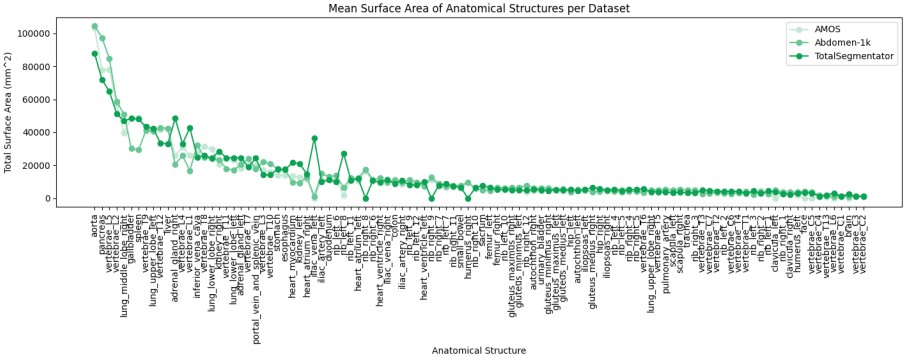

Figure 31: Mean surface area per organ sorted in descending order, split by dataset.

We find that the surface area shows a large level of variation, whilst the number of vertices per organ is more homogenous. Fig. 31 shows the surface area is not necessarily correlated with size, as the second, third, and fourth largest organs on average across datasets are the pancreas, and two vertebral bodies (L5, and L2), which are comparatively small structures volumetrically compared to, for example, the liver (ranked 11th), which is one of the largest internal organs of the human

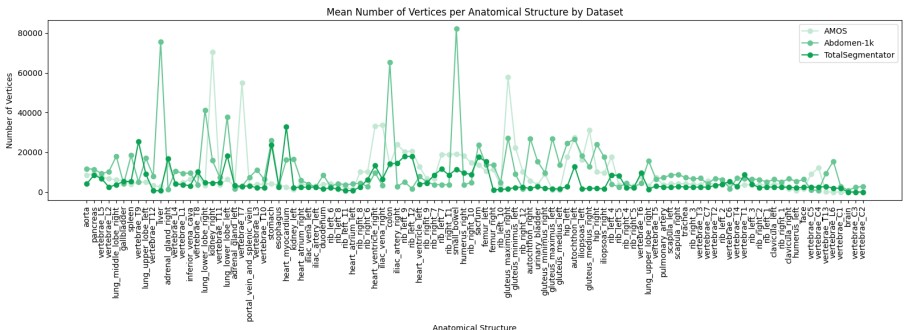

Figure 32: Mean number of vertices per organ sorted in order of largest to smallest mean surface area, split by dataset.

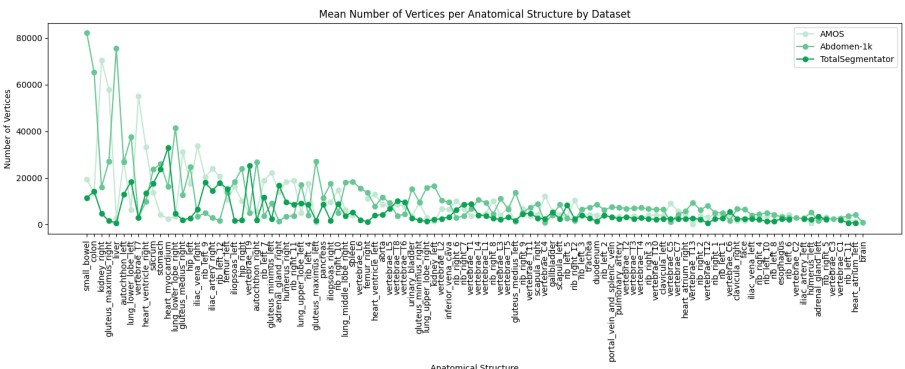

Figure 33: Mean number of vertices per organ sorted in descending order, split by dataset.

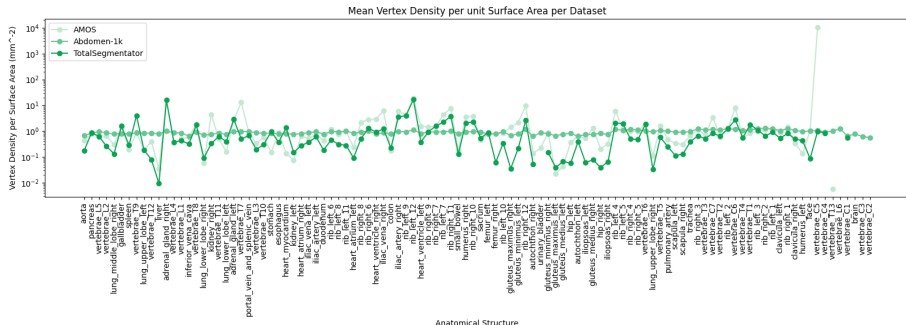

Figure 34: Mean vertex density per mm$^2$ per organ sorted in descending order, split by dataset. For the sake of comparison, the y-axis is plotted on a log-scale.

anatomy. The surface area per organ is consistent amongst organ types across datasets, whilst the number of vertices is more variant amongst datasets (Fig. 33. However, the number of vertices is more intuitively correlated to organ size, as the small bowel, liver, colon, and gluteus are amongst the organs with most vertices (Fig.32). We find no significant trend in the mean vertex density as reported in Fig. 34. We find that the Abdomen-1k is the most homogenous in terms of mesh density across organs, with the TotalSegmentator the least homogenous.

We recommend that future users of the dataset take care to resample the provided meshes to best suit their use, as this may impact performance in graph-type deep learning methods or otherwise.

## H Labelling Analysis of Registered Colons

In order to evaluate the label quality for the colon landmarking annotation, we perform a sub-analysis of the resulting registered classification of colon centerlines with respect to manual labelling of anatomical landmarks. We manually annotate a random sub-sample of 30 centerlines from the initial 155 colon TotalSegmentator colon subset using the same protocol as the labelling procedure for the first template colon reported in the paper. We then compare the resulting registered indices obtained from the deformeable registration of the template for the given colons to the manual labels.

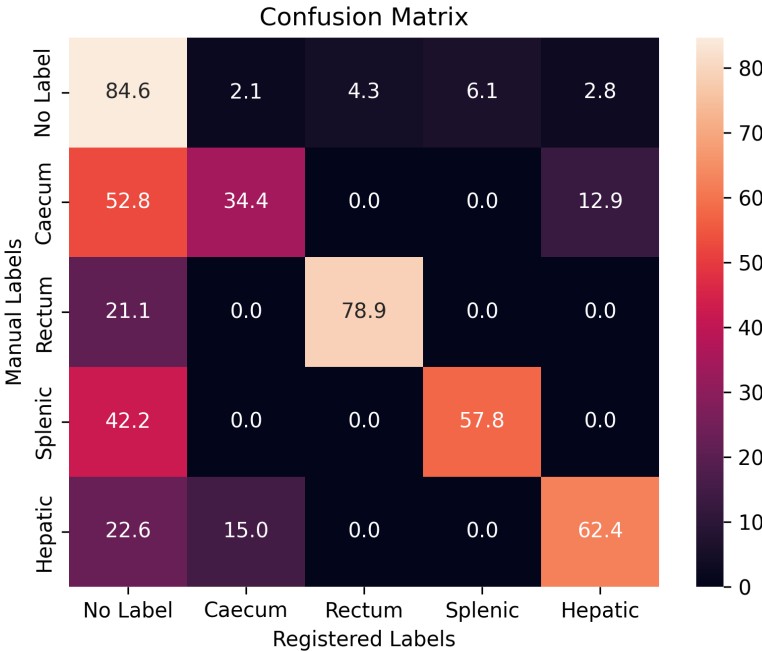

Figure 35: Confusion matrix comparing manual labelling and registration-procedure labelling for the landmark identification for colon experiments. Values for each cell are row-normalised to represent percentage of manual labels classified with a given registration label.

Fig. 35 shows that, for every class bar the caecum, there is a majority correct classification of the labels from registration. Additionally, all registered landmark locations have the most class confusion with the no-label category. This impacts the caecum most highly. The caecum landmark designates the beginning of the large colon - however, authors chose to empirically assign a larger area from the start of the colon towards the hepatic flexure in order to generate more general navigation targets for the navigation task. In particular, the specific task of colonoscopy involves navigation of the endoscope up to the visualisation of the caecum landmark, with subsequent withdrawal of the endoscope from the colon.

The more significant mis-classification in this case could be attributed to the template labelling being under-labelled in comparison to the subsequent labelled colons. The authors also note the high levels of empirically observed anatomical variability, showcased qualitatively in Fig. 36. This, coupled with the anatomical proximity of the caecum and hepatic flexure area(see Fig. 3A in the manuscript for anatomical description of colon and Fig. 36 for further description), could additionally explain class confusion between these labels.

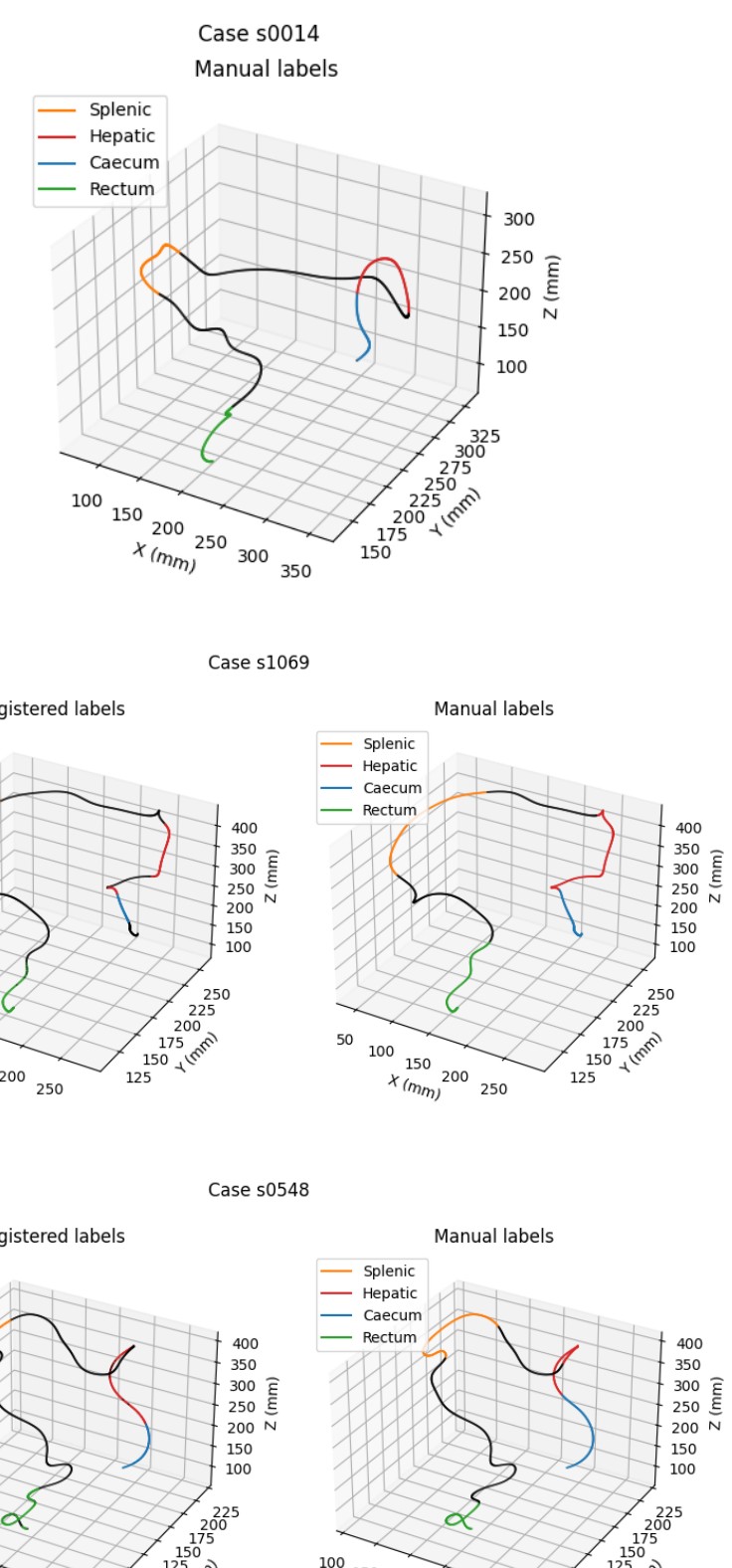

Figure 36: Comparison of labels obtained for navigation targets from manual labelling process and from template registration to case s0014 (upper panel).

# I Analysis of Organ Changes per Dataset

We report the number of organs changed over the AMOS and Abdomen-1k datasets in Fig. 37, the absolute mean number of pixels changed per anatomical structure in Fig. 38. as well as mean and inter-quartile ranges for absolute number of pixels changed for all the organs split by dataset in Tabs. 2 and 3.

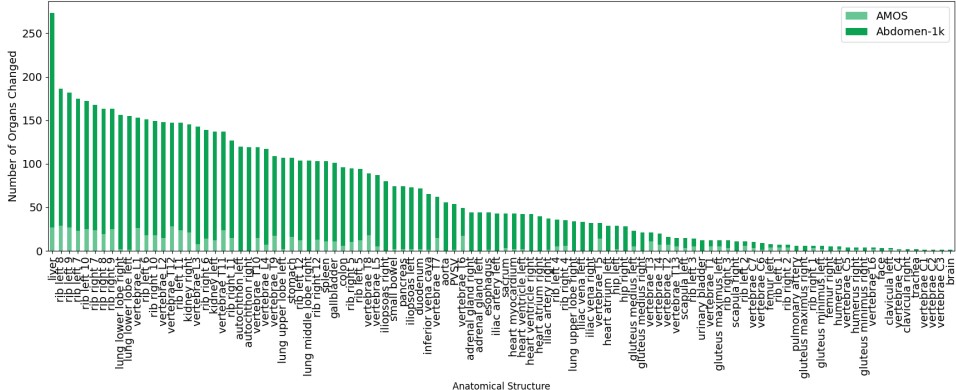

Figure 37: Number of organs changed sorted from most unique organs changed overall to least organs changed overall. We additionally split the organs by dataset.

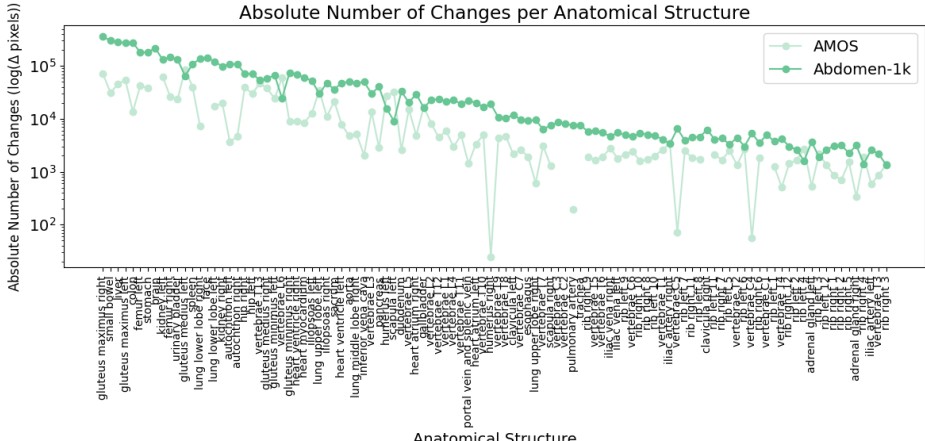

Figure 38: Absolute mean value of pixels changed per anatomical structure on a log scale for the Abdomen-1k and AMOS sub-datasets

| Organ | AMOS | Abdomen-1k |
|---|---|---|
| spleen | 40254 [13726, 207] | 107077 [169263, 713] |
| kidney right | 19658 [28691, 2089] | 97264 [115416, 4785] |
| kidney left | 62477 [28905, 4720] | 131244 [144595, 8090] |
| gallbladder | 16814 [16738, 1621] | 16029 [26273, 2714] |
| liver | 45368 [12832, 55] | 282409 [51956, 57] |
| stomach | 37894 [28636, 1168] | 181320 [262508, 1258] |
| aorta | 4743 [2675, 4] | 50423 [78268, 159] |
| inferior vena cava | 2044 [2313, 7] | 49785 [73094, 100] |
| portal vein and splenic vein | 1421 [943, 118] | 22245 [29468, 7968] |
| pancreas | 2852 [2330, 45] | 40363 [59134, 1220] |
| adrenal gland right | 336 [600, 48] | 3129 [4296, 2148] |
| adrenal gland left | 530 [582, 250] | 3691 [4899, 2263] |
| lung upper lobe left | 34737 [172, 16] | 29645 [30260, 25] |
| lung lower lobe left | 17482 [2431, 38] | 119883 [135844, 34] |
| lung upper lobe right | 614 [164, 6] | 9457 [220, 21] |
| lung middle lobe right | 5086 [226, 26] | 47317 [81134, 27] |
| lung lower lobe right | 7288 [337, 9] | 137421 [91826, 16] |
| vertebrae L6 | 60114 [69702, 50527] | 24304 [26251, 22358] |
| vertebrae L5 | 14911 [12573, 991] | 20360 [24284, 1286] |
| vertebrae L4 | 2931 [3264, 345] | 22896 [34428, 1674] |
| vertebrae L3 | 13693 [18277, 2818] | 29953 [40274, 3182] |
| vertebrae L2 | 8165 [10166, 792] | 22827 [32399, 2431] |
| vertebrae L1 | 5884 [6392, 574] | 21508 [30514, 1888] |
| vertebrae T13 | 46418 [54553, 43623] | 53719 [74064, 19380] |
| vertebrae T12 | 4463 [6206, 916] | 23546 [28126, 3992] |
| vertebrae T11 | 4949 [7386, 918] | 18956 [26029, 1496] |
| vertebrae T10 | 4957 [5315, 742] | 16774 [22854, 1681] |
| vertebrae T9 | 4295 [6719, 623] | 10655 [17207, 1355] |
| vertebrae T8 | 4652 [5910, 570] | 10156 [10454, 550] |
| vertebrae T7 | 3106 [3003, 343] | 6360 [8371, 231] |
| vertebrae T6 | 1911 [1932, 134] | 5572 [8153, 336] |
| vertebrae T5 | 1659 [964, 173] | 5863 [7353, 518] |
| vertebrae T4 | 509 [605, 80] | 4196 [7665, 781] |
| vertebrae T3 | 862 [816, 407] | 2140 [1349, 221] |
| vertebrae T2 | 1339 [2154, 472] | 4409 [10900, 41] |
| vertebrae T1 | 2623 [4109, 868] | 4119 [6546, 1092] |
| vertebrae C7 | 2569 [5040, 90] | 9632 [11294, 5442] |
| vertebrae C6 | 2416 [3533, 1298] | 4649 [7724, 249] |
| vertebrae C5 | 72 [72, 72] | 6502 [7568, 5313] |
| vertebrae C4 | 56 [56, 56] | 5303 [5303, 5303] |
| vertebrae C3 | 0 [0, 0] | 8519 [8519, 8519] |
| vertebrae C2 | 0 [0, 0] | 8021 [8021, 8021] |
| vertebrae C1 | 0 [0, 0] | 5017 [5017, 5017] |
| esophagus | 1921 [2885, 576] | 9182 [12522, 5390] |
| trachea | 0 [0, 0] | 7656 [10682, 4630] |
| heart myocardium | 8446 [10273, 94] | 60413 [82100, 41478] |
| heart atrium left | 3350 [4782, 485] | 20086 [36033, 3443] |
| heart ventricle left | 7722 [10418, 1099] | 47734 [66295, 26855] |
| heart atrium right | 4892 [8447, 395] | 29474 [46517, 11613] |
| heart ventricle right | 9008 [12538, 2281] | 69065 [91937, 44546] |
| pulmonary artery | 191 [191, 191] | 7588 [10412, 212] |
| brain | 0 [0, 0] | 216214 [216214, 216214] |
| iliac artery left | 585 [1083, 12] | 2623 [1942, 92] |

Table 2: Mean [IQR:75, IQR:25] absolute pixel changes for each organ in the AMOS and Abdomen-1k subsets.

| Organ | AMOS | Abdomen-1k |
|---|---|---|
| iliac artery right | 3365 [5231, 2062] | 3366 [2833, 121] |
| iliac vena left | 1794 [2564, 268] | 5489 [2733, 29] |
| iliac vena right | 2773 [4005, 164] | 4623 [2656, 108] |
| small bowel | 31206 [8907, 624] | 299051 [435509, 1652] |
| duodenum | 2546 [2394, 24] | 32777 [49035, 298] |
| colon | 13696 [6763, 68] | 274284 [392699, 66] |
| rib left 1 | 1246 [1556, 544] | 3725 [4580, 3149] |
| rib left 2 | 1676 [1156, 667] | 2597 [3174, 1409] |
| rib left 3 | 2149 [1365, 266] | 1874 [3145, 259] |
| rib left 4 | 2673 [2994, 213] | 1594 [1970, 332] |
| rib left 5 | 2528 [2367, 656] | 3279 [3031, 712] |
| rib left 6 | 2653 [4012, 609] | 3005 [3780, 713] |
| rib left 7 | 2459 [2278, 201] | 3927 [5692, 857] |
| rib left 8 | 1729 [1620, 358] | 4560 [5656, 950] |
| rib left 9 | 2111 [2796, 292] | 4986 [6648, 942] |
| rib left 10 | 1986 [3337, 222] | 4778 [6532, 799] |
| rib left 11 | 2089 [2948, 595] | 4097 [6350, 748] |
| rib left 12 | 1344 [1756, 356] | 2614 [3305, 925] |
| rib right 1 | 850 [1090, 631] | 3078 [3790, 2210] |
| rib right 2 | 699 [813, 603] | 3152 [4710, 1852] |
| rib right 3 | 1366 [808, 363] | 1353 [1755, 566] |
| rib right 4 | 1906 [3596, 307] | 1406 [2105, 247] |
| rib right 5 | 1525 [2240, 368] | 2248 [3100, 750] |
| rib right 6 | 1825 [2680, 701] | 3484 [4009, 873] |
| rib right 7 | 1672 [2864, 464] | 4279 [5847, 764] |
| rib right 8 | 1719 [2349, 400] | 5067 [7190, 1008] |
| rib right 9 | 1919 [2322, 495] | 5715 [8588, 941] |
| rib right 10 | 1588 [2900, 83] | 5383 [8037, 648] |
| rib right 11 | 1817 [2271, 280] | 4514 [6433, 656] |
| rib right 12 | 1432 [1678, 344] | 3002 [3400, 928] |
| humerus left | 26854 [30282, 23425] | 15797 [23688, 970] |
| humerus right | 25 [29, 14] | 18858 [28272, 2188] |
| scapula left | 32167 [46963, 17] | 9068 [10483, 936] |
| scapula right | 1293 [1552, 924] | 7481 [6704, 204] |
| clavicula left | 2162 [3236, 94] | 11974 [11974, 11974] |
| clavicula right | 0 [0, 0] | 6196 [9209, 3184] |
| femur left | 42400 [85472, 3962] | 180016 [227985, 106850] |
| femur right | 26289 [36045, 19586] | 145451 [193543, 102826] |
| hip left | 29813 [35163, 3772] | 71961 [27042, 3071] |
| hip right | 39340 [76328, 690] | 70342 [29356, 1914] |
| sacrum | 21541 [36350, 986] | 35899 [23639, 1036] |
| face | 0 [0, 0] | 144310 [216377, 446] |
| gluteus maximus left | 54101 [99168, 5099] | 271042 [466883, 589] |
| gluteus maximus right | 70083 [104624, 15570] | 358701 [590127, 66084] |
| gluteus medius left | 85769 [160655, 14631] | 64736 [45730, 377] |
| gluteus medius right | 37802 [48854, 30561] | 57151 [33615, 183] |
| gluteus minimus left | 24065 [56203, 1229] | 67576 [94469, 31866] |
| gluteus minimus right | 8872 [12980, 6596] | 74194 [105394, 35438] |
| autochthon left | 3668 [47, 7] | 109548 [188674, 6] |
| autochthon right | 4593 [48, 8] | 107824 [180978, 12] |
| iliopsoas left | 12756 [9840, 6] | 52243 [59470, 8] |
| iliopsoas right | 10963 [850, 19] | 47540 [61348, 11] |
| urinary bladder | 23455 [20369, 3434] | 131854 [121914, 275] |

Table 3: Mean [IQR:75, IQR:25] absolute pixel changes for each organ in the AMOS and Abdomen-1k subsets.

## J Reinforcement Learning Training Algorithm

The training procedure to obtain an optimised policy $\pi_{\theta*}$ which maximises the cumulative reward, representative of navigation performance, is summarised in Algo. 2. After training, this policy may be used to perform navigation intraoperatively.

**Data:** Patient volumes from which to sample camera images $s_t \in \mathcal{S}$
**Result:** Trained RL policy $\pi_{\theta*}$.

**while** *not converged* **do**
  Randomly sample a patient volume;
  Start at $t = 0$;
  Randomly sample a camera pose $c_0 \in \mathbb{R}^6$ within the volume;
  Render the camera image $s_0$ at pose $c_0$;
  Sample the action $a_0$ according to the policy $a_0 \sim \pi_\theta(a_0|s_0)$;
  Compute target-presence-based reward $R_0 = r(s_0, a_0)$;

  **for** $t \leftarrow 1$ **to** $T$ **do**
    Note: $t$ is now iterating starting at $t = 1$;
    Update the camera pose $c_t = c_{t-1} + a_{t-1}$;
    Render the camera image $s_t$ at pose $c_t$;
    Sample the action $a_t$ according to the policy $a_t \sim \pi_\theta(a_t|s_t)$;
    Compute target-presence-based reward $R_t = r(s_t, a_t)$;
    End if target presence detected i.e., at $t_{end}$;
  **end**

  Once $R_{t=1:T}$ or $R_{t=1:t_{end}}$ collected, update RL function using gradient ascent
**end**

**Algorithm 2:** Training procedure to train a navigation policy using reinforcement learning.