# OpenReview forum: "SARAMIS: Simulation Assets for Robotic Assisted and Minimally Invasive Surgery"
_NeurIPS.cc/2023/Track/Datasets_and_Benchmarks — NeurIPS 2023 Datasets and Benchmarks Poster_

### Official Review · Reviewer_zAJD · 2023-07-20
**A potentially nice in-silico dataset for endoscopic vision tasks with limited texture realism**

**Rating:** 7
**Confidence:** 4

**Strengths:**

Uniqueness: First openly accessible dataset that provides rendering assets of abdominal organs that can be used for in-silico validation of algorithms in the context of robot assisted and minimally invasive surgery.

Curation: Careful curation of three existing CT datasets

Open science: Both the dataset with supporting datasheet and all necessary processing procedures will be provided in publicly accessible repositories which enables straightforward reproduction and extension.

**Additional Feedback:**

-

**Clarity:**

The paper is well-written and structured. Remaining points for improvement are listed above in the “Opportunities for improvement”.

**Correctness:**

I want to underline that I was not able to judge the data itself as I was not able to access the sample data even with an AWS account. Hence, I cannot really recommend acceptance at this stage.

See opportunities for improvement for further comments.

**Documentation:**

The data set is well documented in the datasheet and the remaining parts of the Supplementary material. Although the full data set was not released yet, a URL was provided to access exemplary parts of the data set via AWS. However, as mentioned in the “Opportunities for improvement”, I couldn’t find an easy way to access this data without signing up for AWS.

**Ethics:**

-

**Limitations:**

As mentioned above (“Opportunities for improvement”), the realism of the textures might be a bottleneck for the application of the data set. I encourage the authors to thoroughly discuss this aspect.

**Opportunities For Improvement:**

IMPORTANT: The full dataset has not been released upon submission but will - according to the authors - be made publically available upon paper acceptance. Unfortunately, I was not able to access the sample data and can thus not fully judge the paper. Furthermore, I recommend allowing public data access without barriers, such as logins.

The following concerns should be taken into consideration:

Paper focus: As this is a data paper, I think the authors should put more focus on the actual data (especially the label quality) rather than description of their specific reinforcement learning method (whose details could be provided in the Appendix). Please describe exactly the purpose of the dataset and the applications it can be used for, also considering the so far limited realism of textures and lack of pathologies.

Label quality: “a single colon is manually labelled for all anatomically relevant landmarks [...]. Subsequently, the other 154 colons were deformably registered to the manually labelled colon in order to obtain” -> this seems like a very rough procedure. Please provide numbers on the accuracy of the landmarks generated with this process for at least a subset of the data. Also, an annotation protocol would help readers understand the whole annotation process in detail.

Texture realism: A central question to me is how the realism of the textures is assured? Please provide more details. How exactly is the information taken from previous data sets / surgical papers / tutorials transferred to the shader for texturing? The authors mention that the proposed texturing method could be paired with learning-based texturing. An initial attempt using existing methods on e.g. a subset of the data could be a nice addition.

Dataset release: In the datasheet, the questions “Has an analysis of the potential impact of the dataset and its use on data subjects (e.g., a data protection impact analysis) been conducted?” and “When will the dataset be distributed” have not been answered in the data sheet.

Experimental design: The design of the experiments leaves room for improvement. For example (1)  I doubt that any solid conclusions can be drawn from an experiment with four test cases. Please consider removing the experiment, increasing the number or at least performing uncertainty analyses (e.g. bootstrapping) to analyze the robustness of results. (2) Could you explain the purpose of the entire method (which problem that clinicians are facing are you trying to solve? Please provide references) and experiments a bit better from a colonoscopy perspective and then justify the choice of validation metrics accordingly? (3) What was the rationale for choosing a “biomedical imaging researcher” rather than a physician as a human observer?

Model overfitting: The authors report a training and test set. Based on which data was the method development performed? Specifically, did the authors avoid model overfitting? Meaning that they finalized the complete algorithm before running analyses on the test set only one single time?

Further improvements include:

- Overall, this results in “1048, 279, 1200 scans” -> these numbers should appear in Tab. 1 to avoid confusion (and to explain the number of images 2,527, which appears in the text but not in the table).
- Please explain how the “subset of 450 verified CT scans” images was chosen.
- The colon as a special case for segmentation and meshing could be separated a bit more clearly both in the annotation description and the details of the mesh generation.
-“Failure in the task is defined as the inability to navigate to the structure within 256 steps or collision with the colonic wall” -> please justify the choice of 256
- “A total of 155 colon meshes were selected from the SARAMIS dataset” -> please explain how?
- In the datasheet, the strategy for external contributions should be outlined in more detail, e.g. explaining quality control and communication of contributions.
- Please fix inconsistent writing 1,000 vs 1000
- Line 1: Abstract: RAMIS abbreviation misplaced before surgery
- Line 110: From the main text it’s not clear what it entails to become a “trained anatomical annotator”. Make sure to quickly mention this here and/or refer to the Supplementary Material for more detailed information
- Line 122ff: It remains unclear which resolution the final meshes have. Always the resolution of the underlying CT scan?
- Line 140: Appendix A in Supp. Material does not describe the matching algorithm but the dataset. I assume this should reference Appendix C.
- Line 170ff: I find it hard to bring the number of voxels changed into perspective. Wouldn’t it be more interesting to report for which classes changes were necessary?
- Line 197: Please specify what exactly this action entails. I assume it is translation + rotation of the camera?
- Line 228ff: How is the discount factor chosen?
- Line 236: How were the 155 colon meshes used for the experiments chosen?
- Line 244: How were the hyperparameters optimized if the data split was only into training and testing?
- Line 248: What does S.U.S refer to?
- Datasheet, line 326: How is the mesh decimation performed?
- Datasheet, line 333: Abbreviation BSDF has not been introduced.
- Datasheet, line 334: For someone new to blender it might not be obvious how those shader nodes have been created bsed on images / surgical information. Would be good to elaborate on this a bit more detailed.
- Datasheet, line 399: "excluded.From" missing space.
- Datasheet, line 580: How were the parameters b, k, h chosen for the procedural mesh generation and why?
- References [18] and [19] seem to be malformatted and incomplete. Make sure to provide sufficient information.
- Reference [16] in Supplementary Material is incomplete. E.g. missing Author last name/ Journal Name / issue / number / year
- Please use tenses consistently.
- Do you really need decimal numbers in your results? What is the relevance of a “.1 difference”?

**Relation To Prior Work:**

The authors bring their work nicely into the context of related previous contributions. To the best of my knowledge the proposed data set is the first of its kind in the domain of (robot-assisted) minimally invasive surgery.

**Summary And Contributions:**

This paper introduces a dataset (SARAMIS) consisting of rendering assets of over 100 anatomical structures of the human abdomen. The dataset is based on the postprocessing of three existing computed tomography datasets, from which meshes of various anatomical structures were extracted and textured. As an example application on the data set, the authors introduce an automated method for navigating to different targets within the colon. Carefully curated segmentations as well as public availability of all processing methods is a further contribution of this paper.

---

> ### Author Response · Authors · 2023-08-19
> **Response: Part 1 (Paper Focus)**
>
> We thank the reviewer for the highly detailed review, and provide a response for the opportunities for improvement below.
>
> > Paper focus: As this is a data paper, I think the authors should put more focus on the actual data (especially the label quality) rather than description of their specific reinforcement learning method (whose details could be provided in the Appendix).
>
>  Thank you for this suggestion. We have made a number of extensions to our work, both in the main manuscript and the Supplementary Materials. These ammendements are almost entirely focused on the data aspects of the paper, and include adding additional explanations, analyses, figures, tables, and conclusions as they relate to \texttt{SARAMIS}. These include:
>
>     * An analysis of mesh quality as indicated by number of vertices, mesh surface area, and mesh density in Appendix G (L681 Supp. Mat.). We report several figures containing the mean surface area, number of vertices, and mesh density (Figs. 31-34).
>    * An additional analysis of  the labelling accuracy as a result of the registration pipeline reported in L251-259).We label 30 colons manually, and compare the accuracy of obtained registered labels from template registration.
>     The results are reported in Fig. 35 (Supp. Mat. Appendix H L711).
>     * An analysis of the number of organs changed, and a table of all absolute changes with IQRs in Appendix I (L725 onwards).
>
> As a self-contained paper, we believe the RL experiment is important to demonstrate a possible application of this dataset, and therefore believe it reasonable to include this section in the manuscript.
>
> > Please describe exactly the purpose of the dataset and the applications it can be used for, also considering the so far limited realism of textures and lack of pathologies.
>
> We highlight and reiterate the potential uses of \texttt{SARAMIS} as described in the Datasheet for Datasets (Supplementary Materials Appendix B L454):
>
>   *  Synthetic data generation: the 3D models can be paired with a rendering environment to obtain 2D RGB images, 2D depth maps, 2D segmentation maps, and 2D optical flow images.
>     * Deformation simulation: the tetrahedral volumes provided can be used for the simulation of deformation of organs in a surgical setting.
>    *  Generative 3D models: The 3D models could be used to create a 3D generative model of given organs.
>    * Learning textures in surgery: the 3D models could be paired with real intra-operative video (2D RGB images) to learn how to texture different organs in the human body.
>     * Camera-pose estimation: pose labels may be generated from a rendering environment, paired with a 2D image, to learn how to perform camera pose-estimation on different organs in surgery.
>     * Navigation: Like the exemplified case in the paper for this dataset, different organs could be used to design surgical scenes or scenarios, to teach reinforcement learning algorithms how to navigate to different targets, how to perform certain actions (for example, laparoscopic incision/suturing of viscera), or how to interact with the shapes in the environment (e.g. surgical tool manipulation for tissue retraction).
>
> We acknowledge the limitation of our texturing approach with regards to generating pathological textures or structures.
> Whilst we mainly focus on texturing normative organs, the presence of pathology poses an interesting avenue of future work which will be important for the realism of navigation tasks where pathologies exist.
>
>  We have updated the limitations section of the paper to reflect the above points (L319).

---

> ### Author Response · Authors · 2023-08-19
> **Response: Part 2 (Label Quality)**
>
> >  Label quality: “a single colon is manually labelled for all anatomically relevant landmarks [...]. Subsequently, the other 154 colons were deformably registered to the manually labelled colon in order to obtain” -> this seems like a very rough procedure. Please provide numbers on the accuracy of the landmarks generated with this process for at least a subset of the data. Also, an annotation protocol would help readers understand the whole annotation process in detail.
>
> Firstly, we would like to clarify that the manual labelling of the first set of landmarks was performed under the supervision of a clinician.
> In order to address the reviewers concerns, we performed an additional analysis of label quality for the registered colon annotation.
> We manually annotate a random sub-sample of the training set of colons under the same protocol as the template colon reported in the paper.
> We then compare the resulting registered indices obtained from the deformable registration of the template for the given colons to the manual labels.
> Results are reported in the Supplementary Materials (Appendix H 'Labelling Analysis of Registered Colons' L704).
>
> Fig. 35 (Appendix H) shows that, for every class bar the caecum, there is a majority correct classification of the labels from registration.
> Additionally, all registered landmark locations have the most class confusion with the no-label category.
> This impacts the caecum most highly.
> The caecum landmark designates the beginning of the large colon - however, authors chose to empirically assign a larger area from the start of the colon towards the hepatic flexure in order to generate more general navigation targets for the navigation task.
> In particular, the specific task of colonoscopy involves navigation of the endoscope up to the visualisation of the caecum landmark, with subsequent withdrawal of the endoscope from the colon.
>
> The more significant mis-classification in this case could be attributed to the template labelling being under-labelled in comparison to the subsequent labelled colons.
> The authors also note the high levels of empirically observed anatomical variability, specifically in the caecum and hepatic flexure area as well as their proximity, which could additionally explain class confusion for between these labels (see Fig. 3A for anatomical description of colon).
> We illustrate different colon anatomical descriptions as well as predicted versus manual labels from the above analysis in Fig. 36 Appendix H.
>
> Despite label noise, our initial results suggest that sufficient anatomical information is perserved for the broad localisation tasks designed in this study (Tab. 2).
> The high navigation success for the noisier caecum label could be explained by the distinct appearance of the landmark in comparison to the flexure landmarks.

---

> ### Author Response · Authors · 2023-08-19
> **Response: Part 3**
>
> > Texture realism: A central question to me is how the realism of the textures is assured? Please provide more details. How exactly is the information taken from previous data sets / surgical papers / tutorials transferred to the shader for texturing? The authors mention that the proposed texturing method could be paired with learning-based texturing. An initial attempt using existing methods on e.g. a subset of the data could be a nice addition.
>
>  We point the reviewer to Appendix D in  the Supplementary Materials 'Procedural Texturing of Meshes' (L600 Supp. Mat. Appendix D).
> In summary, the textures and materials are procedurally generated by creating a directed acyclic graph (DAG) in the Blender Shader Nodes functionality by visually referencing the images of real, intra-operative textures from the mentioned tutorials, surgical papers and previous datasets, and matching the appearance of the rendered DAG in Blender to the reference images by modifying the organ base colour, textures, and patterns.
> In order to assure that the provided materials were realistic, the process was performed iteratively under the supervision of a clinician with surgical training.
> For completeness and to clarify how the textures are obtained, we have expanded on Appendix D 'Procedural Texturing of Meshes' in the Supplementary Materials.
> Specifically, we have created an additional section introducing the texturing process in Blender shader nodes (Appendix D.2 'Introduction to Blender Shader Nodes'), and expand on the original description of the \texttt{SARAMIS} textures in Appendix D.2 'Generation of \texttt{SARAMIS} textures'.
> For completeness, in Appendix D.2, we have added summaries of the Blender shading graph configurations for each texture. Finally, we have also added a reference to the open-source implementation provided in the \texttt{SARAMIS} repository, which contains the full parametrisation of each shader node as a .blend file.
>
> We acknowledge that, in contrast to a data-driven texturing approach as in \cite{Guerrero2022}, the current approach limits the potential realism and variety of potential material textures.
> We have highlighted the significant potential to explore data-driven texturing for surgical applications in the limitations (L319), and consider this to be a useful future extension.
>
> > Dataset release: In the datasheet, the questions “Has an analysis of the potential impact of the dataset and its use on data subjects (e.g., a data protection impact analysis) been conducted?” and “When will the dataset be distributed” have not been answered in the data sheet.
>
> We have updated the respective sections in the Supplementary Materials.
>
> > Experimental design: The design of the experiments leaves room for improvement. For example (1) I doubt that any solid conclusions can be drawn from an experiment with four test cases. Please consider removing the experiment, increasing the number or at least performing uncertainty analyses (e.g. bootstrapping) to analyze the robustness of results.
>
> We acknowledge the limitation of reporting results on 4 test cases.
> We report additional results on 20 cases for human versus RL agent to navigate to different structures in L275.
> We find that the performance over 24 cases changes to $77.8 \pm 13.2$ and $75.3 \pm 15.6$ for the agent and human experiments, respectively, from the originally reported values of $73.5\pm11.1$ and $76.0\pm11.8$.
> There was no statistically significant difference in RL vs human performance over 24 cases (p-value$=0.83$).
> Our conclusions thus remain unchanged from the originally reported conclusions in the paper despite an expanded analysis.
>
>  We have updated the paper as follows: we have removed the original Fig. 4. Panel C from Fig. 4. It has now been replaced with a breakdown of performance of the agent over different data subsets of the test data.
> We replaced the results at L275 with the results over 24 cases.

---

> ### Author Response · Authors · 2023-08-19
> **Response: Part 4**
>
> > (2) Could you explain the purpose of the entire method (which problem that clinicians are facing are you trying to solve? Please provide references) and experiments a bit better from a colonoscopy perspective and then justify the choice of validation metrics accordingly?
>
> Image-based navigation is an important aspect of MIS and endoscopic procedures, and is therefore vital to a number of diagnostic procedures. These procedures require navigation to a desired target or region of interest using the acquired intraoperative image for guidance, usually without external or internal hardware tracking devices. An example of such a procedure is a colonoscopy, which is a minimally-invasive procedure used to diagnose colorectal cancer, precancerous abnormalities, polyps or ulcers [37]. During the colonoscopy, the colon (large intestine) is examined using a flexible endoscope. The endoscope is inserted at the rectum, navigated to the caecum, and slowly retracted whilst scanning the colon for suspicious lesions, which may be removed once detected. Along similar lines, a sigmoidoscopy is an abridged colonoscopy, where the endoscope is navigated to the sigmoid colon, which is a structure that is just proximal to the rectum.
>
> Overall, there is significant evidence that colonoscopies and sigmoidoscopies reduce mortality due to colorectal cancer in patients when polyps are removed [ 39 , 18 , 27, 12 , 4, 3 ]. One study reports up to 53% reduction of mortality rate in patients that have adenomas removed during a colonoscopy compared to control patients that consequently develop colorectal cancer [39]. In order for colonoscopy screening programmes to be effective and mortality to be reduced, clinicians must have a sufficiently high adenoma detection rate (ADR) [19 ], characterised as the percentage of patients where at least one adenoma was detected. ADR is impacted by several factors: 1) Operator experience [ 26 ] , 2) operator specialty [ 32 ], 3) procedure length [5 ], and colonoscopy technique [25 ], amongst others. Given the length and cost of training experienced clinicians, with an estimated minimum of 450 procedures to achieve a sufficient ADR [ 26 ], as well as procedure difficulty [26 ], improving ADR is imperative to reduce patient mortality for CRC.
>
> An effective strategy to address several of these challenges simultaneously is the automation of the procedure. This approach, potentially trained using feedback from experts, can pave the way for standardizing patient outcomes. Furthermore, another compelling approach is the standardization of clinician training with the assistance of RL trained agents. Such agents can offer valuable feedback by comparing the performance of trainees against their own, which might reduce the number of procedures required for a trainee to achieve an adequate ADR for clinical proficiency. Automating a procedure such as the colonoscopy can further have impacts in under-priviledged areas which may lack specialist endoscopic centres or training programmes for general surgerons and/or gastroenterologists. In the limit, partial (or full) automation of such a procedure could also have a dramatic impact on costs and potantially on waiting times.
>
> Whilst in this work we do not directly examine the direct potential of automated polyp detection, analysis and resection due to the aforementioned lack of pathology, we believe this to be a natural and immediate avenue of future research. Paired with sufficiently accurate automated navigation to targets of interest, SARAMIS may be combined with novel work (published subsequent to the submission of this work) on synthesising realistic polyps [ 9] for the extension into automated polyp detection and navigation.
>
> >  (3) What was the rationale for choosing a “biomedical imaging researcher” rather than a physician as a human observer?
> > Line 248: What does S.U.S refer to?
>
> In our study, we opted for a biomedical imaging researcher over a physician as the human observer primarily due to the researcher’s specific expertise in the clinical task at hand. This researcher [S.U.S, 2nd author of the paper] not only had firsthand experience observing videos of actual procedures [ 6], but also possessed a comprehensive understanding of anatomy, making them well-equipped for the task. Moreover, it’s important to note that clinicians were under time constraints, which further justified our decision to involve a researcher who could dedicate the necessary time and focus to the experiment.

---

> ### Author Response · Authors · 2023-08-19
> **Response: Part 5**
>
> >Model overfitting: The authors report a training and test set. Based on which data was the method development performed? Specifically, did the authors avoid model overfitting? Meaning that they finalized the complete algorithm before running analyses on the test set only one single time?
> > Line 244: How were the hyperparameters optimized if the data split was only into training and testing?
>
>  A subset of the train set was used for model development to monitor rewards and episode lengths during training (this should therefore be considered an explicit validation set). We would like to clarify that the training set is actually formed of the set used for model training as well as the set used for model development. In total we have 91 meshes used for model training, 32 used for model development, and 32 reserved for testing. This has now been clarified in the paper in L260: The subset was split into 91 meshes used for model training, 32 for model development and 32 meshes as a hold-out test set. The model development samples were used to monitor rewards for hyper-parameter tuning.
>
> > Overall, this results in “1048, 279, 1200 scans” -> these numbers should appear in Tab. 1 to avoid confusion (and to explain the number of images 2,527, which appears in the text but not in the table).
>
> We have modified Tab. 1 (L156) to include the included scans, and have included the numbers of excluded scans into the text (L164-166).
>
> > Please explain how the “subset of 450 verified CT scans” images was chosen.
>
>  The subset of 450 scans was sampled from the AMOS and Abdomen-1k datasets that had been previously reviewed by a junior annotator, as described in the annotation protocol.
> The subset of scans included those that had and hadn't been flagged for correction.
>
> > The colon as a special case for segmentation and meshing could be separated a bit more clearly both in the annotation description and the details of the mesh generation
>
>  We have made the following changes to Section 3 (L101) 'Dataset Generation'. Firstly, we have removed the sub-section 3.1 for clarity.
> Then, we have separated the colon generation process under it's own heading 'Colon Mesh Generation' (L123).
> We have additionally moved the section 'Mesh Generation' (L145) below to indicate it's separation from the colon subsection.
>
> > -“Failure in the task is defined as the inability to navigate to the structure within 256 steps or collision with the colonic wall” -> please justify the choice of 256
>
> The colon spans over approximately [200x200x200]mm spatially, and the step max step increments are set to 10mm, which, combined with the maximum step number would give a distance of approx >2560mm.
> Running testing for a larger number of steps would mean that it is unlikely that the structure would be found, for cases e.g. where the agent gets stuck in one place.
> Furthermore, during training and validation we observed a mean of around 100 steps for successful navigation attempts, where the maximum number of steps allowed was 1024.
> This shows that selecting 256 steps as the maximum number of steps taken is sufficient to measure navigation performance.
>
> > “A total of 155 colon meshes were selected from the SARAMIS dataset” -> please explain how?
> > Line 236: How were the 155 colon meshes used for the experiments chosen?
>
> We sub-selected the 155 available colons from the TotalSegmentator dataset to form our training and test set.
> Due to our limited computational power, we were unable to train on a larger subset of the \texttt{SARAMIS} dataset.
> However, we have reported the performance on the remaining subjects from the AMOS and Abdomen-1k subsets in Fig. 4. C, replacing the original panel C (L263).
> We find that success varies a maximum of 10\% between locations and sub-datasets, and overall performance is slightly lower.
> This may be explained by differences between dataset distributions, resulting in slightly lower performance on the AMOS and Abdomen-1k test datasets in comparison to the TotalSegmentator held-out test set.
> Additionally, we have updated Table 2 (L288) to report the overall test performance over all test set cases.
>
> > In the datasheet, the strategy for external contributions should be outlined in more detail, e.g. explaining quality control and communication of contributions.
>
>  We have updated the contribution guidelines (L536 Supplementary Materials Appendix B) to highlight how the contribution requests should be communicated  (on the \texttt{SARAMIS} GitHub), and how the data review process may be carried out.
>
> > Line 110: From the main text it’s not clear what it entails to become a “trained anatomical annotator”. Make sure to quickly mention this here and/or refer to the Supplementary Material for more detailed information
>
>  We have included a line referring to the detailed training and annotation protocol provided in the Datasheet in the Supplementary Materials on L117 of the manuscript.

---

> ### Author Response · Authors · 2023-08-19
> **Response: Part 6**
>
> >  Line 122ff: It remains unclear which resolution the final meshes have. Always the resolution of the underlying CT scan?
>
> As the meshes were extracted from CT scans using marching cubes and laplacian mesh decimation, the resolution is variable between meshes.
> To better quantify the resolution for the meshes, we have expanded our analysis of the data in Appendix G 'Mesh Analysis and Resolution' of the Supplementary Materials (L681).
> Principally, we report the additional analyses as follows
> * Number of vertices per mesh: We report the average number of vertices per mesh, split by organ.
>
> * Surface area of meshes: in 1mm$^2$ units, we calculate the mesh surface area as the sum of triangular face areas.
> * Average vertex density over the surface of the mesh: we additionally report the mesh density as vertices per 1mm$^2$ by dividing the total mesh surface area by the number of vertices.
>
> We report the mean number of vertices per mesh, total surface area of meshes, sorted by number of vertices and by surface area, and mean vertex density in Appendix F across the different sub-datasets.
>
> We find that the surface area shows a large level of variation, whilst the number of vertices per organ is more homogenous.
> Fig. 31 Appendix F shows the surface area is not necessarily correlated with size, as the second, third, and fourth largest organs on average across datasets are the pancreas, and two vertebral bodies (L5, and L2), which are comparatively small structures volumetrically compared to, for example, the liver (ranked 11th), which is one of the largest internal organs of the human anatomy.
> However, the number of vertices is more intuitively correlated to organ size, as the small bowel, liver, colon, and gluteus are amongst the organs with most vertices (Fig. 33 Appendix F).
> We find no significant trend in the mean vertex density as reported in Fig. 34 Appendix F, except that the Abdomen-1k is the most homogenous across organs, with the TotalSegmentator the least homogenous.
>
> We recommend that future users of the dataset take care to resample the provided meshes to best suit their use, as this may impact performance in graph-type methods.
>
> > Line 170ff: I find it hard to bring the number of voxels changed into perspective. Wouldn’t it be more interesting to report for which classes changes were necessary?
>
> We report on L178 the organs for which changes were most and least necessary.
> We provide additional analyses to address this comment in Appendix I 'Analysis of Organ Changes per Dataset' in the Supplementary Materials (L725).
> We report the number of total organs changed per dataset in Fig. 37, and the variation in number of changes for organs that were changed within the AMOS and Abdomen-1k datasets in  Fig. 38.
>
> We report significant sources of confusion for the liver included undersegmentation due to observed pathology or miss-classification of vasculature as non-liver tissue.
> Additionally, a large number of ribs were edited due to the significant observed volumes of class confusion.
> Smaller proportions of edits on the face, brain, and cervical vertebrae can be explained by the anonymisation of the CT scans resulting in few instances of these structures with full visibility across the dataset.
>
> > Line 197: Please specify what exactly this action entails. I assume it is translation + rotation of the camera?
>
> That is correct - as the reviewer intuits, the action is a 6-vector split into three translation parameters, and a 3-vector defining the camera look-at direction. We point the reviewer to the implementation provided at the code source for further details.
>
> > Line 228ff: How is the discount factor chosen?
>
> As can be seen in the code provided in the Supplementary Materials (Appendix A ’Data Location and Code’) the discount factor γ was set to 0.99. This was set empirically [ 33 ]. In practice we observed limited impact on performance with varying values of the discount factor.
>
> > Datasheet, line 326: How is the mesh decimation performed?
>
>  We perform a quadric edge collapse using an implementation from MeshLab. The specific function for mesh decimation can be found here: \url{https://github.com/NMontanaBrown/saramis/blob/main/saramis/mesh/mesh_finetuning.py}
> We have updated the Datasheet for Datasets with specific details on the mesh decimation (Appendix B 'Datasheet for Datasets' Supplementary Materials L324)

---

> ### Author Response · Authors · 2023-08-19
> **Response: Part 7**
>
> >  Datasheet, line 334: For someone new to blender it might not be obvious how those shader nodes have been created bsed on images / surgical information. Would be good to elaborate on this a bit more detailed.
>
> We thank the reviewer for this important comment. In order to clarify how the textures are obtained, we have expanded on Appendix D
> 'Procedural Texturing of Meshes' in the Supplementary Materials (L600).
> Specifically, we have created an additional section introducing principles in Blender shader nodes (Appendix D.1 'Introduction to Blender Shader Nodes'), and expand on and separate the original description of the \texttt{SARAMIS} textures in Appendix D.2 'Generation of \texttt{SARAMIS} textures'.
> For completeness, in Appendix D.2, we have added summaries of the Blender shading graph configurations for each textures. Finally, we have also added a reference to the open-source implementation provided in the \texttt{SARAMIS} repository, which contains the full parametrisation of each shader node as a .blend file.
>
> > Datasheet, line 580: How were the parameters b, k, h chosen for the procedural mesh generation and why?
>
> Thank you for this important question. The {b, k, h} parameters represent the base radius, amplitude and frequency of the generated colon meshes, respectively (as in Eqn. 2 near L589 Appendix C Supplementary Materials). Please see Fig. 1 (Supplementary Materials Appendix C ’Procedural Generation of Colon Meshes’ L593) for an illustration of the effects of changing these parameters. These parameters can be chosen by a user after downloading the assets to produce meshes with different possible representations of the colonic mucosa. In our paper, we choose these parameters with guidance from clinicians. We would note, however, that the human colon is a deeply heterogenous structure, with massive inter-individual variability [ 10 , 22 ] Therefore, whilst we’ve selected our parameters coextensively with our clinicians, these parameters can be used to generate a large number of biologically plausible large bowel meshes.
>
> > Minor Comments (typos, referencing)
>
> * References [18] and [19] seem to be malformatted and incomplete. Make sure to provide sufficient information.
> * Reference [16] in Supplementary Material is incomplete. E.g. missing Author last name/ Journal Name / issue / number / year
> * Datasheet, line 399: "excluded.From" missing space.
> * Datasheet, line 333: Abbreviation BSDF has not been introduced.
> * Line 140: Appendix A in Supp. Material does not describe the matching algorithm but the dataset. I assume this should reference Appendix C.
> * Line 1: Abstract: RAMIS abbreviation misplaced before surgery
> * Please use tenses consistently.
> *Please fix inconsistent writing 1,000 vs 1000
>
> We thank the reviewer for their attention to detail; we have updated the mistakes accordingly

---

> ### Author Response · Authors · 2023-08-19
> **Response: Part 8 (Data Accessibility)**
>
> > The data set is well documented in the datasheet and the remaining parts of the Supplementary material. Although the full data set was not released yet, a URL was provided to access exemplary parts of the data set via AWS. However, as mentioned in the “Opportunities for improvement”, I couldn’t find an easy way to access this data without signing up for AWS.
>
> >  IMPORTANT: The full dataset has not been released upon submission but will - according to the authors - be made publically available upon paper acceptance. Unfortunately, I was not able to access the sample data and can thus not fully judge the paper. Furthermore, I recommend allowing public data access without barriers, such as logins.
>
> We acknowledge that using AWS CLI can present some barriers to accessing \texttt{SARAMIS}.
> In order to address this concern, we have made the AWS bucket completely publically available at the following link: \url{https://saramis.s3.eu-north-1.amazonaws.com/}. Items in the bucket may be downloaded by adding the prefix associated to the file to be downloaded, as detailed in the Supplementary Materials Appendix A. 'Data Location And Code'.
> Additionally, to make the dataset more visible, we have added the information to access the data to the \texttt{SARAMIS} repository, which has been added to the abstract of the paper (L11-13).
> Finally, we would like to clarify that the full dataset was made available upon the submission of the paper.

---

> ### Author Response · Authors · 2023-08-19
> **Response: References**
>
> [3] W. Atkin, J. Northover, J. Cuzick, and D. Whynes. Prevention of colorectal cancer by once-only
> sigmoidoscopy. The Lancet, 341(8847):736–740, 1993.
>
> [4] W. Atkin, K. Wooldrage, D. M. Parkin, I. Kralj-Hans, E. MacRae, U. Shah, S. Duffy, and
> A. J. Cross. Long term effects of once-only flexible sigmoidoscopy screening after 17 years
> of follow-up: the uk flexible sigmoidoscopy screening randomised controlled trial. The Lancet,
> 389(10076):1299–1311, 2017.
>
> [5] R. L. Barclay, J. J. Vicari, A. S. Doughty, J. F. Johanson, and R. L. Greenlaw. Colonoscopic
> withdrawal times and adenoma detection during screening colonoscopy. New England Journal of
> Medicine, 355(24):2533–2541, 2006.
>
> [6] H. Borgli, V. Thambawita, P. H. Smedsrud, S. Hicks, D. Jha, S. L. Eskeland, K. R. Randel, K.
> Pogorelov, M. Lux, D. T. D. Nguyen, et al. Hyperkvasir, a comprehensive multi-class image and
> video dataset for gastrointestinal endoscopy. Scientific data, 7(1):283, 2020.
>
> [9] T. Dowrick, L. Chen, J. Ramalhinho, J. González-Bueno Puyal, and M. J. Clarkson. Procedurally
> generated colonoscopy and laparoscopy data for improved model training performance. Under
> Review, 2023
>
> [12] K. Garborg, Ø. Holme, M. Løberg, M. Kalager, H. Adami, and M. Bretthauer. Current status of
> screening for colorectal cancer. Annals of oncology, 24(8):1963–1972, 2013
>
> [14] P. Guerrero, M. Hašan, K. Sunkavalli, R. Mˇech, T. Boubekeur, and N. J. Mitra. Matformer: A
> generative model for procedural materials. arXiv preprint arXiv:2207.01044, 2022.
>
> [18] C. J. Kahi, T. F. Imperiale, B. E. Juliar, and D. K. Rex. Effect of screening colonoscopy on
> colorectal cancer incidence and mortality. Clinical gastroenterology and hepatology, 7(7):770– 775,
> 2009
>
> [19] M. F. Kaminski, J. Regula, E. Kraszewska, M. Polkowski, U. Wojciechowska, J. Didkowska, M.
> Zwierko, M. Rupinski, M. P. Nowacki, and E. Butruk. Quality indicators for colonoscopy and the
> risk of interval cancer. New England journal of medicine, 362(19):1795–1803, 2010
>
> [25] R. H. Lee, R. S. Tang, V. R. Muthusamy, S. B. Ho, N. K. Shah, L. Wetzel, A. S. Bain, E. E.
> Mackintosh, A. M. Paek, A. M. Crissien, et al. Quality of colonoscopy withdrawal technique and
> variability in adenoma detection rates (with videos). Gastrointestinal endoscopy, 74(1):128–134,
> 2011
>
> [26] C. A. Munroe, P. Lee, A. Copland, K. K. Wu, T. Kaltenbach, R. M. Soetikno, and S. Friedland.
> A tandem colonoscopy study of adenoma miss rates during endoscopic training: a venture into
> uncharted territory. Gastrointestinal endoscopy, 75(3):561–567, 2012.
>
> [27] R. Nishihara, K. Wu, P. Lochhead, T. Morikawa, X. Liao, Z. R. Qian, K. Inamura, S. A. Kim,
> A. Kuchiba, M. Yamauchi, et al. Long-term colorectal-cancer incidence and mortality after lower
> endoscopy. New England Journal of Medicine, 369(12):1095–1105, 2013.
>
> [32] D. K. Rex, E. Y. Rahmani, J. H. Haseman, G. T. Lemmel, S. Kaster, and J. S. Buckley. Relative
> sensitivity of colonoscopy and barium enema for detection of colorectal cancer in clinical practice.
> Gastroenterology, 112(1):17–23, 1997
>
> [33] J. Schulman, F. Wolski, P. Dhariwal, A. Radford, and O. Klimov. Proximal policy optimization
> algorithms. arXiv preprint arXiv:1707.06347, 2017
>
> [37] . Williams and R. Teague. Colonoscopy. Gut, 14(12):990–1003, 1973
>
> [39] A. G. Zauber, S. J. Winawer, M. J. O’Brien, I. Lansdorp-Vogelaar, M. van Ballegooijen, B.
> F. Hankey, W. Shi, J. H. Bond, M. Schapiro, J. F. Panish, et al. Colonoscopic polypectomy and
> long-term prevention of colorectal-cancer deaths. New England Journal of Medicine, 366(8):687–696,
> 2012

---

### Official Review · Reviewer_1urX · 2023-07-21
**Comprehensive dataset, Mind-blowing application**

**Rating:** 8
**Confidence:** 4
**Correctness:** submission correct
**Clarity:** well written

**Strengths:**

The team creates the world's largest MIS Dataset. The dataset is very meaningful to the biomedical community, and can be used to improve and evaluate the performance of autonomous navigation tasks. Real world data is hard to get, to tackle this the team builds high quality synethetic data with through computer graphic approaches and human expert validation.
    - The team incorportate existing open-source software such as Blender to create realistic simulation of depth maps, stereo and monocular cameras, and deformation of soft tissue.
    - The team explores one important application of the dataset, which is autonomous navigation tasks, setting a benchmark for future research.

**Additional Feedback:**

what is the future plan for improve texture quality?

**Documentation:**

Large-scale dataset of 3D rendering asset of human abdominal
largest dataset of heterogenous patient-derived meshes. 116018 meshes from 2529 patient, 104 different organs
GPU rendering environment to generate datasets

**Ethics:**

no ethical concerns

**Limitations:**

limitations addressed

**Opportunities For Improvement:**

It uses procedural texturing in a data-driven way to overcome this. However it's not clear how this is done and how the texture is applied to the mesh. It is also unclear that whether different textures as well as the reflectiveness of the material may make a difference in the sampled videos. It would be better to include these details and discuss the efficacy of producing the texture in this way

The dataset does not provide details for physical-based rigid/soft body simulation. These physical properties of different anatomic structure can be important for realisitc simulation environment.

The experiment is only conducted on 155 colons, which is small compared to the size of the whole database. The starting location is randomly set, which may not capture the real scenario as the colonoscopy is performed at some fixed location.


**Relation To Prior Work:**

unique and original

**Summary And Contributions:**

SARAMIS is a large-scale dataset of 3D human abdominal renderings, the most comprehensive collection of patient-derived meshes with over 116,018 meshes from 2,529 patients and 104 different organs. The dataset was created from open-source medical images like CT scans from Abdomen-1k and Amos datasets. The CT scans were segmented into 104 organs using TotalSegmentor, and the segmentation was validated by experts. The process included generating meshes for all organs using the Marching cubes algorithm and surface smoothing, and specific processes for the human bowel.

The dataset supports autonomous navigation tasks, including simulating intraoperative images for colonoscopy navigation tasks. This process forms a partially observable Markov decision process (PO-MDP) with the camera's 6DOF pose as the action & state and the target's visibility and collision as rewards. An experiment with 155 colons to find anatomical landmarks revealed no significant difference between human experts and reinforcement learning (RL) agents in finding targets.

---

> ### Author Response · Authors · 2023-08-19
>
> We thank the reviewer for their review, and address the opportunities for improvement on a point by point basis.
>
> > It uses procedural texturing in a data-driven way to overcome this. However it's not clear how this is done and how the texture is applied to the mesh. It is also unclear that whether different textures as well as the reflectiveness of the material may make a difference in the sampled videos. It would be better to include these details and discuss the efficacy of producing the texture in this way}
>
>  We would like to clarify that the texturing is based on procedural nodes using Blender based on available data of intra-operative appearances of the organs of interest, as outlined in the Supplementary Material (Appendix D L600) and additionally in L151. In order to ensure realism of the textures, the process was conducted under the supervision of a clinician with surgical experience.
>
> We agree with the reviewer that it is likely that significant differences in the textures and general appearance of the assets during rendering will impact the generated synthetic data.
> Whilst various parameters of the procedural shading nodes may be programmatically changed, significant deviation from the proposed and clinician-verified shading configurations could result in non-clinically feasible renders.
> We acknowledge the limitations of this approach, which is both time-consuming and must be performed under clinical supervision, which is not always widely accessible to the wider public.
>
> In order to clarify how the textures are obtained, we have expanded on Appendix D 'Procedural Texturing of Meshes in the Supplementary Materials (L600 Supp. Mat.).
> Specifically, we have created an additional section introducing principles in Blender shader nodes (Appendix D.1 'Introduction to Blender Shader Nodes'), and expand on and separate the original description of the \texttt{SARAMIS} textures in Appendix D.2 'Generation of \texttt{SARAMIS} textures'.
> For completeness, in Appendix D.2, we have added summaries of the Blender shading graph configurations for each texture. Finally, we have also added a reference to the open-source implementation provided in the \texttt{SARAMIS} repository, which contains the full parametrisation of each shader node as a .blend file.
>
> Finally, we have expanded the Limitations section of the paper to address these points (L319).
>
> >The dataset does not provide details for physical-based rigid/soft body simulation. These physical properties of different anatomic structure can be important for realisitc simulation environment.
>
> We thank the reviewer for their valuable insight.
> We acknowledge the importance of physical properties of different types of tissue for the realism of simulation.
> We have compiled various values across the literature and incorporated a table in the \text{SARAMIS} source code for ease of access to future users.
> We detail these changes in Appendix F of the Supplementary Materials (L676).
> Systematic investigation of the soft-body simulation properties is beyond the scope of the current work, however represents an important direction of future research.
>
> > The experiment is only conducted on 155 colons, which is small compared to the size of the whole database. The starting location is randomly set, which may not capture the real scenario as the colonoscopy is performed at some fixed location.
>
> We sub-selected the 155 available colons from the TotalSegmentator dataset to form our training and test set.
> We acknowledge the limited size of the training set, but due to our limited computational power, we were unable to train on a larger subset of the \texttt{SARAMIS} dataset.
> However, we have additionally reported the performance on the remaining subjects from the AMOS and Abdomen-1k subsets in Fig. 4C (L263), as well as updating Tab. 2 (L288) in order to reflect the new overall performance on the complete test set.
> We find that overall performance decreases slightly in comparison to the original results - this could be attributed to slight test set distribution shift from the new test datasets (AMOS and Abdomen-1k).
> This is supported by the fact that general test performance is higher on the TotalSegmentator subset.
>
> The combination of prior and new results show that despite training on the subset of available data, our model is able to generalise well to unseen test cases, with good navigation performance on the reported navigation targets.

---

> > ### Comment · Reviewer_1urX · 2023-08-20
> >
> > Thanks for the additional information provided. I enjoy your work and keep my rating as a clear accept.

---

> > ### Comment · Reviewer_zAJD · 2023-08-25
> > **Strong rebuttal**
> >
> > The authors changed the paper and supplementary material substantially to clarify and detail various aspects of the dataset. The added analyses (eg 20 new test scenarios, added analysis on the labeling) further support the initial claims of the paper. The data access issue has also been resolved although I was not able to carefully check the data given the short amount of time.
> >
> > The reinforcement learning (RL) experiment comparing human vs RL is now performed using 24 instead of 4 test scenarios. Although this increases the conclusive value of the experiment, I'm not convinced that this comparison using only one human operator is sufficient to draw robust conclusions. Maybe if phrased more like an example application with initial results it would better suited for this paper. Furthermore, I would encourage the authors to add all the clarifications to the manuscript and double-check the numbers in Tab. 1.
> >
> > Overall, the authors did a great job revising the manuscript such that I will increase my score.

---

### Official Review · Reviewer_PbKg · 2023-07-30
**Review Paper 932**

**Rating:** 2
**Confidence:** 5
**Correctness:** See "Opportunities For Improvement"

**Strengths:**

The idea of having a simulation for autonomous navigation in colonoscopy is interesting. The paper is quite straightforward to understand.

**Additional Feedback:**

Overall, I think the paper is interesting. However, there are several points that should be improved before publication:
- The comparison with related work.
- The evaluation of the reconstruction method to check the quality of the final data, and quantitative results/metric.
- The RL part is trivial and not novel.

**Clarity:**

Overall, it is easy to follow the paper but a lot of important discussion/justification/comparison are missing.

**Documentation:**

Not clear. Only sample data are provided.

**Ethics:**

No problem with ethics. However, the authors re-used the data from others, which raises concerns about the license.

**Limitations:**

See "Opportunities For Improvement"

Edit: After the rebuttal, I decreased my score as the main problem of the paper remains, no code documentation and public data are available for review. The tasks and claims are unclear.

**Opportunities For Improvement:**

My main concerns are:
- The contribution of the dataset paper is limited (clarify below)
- The authors did not collect new data, but re-used 3 existing datasets.
- The process of generating the dataset/simulation files (Section 3) seems ad-hoc and lacks of novelty. Only existing works are used without any novel contribution or deep justification.
- The authors do not provide any quantitative results to demonstrate the quality of the reconstructed simulation. The simulation files are reconstructed by algorithms, and only be manually checked by clinicians in the end, so I believe there would be non-perfect cases, especially the authors claims that the proposed dataset is large-scale.
- The comparison with related work is not clear and missing. After reading the paper, it's not obvious to me why we need a new dataset like this. Do we have existing similar datasets? If yes, what is the comparison between this one and existing ones? E.g, [1] "Colonoscopy 3D Video Dataset with Paired Depth from 2D-3D Registration" is a recent dataset that I believe is very related to this work (although [1] dataset is collected manually).
- The "Autonomous Navigation with Colonoscopy" part seems trivial. The RL algorithm is well-known and can be applied to any dataset, not only the one proposed by the authors.
- The paper raised a lot of tasks that can be done using the proposed dataset, however, it's not clear (in the related work) what are the other datasets that can do (one or more) similar tasks. No evaluation metrics or clear quantitative results are provided.
- The dataset is not publicly available yet, only sample data are provided to reviewers, with very poor documents and no source code (for verifying the reconstruction, or RL part).

**Relation To Prior Work:**

Not clear. Missing many related works and detailed comparisons.

**Summary And Contributions:**

The paper presents a method to convert open-source CT scans to simulation files for MIS applications. The authors also show an RL algorithm on the reconstructed colonoscopy.

---

> ### Author Response · Authors · 2023-08-19
> **Response: Part 1 (Scope of the Work)**
>
> > The authors did not collect new data, but re-used 3 existing datasets.
>
> We cite the specific scope of the NeurIPS Datasets and Benchmarks Track:
>
> >> SCOPE. This track welcomes all work on data-centric machine learning research (DMLR), ... [including] but ... not limited to:
> >> * New datasets, or carefully and thoughtfully designed **(collections of) datasets based on previously available data.**
> >> * Benchmarks on new or existing datasets, as well as benchmarking tools."
>
> First, we would like to underscore that deriving new datasets from previously available data is within the scope of the track, and therefore should not be considered a limitation of the submission or held against our work. Second, we would note that our work is not a simple concatenation of three open-source datasets, instead, we accrue data for a given modality (in this case CT scans of the thoracic, abdominal, and pelvic viscera), and focus our contributions on multi-organ segmentation, the creation of new labelling for all assets, data curation, and camera path generation. These tasks enable simulation-based experimentation not possible with CT scans, as we demonstrate in the case of colonoscopy. Finally, it should be noted that this work dramatically increases the size of the publicly available consistently labelled datasets, and thereby improves data accessibility for an important domain.

---

> ### Author Response · Authors · 2023-08-19
> **Response: Part 2 (Justification of Pipeline)**
>
> > The process of generating the dataset/simulation files (subsection 3) seems ad-hoc and lacks of novelty. Only existing works are used without any novel contribution or deep justification.
>
> Whilst the constituent elements of our work are based on previous literature, it should be noted that we develop a carefully reasoned and well justified pipeline for creating the simulation assets, and do not believe our choices to be ad-hoc or without deep justification.
>
> The conversion of meshing 3D volumetric files is a well established practice in computer-aided design [1]. Specifically, the meshing of 3D volumes in medical imaging is a wide practice with applications across many anatomies and applications. In addition to the prior work cited in the related work section of the main paper, we additionally highlight alternate applications such as haemodynamic modelling of aortas [35], deformation simulation of livers [ 30 , 29 , 23 ], surgical training using VR [ 21 , 20 ], or modelling of bite in patient-derived jaws [13]. This practice has led to highly cited software toolkit papers which facilitate the loading, visualisation, and processing of common formats to this aim, including 3DSlicer [31] and Medical Imaging Toolkit [38].
>
> Typically, a small number (between 1-20) scans may be manually loaded, edited and finally meshed in 3DSlicer for further use (see Related Work). We anecdotally report that the meshing of a single scan manually may take between 5-60 minutes using the 3DSlicer GUI. A significant difference between our pipeline and previously reported pipelines is the highly decreased level of manual interaction required to achieve the meshing step. In contrast to previously reported pipelines[35 , 30 , 29 , 23, 21, 20 , 13 ], we pair VTK with 3DSlicer through python scripts in order to remove the GUI interaction with 3DSlicer, thus enabling us to batch process three orders of magnitude more data than previously reported. We strongly argue that this portion of the pipeline is critical to the generation of SARAMIS assets. Additionally, by detailing our pipeline in the manuscript and sharing the source code, software stack, and processing scripts, we may facilitate future batch processing of large amounts of medical data in the community.
>
> Similarly to meshing from 3D voxel representation, tetrahedral volume generation from volumetric data formats has been reported in the literature over the past three decades (for example, [ 11 , 15 ,36 , 34 ]). Tetrahedral meshing has important implications for the field of MIS+RAMIS as detailed within the related work section; it is critical to understand the plastic and elastic behaviour of organs for realistic simulation, which is typically achieved through finite-element modelling. We chose to perform our tetrahedralisation through fTetWild [ 17 ] due to it’s superior performance versus more classic methods in terms of speed, final mesh quality and recovery, and memory requirements. Given the large volumes of data processed for SARAMIS, these efficiencies significantly sped up the production of volumes.
>
> With regards to our texturing approach, we build on common techniques in computer-generated animation [ 7 , 24 ]. In spite of the field’s longevity and continuous advancements reported in the general computer-graphics (SIGGRAPH, CVPR) venues, significantly fewer works have examined this approach for medical imaging. The closest work to SARAMIS is the recently published [ 9 ], yet SARAMIS provides an order of magnitude larger number of textures associated to three orders of magnitude larger number of anatomical targets. In this case, whilst the techniques are significantly well-established in the literature, we argue the application of said techniques at scale for MIS+RAMIS constitutes the novelty of this portion of the pipeline.
>
> In summary, we strongly rebut the possibility that the outlined data-generation pipeline is ad-hoc and not deeply justified in the context and best practices outlined in the literature. We would also like to take this opportunity to highlight that to the best of the authors’ knowledge, the complete pipeline is the first application of combined methods to a large scale medical imaging dataset in the literature.

---

> ### Author Response · Authors · 2023-08-19
> **Response: Part 3**
>
> > The authors do not provide any quantitative results to demonstrate the quality of the reconstructed simulation. The simulation files are reconstructed by algorithms, and only be manually checked by clinicians in the end, so I believe there would be non-perfect cases, especially the authors claims that the proposed dataset is large-scale.
>
> Thank you for this important point. We agree with the reviewer that there may very well be imperfect cases, however, would argue that this is a likely occurrence in any large-scale dataset, including for ostensibly simpler tasks such as number-labelling in MNIST, CIFAR-10, or ImageNet [28].
>
> To the broader point about checking outputs with clinicians: we would like to briefly reiterate and summarize our quality assurance from the main manuscript. We take randomized, unbiased samples from our large-scale dataset for manual assessment of segmentations. Where appropriate, we perform per-pixel-level corrections following assessment by a team of specialist, expert radiologists at a central university teaching hospital. We note that manual assessments by clinicians, and in particular by expert radiologists, is widely considered to be the gold standard for visceral organ segmentation [16, 8, 2].
>
> In addition, all scans were manually assessed on a per-slice basis for all volumes by a team of postgraduate researchers with a specialism in medical imaging and medical image computing. All scan assessments were conducted under the guidance of at least one clinician. For all challenging cases, for example in the cases of fused/missing vertebrae, organ transplantation, unclear pathology (including ascites and/or solid-organ tumours etc.), clinician input was sought to manually correct/validate the segmentations and associated labels. Whilst imperfections might of course still exist, we believe that our large efforts help ensure a very high-quality, well-validated dataset.
>
> > After reading the paper, it's not obvious to me why we need a new dataset like this. Do we have existing similar datasets? If yes, what is the comparison between this one and existing ones? E.g, [1] "Colonoscopy 3D Video Dataset with Paired Depth from 2D-3D Registration" is a recent dataset that I believe is very related to this work (although [1] dataset is collected manually).
>
> As highlighted in our discussion of previous work (L59), the availability of 3D rendering assets for minimally-invasive surgery is highly limited, comprising of sets of data in the order of magnitude of up to 10 distinct 3D models, and limited to specific types of anatomy (for example, the liver or the colon). SARAMIS tackles both these issues, by not only providing a much larger set of potential anatomical targets (over two orders of magnitude in scope versus currently available 3D assets), but additionally provides a much larger number of assets.
>
> Furthermore, we would like to specifically point out that, unfortunately, the 3D assets for the CV3D dataset referred to by this author are not available to download as of the date of this review. Additionally, this dataset comprises of a singular colon 3D mesh. Therefore, whilst similar in scope to the data proposed in this paper, especially in the context of experiments explored for autonomous navigation, the CV3D dataset cannot be justly compared to the broader range of assets that the SARAMIS paper proposes, covering over 104+ classes and over 3x larger volume of total meshes.
>
> > The "Autonomous Navigation with Colonoscopy" part seems trivial. The RL algorithm is well-known and can be applied to any dataset, not only the one proposed by the authors.
>
> We acknowledge the established nature of the RL algorithm employed for the experiments part of the paper. However, we would like to highlight that this paper reports the first application of this algorithm to the application of colonoscopy. We believe this to be both an important finding, and also to act as a good validation of our assets. Additionally, we have introduced a novel reward and set of constraints for the specific application, which has enabled the training of the RL algorithm for the purpose of colonoscopy.

---

> ### Author Response · Authors · 2023-08-19
> **Response: Part 4**
>
> > The paper raised a lot of tasks that can be done using the proposed dataset, however, it's not clear (in the related work) what are the other datasets that can do (one or more) similar tasks.
>
> We point the reviewer to L71-77, which reviews related works which may be used for the proposed tasks, and their limitations in comparison to \texttt{SARAMIS}. Specifically, from L73:
>
> ''However, current work either does not release 3D assets to simulate or manipulate scenes [51] or uses non-open-source frameworks [57 , 16] and can be limited in terms of application [ 57, 69 , 16 , 67, 66].
> Existing work is further limited in the number of anatomical variations of 3D assets due to the use of a small cohort to produce the datasets
> [57, 33, 16, 64, 66].''
>
> We believe the above addresses the principal limitations of the prior works as compared to \texttt{SARAMIS}.
>
> > No evaluation metrics or clear quantitative results are provided.
>
> We would like to point the reviewer to the quantitative results and metrics provided within the paper:
>
>    * We provide \texttt{SARAMIS} labelling statistics, describing the number of assets in SARAMIS, results from the data analysis section of the paper between L158-169, as well as dataset statistics in Tab. 1 (L156).
>    * We report evaluation metrics for the navigation success experiment on L277, as well as quantitative results of said experiment in Tab. 2 (L288).
>    * Additionally, we compare the performance of a human versus the agent over 24 cases in L275. We additionally report that no statistical significance was found between human and RL agents (L276).
>
> Additionally, in line with other reviewer comments, we have added further analyses of \texttt{SARAMIS} as follows:
>
>    * We expanded our analysis of human versus RL performance from 4 to 24 cases, and provided a sub-analysis of performance separated by sub-dataset in Fig 4.C (c. L263).
>
>    * We provide an analysis of mesh quality as indicated by number of vertices, mesh surface area, and mesh density in Appendix G (L681 Supp. Mat.). We report several figures containing the mean surface area, number of vertices, and mesh density (Figs. 31-34).
>     * We provide an additional analysis of  the labelling accuracy as a result of the registration pipeline reported in L251-259). We label 30 manual colons, and compare the accuracy of obtained registered labels from template registration.
>     The results are reported in Fig. 35 (Supp. Mat. Appendix H L711).
>     * Finally, we report an additional analysis of the number of organs changed, and a table of all absolute changes with IQRs in Appendix I (L725 onwards).
>
> > The dataset is not publicly available yet, only sample data are provided to reviewers, with very poor documents and no source code (for verifying the reconstruction, or RL part).
>
> Upon submission the full dataset was provided, accessible at AWS via the AWS CLI.
> Full instructions were provided in the Supplementary Materials to access the data using AWS CLI.
> Additionally, the source code to process the data was provided in L14 of the Supplementary Material.
>
> We recognise these details were not directly provided within the paper submitted.
> Therefore, in line with other comments from the reviewers, we have updated the paper (L11-13) to include a link to the \texttt{SARAMIS} repository: \url{https://github.com/NMontanaBrown/saramis/}, which contains the source code and dataset download location in the README.
>
> > "Relation to prior work: Not clear. Missing many related works and detailed comparisons.
> > The comparison with related work is not clear and missing.
>
> Could the reviewer please clarify which related works are missing so that we can improve our paper?
>
> > "Ethics": No problem with ethics. However, the authors re-used the data from others, which raises concerns about the license.
>
> We have thoroughly discussed the license implications for the \texttt{SARAMIS} paper in the Datasheet for Datasets provided in the Supplementary Materials (Appendix B, L483 onwards).
> However, we understand that these details are not clarified in the paper.
> Therefore, we have added the licensing conditions of \texttt{SARAMIS} to the abstract of the paper (L13).

---

> ### Author Response · Authors · 2023-08-19
> **Response: References for response, Pt. 1**
>
> [1] A. Agathos, I. Pratikakis, S. Perantonis, N. Sapidis, and P. Azariadis. 3d mesh segmentation
> methodologies for cad applications. Computer-Aided Design and Applications, 4(6):827–841, 2007.
>
> [2] M. Antonelli, A. Reinke, S. Bakas, K. Farahani, A. Kopp-Schneider, B. A. Landman, G. Litjens,
> B. Menze, O. Ronneberger, R. M. Summers, et al. The medical segmentation decathlon. Nature communications, 13(1):4128, 2022
>
> [7] E. Catmull. A system for computer generated movies. In Proceedings of the ACM annual
> conference-Volume 1, pages 422–431, 1972.
>
> [8] M. S. Choi, B. S. Choi, S. Y. Chung, N. Kim, J. Chun, Y. B. Kim, J. S. Chang, and J. S. Kim.
> Clinical evaluation of atlas-and deep learning-based automatic segmentation of multiple organs
> and clinical target volumes for breast cancer. Radiotherapy and Oncology, 153:139–145, 2020.
>
> [9] T. Dowrick, L. Chen, J. Ramalhinho, J. González-Bueno Puyal, and M. J. Clarkson. Procedurally
> generated colonoscopy and laparoscopy data for improved model training performance. Under Review, 2023.
>
> [11] R. Gao and J. Peters. Improving hexahedral-fem-based plasticity in surgery simulation. In
> Medical Image Computing and Computer Assisted Intervention–MICCAI 2021: 24th International Conference, Strasbourg, France, September 27–October 1, 2021, Proceedings, Part IV
> 24, pages 571–580. Springer, 2021.
>
> [13] T. Gholamalizadeh, F. Moshfeghifar, Z. Ferguson, T. Schneider, D. Panozzo, S. Darkner,
> M. Makaremi, F. Chan, P. L. Søndergaard, and K. Erleben. Open-full-jaw: An open-access
> dataset and pipeline for finite element models of human jaw. Computer Methods and Programs
> in Biomedicine, 224:107009, 2022
>
> [15] S. Hang. Tetgen, a delaunay-based quality tetrahedral mesh generator. ACM Trans. Math. Softw, 41(2):11, 2015
>
> [16] M. H. Hesamian, W. Jia, X. He, and P. Kennedy. Deep learning techniques for medical image
> segmentation: achievements and challenges. Journal of digital imaging, 32:582–596, 2019.
>
> [17] Y. Hu, T. Schneider, B. Wang, D. Zorin, and D. Panozzo. Fast tetrahedral meshing in the wild.
> ACM Trans. Graph., 39(4), July 2020.
>
> [20] H. Kenngott, J. Wünscher, M. Wagner, A. Preukschas, A. Wekerle, P. Neher, S. Suwelack,
> S. Speidel, F. Nickel, D. Oladokun, et al. Openhelp (heidelberg laparoscopy phantom): develop-
> ment of an open-source surgical evaluation and training tool. Surgical endoscopy, 29:3338–3347, 2015.
>
> [21] H. G. Kenngott, M. Pfeiffer, A. A. Preukschas, L. Bettscheider, P. A. Wise, M. Wagner,
> S. Speidel, M. Huber, F. Nickel, A. Mehrabi, et al. Imhotep: cross-professional evaluation of
> a three-dimensional virtual reality system for interactive surgical operation planning, tumor
> board discussion and immersive training for complex liver surgery in a head-mounted display.
> Surgical Endoscopy, pages 1–9, 2021.
>
> [23] B. Koo, E. Özgür, B. Le Roy, E. Buc, and A. Bartoli. Deformable registration of a preoperative
> 3d liver volume to a laparoscopy image using contour and shading cues. In International
> conference on medical image computing and computer-assisted intervention, pages 326–334.
> Springer, 2017
>
> [24] J. Lasseter. Principles of traditional animation applied to 3d computer animation. In Seminal
> graphics: pioneering efforts that shaped the field, pages 263–272. 1998.
>
> [28] C. G. Northcutt, A. Athalye, and J. Mueller. Pervasive label errors in test sets destabilize
> machine learning benchmarks. In Proceedings of the 35th Conference on Neural Information
> Processing Systems Track on Datasets and Benchmarks, December 2021.
>
> [29] E. Özgür, B. Koo, B. Le Roy, E. Buc, and A. Bartoli. Preoperative liver registration for
> augmented monocular laparoscopy using backward–forward biomechanical simulation. Inter-
> national journal of computer assisted radiology and surgery, 13:1629–1640, 2018
>
> [30] M. Pfeiffer, C. Riediger, S. Leger, J.-P. Kühn, D. Seppelt, R.-T. Hoffmann, J. Weitz, and S. Spei-
> del. Non-rigid volume to surface registration using a data-driven biomechanical model. In
> Medical Image Computing and Computer Assisted Intervention–MICCAI 2020: 23rd Interna-
> tional Conference, Lima, Peru, October 4–8, 2020, Proceedings, Part IV 23, pages 724–734.
> Springer, 2020
>
> [31] S. Pieper, M. Halle, and R. Kikinis. 3d slicer. In 2004 2nd IEEE international symposium on
> biomedical imaging: nano to macro (IEEE Cat No. 04EX821), pages 632–635.
>
> [34] H. Si and K. Gärtner. Meshing piecewise linear complexes by constrained delaunay tetrahedral-
> izations. In Proceedings of the 14th international meshing roundtable, pages 147–163. Springer,
> 2005

---

> ### Author Response · Authors · 2023-08-19
> **Response: Reference for Response Pt. 2**
>
> [35] C. Stokes, M. Bonfanti, Z. Li, J. Xiong, D. Chen, S. Balabani, and V. Díaz-Zuccarini. A novel
> mri-based data fusion methodology for efficient, personalised, compliant simulations of aortic
> haemodynamics. Journal of Biomechanics, 129:110793, 2021.
>
> [36] N. P. Weatherill and O. Hassan. Efficient three-dimensional delaunay triangulation with
> automatic point creation and imposed boundary constraints. International journal for numerical
> methods in engineering, 37(12):2005–2039, 1994.
>
> [38] I. Wolf, M. Vetter, I. Wegner, T. Böttger, M. Nolden, M. Schöbinger, M. Hastenteufel, T. Kunert,
> and H.-P. Meinzer. The medical imaging interaction toolkit. Medical image analysis, 9(6):594–
> 604, 2005

---

> ### Comment · Area_Chair_WtG5 · 2023-08-29
> **Confirming/updating the rating for submission 932 / SARAMIS**
>
> Dear reviewer PbKg,
>
> If you have a moment, could you please respond to the author comments on submission 932 / SARAMIS, and if you believe this merits an updated rating/score, please also do so (or confirm the current rating as is).
>
> Thank you very much in advance. With best regards,
>
> Jochen Weber (Area Chair WtG5)

---

> > ### Comment · Reviewer_PbKg · 2023-08-29
> >
> > I thank the authors for the rebuttal. After reading the rebuttal and the revised version, I see the paper still has several problems:
> > - The labeling progress of the data utilized a list of algorithms, but the results of this were never qualitatively evaluated. I do not agree with the authors that just using some CVPR/SIGGRAPH papers is sufficient enough. The authors did claim that their pipeline can be useful for future work, so I would like to see the accuracy of this automatic pipeline.
> > - The related work and comparison with other datasets are not clear. Indeed, the authors claim the dataset to be large-scale, multi-organs, multi-tasks, but it never is clear to me which tasks, which organs, or how large scale (with respect to a particular organ/task) compared to the existing datasets. The comparison should be made clear as a table, or chart, instead of just pure text argument.
> > - The RL part is trivial. The proposed task is not studied in-depth, no comparison with other algorithms/baseline, except humans. No physics/contact/safety are discussed. I also do not agree with the authors that this work is the first one to solve autonomous navigation with colonoscopy. There are already several works on this task.
> > - Scope of the paper: The author claimed that the dataset is for Minimally Invasive Surgery (MIS), which I believe to be vague. MIS has several tasks, and it is not clear what and how this dataset can be utilized for each task. The instruction/example code for each task should be available to the public.
> > - Writing problems: There are too many "text-based", or "citations-based" arguments,  while no qualitative results. I also do not think that writing a rebuttal with 38 citations is a good idea. The authors should find a better way to extract the information from all the related papers rather than asking people to go to all these citations. Several bulk citations are used in the paper with no clear meaning.
> > - Coding problems: As of 30/8/2023, when I check the code: https://github.com/NMontanaBrown/saramis: - no document on how to run the labeling part (the README stops at "Installing") - no document to run/benchmark the RL part. Overall, no document for any piece of the code. I can find the RL implementation but can't run the code to reproduce the results as shown in the Supplementary Material.
> > - The documentation: https://saramis.readthedocs.io/ is still empty.
> > - No available data for the public: The authors reply to the question "When will the dataset be distributed?" as "The dataset is made publically available at the SARAMIS repository, with source code and links to download the data: https://github.com/NMontanaBrown/saramis." However, the GitHub page only shows provisional data with no instructions on how to get it for the public. It is a pity that this is the dataset paper but none of the reviewers actually know what the dataset looks like.
> >
> > Overall, I see this work has the potential. However, with the current writing/evaluation and the limited code/data available for the public at the moment, I do not see how it fits the current literature and how it will be useful for the public with very limited data/code.

---

> > > ### Author Response · Authors · 2023-08-30
> > >
> > > > The labeling progress of the data utilized a list of algorithms, but the results of this were never qualitatively evaluated. I do not agree with the authors that just using some CVPR/SIGGRAPH papers is sufficient enough. The authors did claim that their pipeline can be useful for future work, so I would like to see the accuracy of this automatic pipeline.
> > >
> > > > Writing problems: There are too many "text-based", or "citations-based" arguments, while no qualitative results. I also do not think that writing a rebuttal with 38 citations is a good idea. The authors should find a better way to extract the information from all the related papers rather than asking people to go to all these citations. Several bulk citations are used in the paper with no clear meaning.
> > >
> > >
> > >  We thank the reviewer for their response. We would like to note that we have in fact qualitatively evaluated all scans. The scans were evaluated a posteriori after running the pipeline. By this, we mean that we provided unbiased samples from the pipeline to expert radiologists to assess the accuracy of the new labels and made amendments where necessary. Furthermore, the scans were reviewed by at least one clinician, and again where errors were found, they were manually amended by our team. We point to our [Response: Part 3] to this reviewer for a more thorough discussion of this point.
> > >
> > > We want to additionally note that expert radiologist assessment of anatomical segmentations is the current gold-standard method of qualitatively assessing this type of labelling. We provide several relevant citations to this effect, also in our original response [Response: Part 3]. It is entirely possible that the reviewer has a different metric or methodology in mind when talking of 'qualitative assessment', or 'accuracy of the pipeline'. If this is the case, we would ask that the reviewer clarify explicitly what they mean by this, so that we may further discuss and/or remedy this point. But again, we want to highlight that we have put a large amount of effort into ensuring well labelled, highly accurate anatomical assets, which were indeed carefully assessed by our expert clinicians, who are co-authors of this work.
> > >
> > > Briefly on the point of using citations here in our rebuttal - in the first phase of reviews, we felt it was very important to answer the question of whether our pipeline was 'ad-hoc', or without 'deep justification'. As such, we wanted to provide a clear, but also comprehensive response to the reviewer of precisely why and how we came to make our design decisions, and to justify all constituent parts of our pipeline. Whilst we acknowledge that this makes for slightly lengthier reading, we believe our referencing and responses to be comprehensive and relevant and stick by our decision to do this.
> > >
> > > Finally, on the point of making 'bulk citations in the paper with no clear meaning' - it would be useful to have an example of this in the paper, so that we may assess this ourselves, and either defend each citation, or adjust them as necessary.
> > >
> > > > The related work and comparison with other datasets are not clear. Indeed, the authors claim the dataset to be large-scale, multi-organs, multi-tasks, but it never is clear to me which tasks, which organs, or how large scale (with respect to a particular organ/task) compared to the existing datasets. The comparison should be made clear as a table, or chart, instead of just pure text argument.
> > >
> > > Thank you for this comment. However, we disagree on this point. Please note Section 2 - Open Source MIS Datasets for a detailed comparisons with the literature, and with other available datasets. The dataset is indeed large-scale and multi-organ. Please see Section 4.1 - 3D Dataset Generation for a discussion of available size.
> > > Please also note Fig 3. in the paper, which outlines the dataset number of meshes split by organ type, and please note the list of organs produced for this dataset in the paper's Datasheet for Datasets (Supplementary Materials, Appendix B, Section ''Composition'', L164).
> > > Finally, please note the listed potential applications of the SARAMIS dataset in the Datasheet for Datasets (Supplementary Materials, Appendix B, Section ''Uses'', L456).
> > > We believe this represents the largest set of simulation assets of the visceral anatomy, and strongly believe it would provide a myriad of uses going forward.

---

> > > ### Author Response · Authors · 2023-08-30
> > >
> > > >  Scope of the paper: The author claimed that the dataset is for Minimally Invasive Surgery (MIS), which I believe to be vague. MIS has several tasks, and it is not clear what and how this dataset can be utilized for each task. The instruction/example code for each task should be available to the public.
> > >
> > > > Coding problems: As of 30/8/2023, when I check the code: https://github.com/NMontanaBrown/saramis: - no document on how to run the labeling part (the README stops at "Installing") - no document to run/benchmark the RL part. Overall, no document for any piece of the code. I can find the RL implementation but can't run the code to reproduce the results as shown in the Supplementary Material.
> > >
> > > > The documentation: https://saramis.readthedocs.io/ is still empty.
> > >
> > > Thank you for these important notes. We agree that minimally invasive surgery (as well as endoscopic procedures), represents a large body of possible tasks (there are many hundreds of laparoscopic surgical procedures which are possible on the abdominal organs alone, for example). We therefore select a task which we believe to be well motivated and important for inclusion in this work (in this case, the task of colonoscopy), and provide a complete `case-study' of this as an example task. We have now included documentation on how to run the RL experiments directly on the GitHub page for the project, at https://github.com/NMontanaBrown/saramis/tree/main/docs/RL .
> > >
> > > With regards to documentation to produce the labelling, we have additionally added all relevant instructions to run the pipeline to the GitHub page for the project, under https://github.com/NMontanaBrown/saramis/tree/main/docs/labelling.
> > > Finally, we have removed external documentation links to ensure that all relevant instructions are clearly documented in the project's GitHub. We hope that this makes the code significantly more accessible, and allows for the easy running of experiments.
> > >
> > > > No available data for the public: The authors reply to the question "When will the dataset be distributed?" as "The dataset is made publically available at the SARAMIS repository, with source code and links to download the data: https://github.com/NMontanaBrown/saramis." However, the GitHub page only shows provisional data with no instructions on how to get it for the public. It is a pity that this is the dataset paper but none of the reviewers actually know what the dataset looks like.
> > >
> > > This is indeed the case. The data is made publicly available, in full, as previously explained. However, we want to ensure this process is not obfuscated in any manner to allow for easy download of the assets. Therefore, in the interests of complete clarity, we would like to give an example of how to download the data.
> > >
> > > On the GitHub page, there is a data sub-heading, at this URL: https://github.com/NMontanaBrown/saramis. Under the data subheading is a link to the AWS bucket: https://saramis.s3.eu-north-1.amazonaws.com/
> > >
> > > Clicking on the link above will open an XML file which provides the relevant options for asset download. For example, supposing you want to download the Abdomen-1k data. In that case, you would simply append the relevant option to the URL: https://saramis.s3.eu-north-1.amazonaws.com/abdomen.tar.gz
> > >
> > > Clicking on the link above will trigger a download of all relevant Abdomen-1k derived assets (1.1 TB), for example. To make this process as clear as possible, we now also provide all relevant stable links to download any assets of choice directly.
> > > We include them here for reference:
> > >
> > > https://saramis.s3.eu-north-1.amazonaws.com/abdomen.tar.gz
> > >
> > > https://saramis.s3.eu-north-1.amazonaws.com/amos.tar.gz
> > >
> > > https://saramis.s3.eu-north-1.amazonaws.com/total.tar.gz
> > >
> > > We have additionally included the metadata and RL experiment data separately, for ease of reproducibility.

---

> > > ### Author Response · Authors · 2023-08-30
> > > **Provision of Tabular Comparison of Existing Datasets In Manuscript**
> > >
> > > > The related work and comparison with other datasets are not clear. Indeed, the authors claim the dataset to be large-scale, multi-organs, multi-tasks, but it never is clear to me which tasks, which organs, or how large scale (with respect to a particular organ/task) compared to the existing datasets. The comparison should be made clear as a table, or chart, instead of just pure text argument.
> > >
> > > We have provided an additional table in the manuscript (Section 2.1  Related Work - Open Source MIS Datasets, Table 1, L84) which compares existing 3D datasets for simulation in MIS+RAMIS to SARAMIS in terms of number of assets, number of patients, number of organs, and availability to the public. Due to manuscript length constraints, we have moved Algorithm 1 to the Supplementary Materials (Appendix J, "Reinforcement Learning Training Algorithm").

---

### Official Review · Reviewer_Mims · 2023-08-01
**A very useful large medical dataset of human abdominal anatomy for minimal-invasive surgery simulation.**

**Rating:** 7
**Confidence:** 4
**Correctness:** The proposed methodology for dataset …
**Clarity:** The paper is well written and structu…

**Strengths:**


- The paper introduces large labeled medical dataset. Rather than introducing new data, the paper sources its data from multiple existing datasets and focuses its contribution on segmentation, labeling, curation and camera path generation.
This increases the dataset size and simplifies the use of multiple datasets.

- Efforts like this paper are much needed to counter the notorious lack of available medical data for tackling various machine learning and computer vision problems. Highly realistic synthetic data is a great and lacking step to close the large training gap in the medical image domain.

**Additional Feedback:**

Minor comments:

- Please number all equations (e.g. rewards, policy), even if you do not reference an equation, others might want to so.
- L132: categorized -> categorised  (AE vs. BE consistency)

**Documentation:**


There is a detailed documentation in the supplementary material and download instructions to an AWS cloud bucket.
It would be nice if this information could be copied to a dedicated website which makes this information searchable/findable/linkable in the internet.


**Ethics:**


Since this dataset contains human data, ethics considerations are important.
Importantly, the data is anonymous and since this datasets is sourced by existing other public datasets, it inherits their properties regarding ethics and others (e.g. privacy, IP, availability)
In sum, I do not see any severe ethical issues.


**Limitations:**


Limitations are briefly discussed the last paragraph of the conclusion. I recommend extending this paragraph (see above) and make it more prominently visible with a paragraph header and separation from future work and concluding statements.
No societal impact is discussed, but I do not expect negative impact from this work.



**Opportunities For Improvement:**


- From the paper it is unclear how the textures are generated, only a side-node in the conclusion mentions the procedural approach.
To make the paper better self-contained, this should be mentioned earlier with an explicit reference to the suppl. material where further details are provided.


- The current evaluation of generated navigation paths is rather weak: Fig. 4 shows only 4 cherry-picked case examples.
Although the reward function penalizes wall intersections, this does by no means guarantee they will not occur in the generated data.
A statement or evaluation about the eventual number of generated intersections would be a critical addition to paper since they rule out many downstream applications.

There is also no smoothness prior or regularization on the generated path, which means that they could be potentially quite noisy and highly non-smooth which in turn would deteriorate the realism of the generated data.
For this, one could compare path smoothness measures of real trajectories with generated trajectories.

Regarding realism, it is clear that the motion patterns real endoscopy camera paths are highly constraint by the physical properties for the endoscope and surrounding organ shape. None of these are modeled in the present formulation and I would expect significant domain gaps between simulated and real data which could for example effect camera pose estimation algorithms. This is certainly a limitation that could be added to the paper and addressed in future work.


- The contribution statement should be adapted to avoid over-claims like "simulation of [...] soft tissue via probe-tissue interaction and other intraoperative forces". While this work might enable such application which could also be mentioned in the intro, the paper does not discuss any of these things and cannot be considered a contribution.




**Relation To Prior Work:**

Prior work seems appropriately discussed.

**Summary And Contributions:**


This paper presents dataset for minimal-invasive surgery simulation that comprises of a large set of 3D rendering assets of human abdominal anatomy. Due to rendering simulation the dataset can be regarded as synthetic with the great advantage of providing perfect ground-truth data for camera pose, organ shapes, lighting as well as appearance and material properties  which is typically lacking for real medical data.
Nevertheless, the data is semi-real since all the organic shapes originate from true scans and derived organ segmentations.
A further contribution is a reinforcement learning-based camera path generation.

---

> ### Author Response · Authors · 2023-08-19
> **Response: Part 1**
>
> > From the paper it is unclear how the textures are generated, only a side-node in the
> conclusion mentions the procedural approach. To make the paper better self-contained, this should be
> mentioned earlier with an explicit reference to the suppl. material where further details are provided.
>
> We would like to thank the reviewer for their positive comments, and important improvement suggestions. We included a reference to the Supp. Mat. on the approach in L153 under Section 3.1 - "3D asset generation" in the paper, which describes the procedures for texture generation.
>
> We additionally point to Appendix D  'Procedural Texturing of Meshes (L600 Supp. Mat.) of the Supplementary Materials, where further details are included. Please also note the newly added Appendix D.1  'Introduction to Blender Shader Nodes' of the Supplementary Materials, which illustrates how Blender may be used to create example textures. Furthermore, we have expanded Appendix D.2 'Generation of \texttt{SARAMIS} textures', adding summaries of the Blender shading graph configurations for each of the textures.
>  Finally, due to the complexity of some of the Blender shading graph summaries, we have also added a reference to the open-source implementation provided in the \texttt{SARAMIS} repository, which contains the full parametrisation of each shader node as a .blend file.
>
> > The current evaluation of generated navigation paths is rather weak: Fig. 4 shows only 4 cherry-picked case examples. Although the reward function penalizes wall intersubsections, this does by no means guarantee they will not occur in the generated data. A statement or evaluation about the eventual number of generated intersubsections would be a critical addition to paper since they rule out many downstream applications.
>
> Thank you for these important considerations. First, we have now expanded the human-agent performance comparison from 4 to 24 cases so as to demonstrate a more representative comparison (L275).
> We find that the performance over 24 cases changes to $77.8 \pm 13.2$ and $75.3 \pm 15.6$ for the agent and human experiments, respectively, from the originally reported values of $73.5\pm11.1$ and $76.0\pm11.8$.
> There was no statistically significant difference in RL vs human performance over 24 cases (p-value$=0.83$).
> Our conclusions thus remain unchanged from the originally reported conclusions in the paper.
>
> Second, we would like to clarify that wall intersection or successful target localisation lead to episode termination, as can be seen in the provided supplementary material(code location: saramis/nav\_rl/env.py: L49-L60). That is to say that on L212 in the paper, $r_{image}$ is a terminal reward for the 'target in image $s_t$' and 'camera intersects with wall' conditions. The other rewards do not lead to episode termination. Such terminal conditions prevent wall intersections in the final trained model. We have now clarified this relevant information directly within the paper by specifying that the two signals are terminal signals in Equation 1 (L212) in the paper by modifying the definition of $r_{image}$.
>
> > There is also no smoothness prior or regularization on the generated path, which means that they could be potentially quite noisy and highly non-smooth which in turn would deteriorate the realism of the generated data. For this, one could compare path smoothness measures of real trajectories with generated trajectories.
>
> We thank the reviewer for their insightful comment.
> We provide an additional figure in Appendix E 'Trajectory Comparison Between Human and RL Agent
> ' (L670) of the Supplementary Materials, which showcases 5 comparisons between human and RL navigation trajectories on unseen test cases on a representative colonoscopy case (that is, insertion at the colon and navigation to the caecum).
> The figure suggests that the human navigation is smoother in comparison to the RL paths.
> As the reviewer has intuited, this may arise from the lack of a smoothness prior or regularization on the generated path.
>
> We make reference to this potential improvement in the limitations section of the paper (L311-312), and point to the additional figure in Appendix E from the Supplementary Materials in the manuscript.

---

> ### Author Response · Authors · 2023-08-19
> **Response: Part 2**
>
> > Regarding realism, it is clear that the motion patterns real endoscopy camera paths are highly constraint by the physical properties for the endoscope and surrounding organ shape. None of these are modeled in the present formulation and I would expect significant domain gaps between simulated and real data which could for example effect camera pose estimation algorithms. This is certainly a limitation that could be added to the paper and addressed in future work.
>
> We agree with the reviewer that not all constraints present in endoscopic procedures are modelled in our work. We have, however, modelled wall intersections, which are arguably the biggest constraint within such procedures. Additional constraints that could be modelled within future work are camera pose constraints e.g., camera may not face directly opposite to the endoscopy insertion path. This has now been mentioned in the discussion in L308 of the paper: 'While we model wall intersection constraints within our work, we do not account for all possible constraints in the endoscopic settings - for example, camera pose constraints such that the camera may not face directly opposite the direction of endoscope insertion from one step to the next, or extra-luminal boundaries imposed by surrounding visceral organs.
> Additionally, we qualitatively observe (Supp. Mat. Appendix E L670) that human trajectories are smoother than RL trajectories,
> which may arise from the lack of a smoothness prior or regularization on the generated actions.
> Accounting for these constraints represents a natural avenue of future research.
>
> >  The contribution statement should be adapted to avoid over-claims like "simulation of [...] soft tissue via probe-tissue interaction and other intraoperative forces". While this work might enable such application which could also be mentioned in the intro, the paper does not discuss any of these things and cannot be considered a contribution.
>
>  We acknowledge the reviewer's feedback, and have adjusted the statement to avoid over-claims not addressed in the work (L50).
>
> > Limitations are briefly discussed the last paragraph of the conclusion. I recommend extending this paragraph (see above) and make it more prominently visible with a paragraph header and separation from future work and concluding statements. No societal impact is discussed, but I do not expect negative impact from this work.
>
>  We thank the reviewer for this suggestion. We have modified the paper accordingly by separating the discussion, limitations, and conclusions into distinct sections (L292, L318, L331, respectively).
>
> > There is a detailed documentation in the supplementary material and download instructions to an AWS cloud bucket. It would be nice if this information could be copied to a dedicated website which makes this information searchable/findable/linkable in the internet.
>
>  We acknowledge the potential barrier to using AWS as a hosting platform long term.
> We have made the AWS bucket publically available at:  \url{https://saramis.s3.eu-north-1.amazonaws.com/}
>  Items in the bucket may be downloaded by adding the prefix associated to the file to be downloaded, as detailed in the Supplementary Materials Appendix A. 'Data Location And Code'.
>  Additionally, to make the dataset more visible, we have added the information to access the data to the \texttt{SARAMIS} repository, which has been added to the abstract of the paper: \url{https://github.com/NMontanaBrown/saramis/} (L11-13).
>
> > Since this dataset contains human data, ethics considerations are important. Importantly, the data is anonymous and since this datasets is sourced by existing other public datasets, it inherits their properties regarding ethics and others (e.g. privacy, IP, availability) In sum, I do not see any severe ethical issues.
>
>  We would like to highlight the ethics and licensing discussions provided in the Datasheet for Datasets (Supp. Mat. Appendix B), specifically at L389-L422 (Section 'Collection Process', 'Were any ethical review processes conducted (e.g., by an institutional review board)' subheading) on use of human data, and L483-521 (Section 'Distribution') on licensing.
>
> >  Please number all equations (e.g. rewards, policy), even if you do not reference an equation, others might want to so.
> L132: categorized -> categorised (AE vs. BE consistency)
>
>  We have fixed the AE vs BE for consistency at L135, and added number to the equations in the paper.

---

> ### Comment · Reviewer_Mims · 2023-08-21
>
> I thank the authors for their great effort to respond to all reviewer concerns in detail.
>
> I think the authors clarified most concerns well and I am happy about the changes and improvements to the paper.
> Of course there are remaining weaknesses like the texture generation, but these are research problems for themselves and non-trivial to solve properly within the scope of this paper. In my view the paper's contributions are sufficient, the dataset is already valuable and well enough documented to be immediately usable by the community.
>
> Overall, I am happy with the author's rebuttal and I keep my recommendation to accept the paper.

---

### Author Response · Authors · 2023-08-19

We thank the reviewers for their insightful comments on the paper. Reviewers have praised the paper’s uniqueness and novelty with relation to datasets in the prior art [1urX+zAJD], the sound and thorough dataset construction methodology [Mims+1urX], and the high relevance of curation efforts [Mims+1urX+zAJD], especially for the medical and biomedical community [Mims+1urX]. Furthermore, reviewers highlighted the interesting and important application of the experiment of autonomous navigation for colonoscopy [PbKg+1urX]. Reviewers have additionally noted the paper is well-written [Mims+1urX+zAJD] and clear [Mims+PbKg+1urX+zAJD]. Finally, reviewers commented the paper is well-contextualised within the scope of the literature [Mims+1urX+zAJD]. We have ensured easy public accessibility to the SARAMIS dataset in full (more details below and in the manuscript). We kindly refer the reviewers to their individual sections for our rebuttal points.

---

### Decision · Program_Chairs · 2023-09-22

**Decision:**

Accept (Poster)

**Comment:**

Submission: SARAMIS (**#932**) Bottom line: **accept (as poster)**

**AC Special Note**: one out of the four reviewers obviously believes the authors did not engage in (sufficiently original) work to warrant consideration (score of 2), whereas the three other reviewers (scores of 7, 8, and 7) believe the work to present novel and clearly original contributions to the field; the **AC chose to accept (poster), with a confidence of 3 (somewhat confident), and would appreciate the SAC/PC to take a look at whether or not the lowest-scoring review (Reviewer PbKg) is yet a (more more credible) domain expert, and thus should carry the decision**.

**Summary**: the authors combined pre-existing datasets of primarily soft-abdominal-tissue CT scans and then **created additional digital assets together with a clear outline of use cases** for the support of developing test environments for minimally invasive surgery. The assets are well described and tested; however, the final application (fully implemented test environment for MIS) is not provided, and as such it is difficult to adjudicate just how soon this dataset will see a definitive use case (as intended). As such, next to the Special Note above, this dataset might also be more appropriate for a "surgery specialty journal", compared to the broad domain-nonspecific NeurIPS...

Pros: from reading the paper and looking through the assets in general, it seems obvious that the authors have spent quite some time preparing, describing, and working with the data they put together

Cons: while the application (simulating minimally invasive surgery) is conceptually clear, and a dataset of assets for this purpose is needed, it is somewhat unclear to the AC how much work the creation of such a simulation environment entails, and whether it is primarily restricted by the lack of a comprehensive dataset (as presented) or rather by the difficulty of developing such a system for other reasons; as such the AC finds it hard to judge the **relative value** of this dataset for the community beyond being a genuinely fascinating case of taking existing datasets and creating a (labor-intensive!) derivative dataset of assets. The value would also seem primarily for a surgical robotics community rather than within broader machine-learning or computer vision. Given my lack of domain knowledge (no experience with how CV/machine learning is applied **in the field of surgical robotic simulation**!), the AC is only somewhat confident, relying on the median (three out of four reviewers scored 7 or higher) rating.

Decision: the AC follows the median reviewer in proposing accepting the dataset -- under the proviso that the SAC and PCs confirm that the outlier review (Reviewer PbKg) isn't actually the person with the strongest domain expertise.

----

Reviewer Mims: the limitations section was highlighted by the authors (as requested by the reviewer), and other concerns seem to have been addressed satisfactorily

----

Reviewer PbKg (**high-confidence outlier with score 2**): From the entire (very lengthy) exchange between the reviewer and authors, my read is that the reviewer infers a lack of authors' contribution to **novelty** w.r.t. dataset and methods. The back and forth hinges on whether or not the work of the authors, as presented, does in fact correspond to a dataset asset that, in its present form, is "missing" from the literature. In my general opinion (as AC, without deep domain knowledge), I would say that the other three reviewers' score suggests to me that this reviewer either has a very different idea of a standard for "novel, original, and significant contribution", or has a strong domain expertise not shared by the other reviewers; it is obvious from the unchanged score that the reviewer does not believe that the authors have sufficiently addressed the concerns. As AC, I took the liberty to override this reviewer's score, but see Special Note and my other comments above!

----

Reviewer 1urX: it seems that the concerns raised were addressed, and another reviewer commented that the authors' clarifications also increased their score.

----

Reviewer zAJD: offered **many** points, but seems very satisfied with the overall rebuttals and how authors addressed the concerns.

----

In summary, I would kindly ask the SAC and PCs to take a look at the low-scoring reviewer's comments and, together with the question of fit (surgical robotics audience) make a final decision as to whether or not to accept this contribution.